# Diminishing benefits of urban living for children and adolescents' growth and development

**NCD Risk Factor Collaboration (NCD-RisC)\***

Optimal growth and development in childhood and adolescence is crucial for lifelong health and well-being[1–6]. Here we used data from 2,325 population-based studies, with measurements of height and weight from 71 million participants, to report the height and body-mass index (BMI) of children and adolescents aged 5–19 years on the basis of rural and urban place of residence in 200 countries and territories from 1990 to 2020. In 1990, children and adolescents residing in cities were taller than their rural counterparts in all but a few high-income countries. By 2020, the urban height advantage became smaller in most countries, and in many high-income western countries it reversed into a small urban-based disadvantage. The exception was for boys in most countries in sub-Saharan Africa and in some countries in Oceania, south Asia and the region of central Asia, Middle East and north Africa. In these countries, successive cohorts of boys from rural places either did not gain height or possibly became shorter, and hence fell further behind their urban peers. The difference between the age-standardized mean BMI of children in urban and rural areas was <1.1 kg m$^{-2}$ in the vast majority of countries. Within this small range, BMI increased slightly more in cities than in rural areas, except in south Asia, sub-Saharan Africa and some countries in central and eastern Europe. Our results show that in much of the world, the growth and developmental advantages of living in cities have diminished in the twenty-first century, whereas in much of sub-Saharan Africa they have amplified.

The growth and development of school-aged children and adolescents (ages 5–19 years) are influenced by their nutrition and environment at home, in the community and at school. Healthy growth and development at these ages help consolidate gains and mitigate inadequacies from early childhood and vice versa[1], with lifelong implications for health and well-being[2–6]. Until recently, the growth and development of older children and adolescents received substantially less attention than in early childhood and adulthood[7]. Increasing attention on the importance of health and nutrition during school years has been accompanied by a presumption that differences in nutrition and the environment lead to distinct, and generally less healthy, patterns of growth and development at these ages in cities compared to rural areas[8–17]. This presumption is despite some empirical studies showing that food quality and nutrition are better in cities[18,19].

Data on growth and developmental outcomes during school ages are needed, alongside data on the efficacy of specific interventions and policies, to select and prioritize policies and programmes that promote health and health equity, both for the increasing urban population and for children who continue to grow up in rural areas. Consistent and comparable global data also help benchmark across countries and territories and draw lessons on good practice. Yet, globally, there are fewer data on growth trajectories in rural and urban areas in these formative ages than

for children under 5 years of age[20] or for adults[21]. The available studies have been in one country, at one point in time and/or in one sex and narrow age groups. The few studies that covered more than one country[22–24] mostly focused on older girls and used at most a few dozen data sources and hence could not systematically measure long-term trends. Consequently, many policies and programmes that aim to enhance healthy growth and development in school ages focus narrowly and generically on specific features of nutrition or the environment in either cities or rural areas[10,13,25–28]. Little attention has been paid to the similarities and differences between relevant outcomes in these settings or to the heterogeneity of the urban–rural differences across countries.

Here we report on the mean height and BMI of school-aged children and adolescents residing in rural and urban areas of 200 countries and territories (referred to as countries hereafter) from 1990 to 2020. Height and BMI are anthropometric measures of growth and development that are influenced by the quality of nutrition and healthiness of the living environment and are highly predictive of health and well-being throughout life in observational and Mendelian randomization studies[2–6]. These studies have shown that having low height and excessively low BMI increases the risk of morbidity and mortality, and low height impairs cognitive development and reduces educational performance and work productivity in later life[2–4]. A high BMI in these

\*A list of authors and their affiliations appears online.

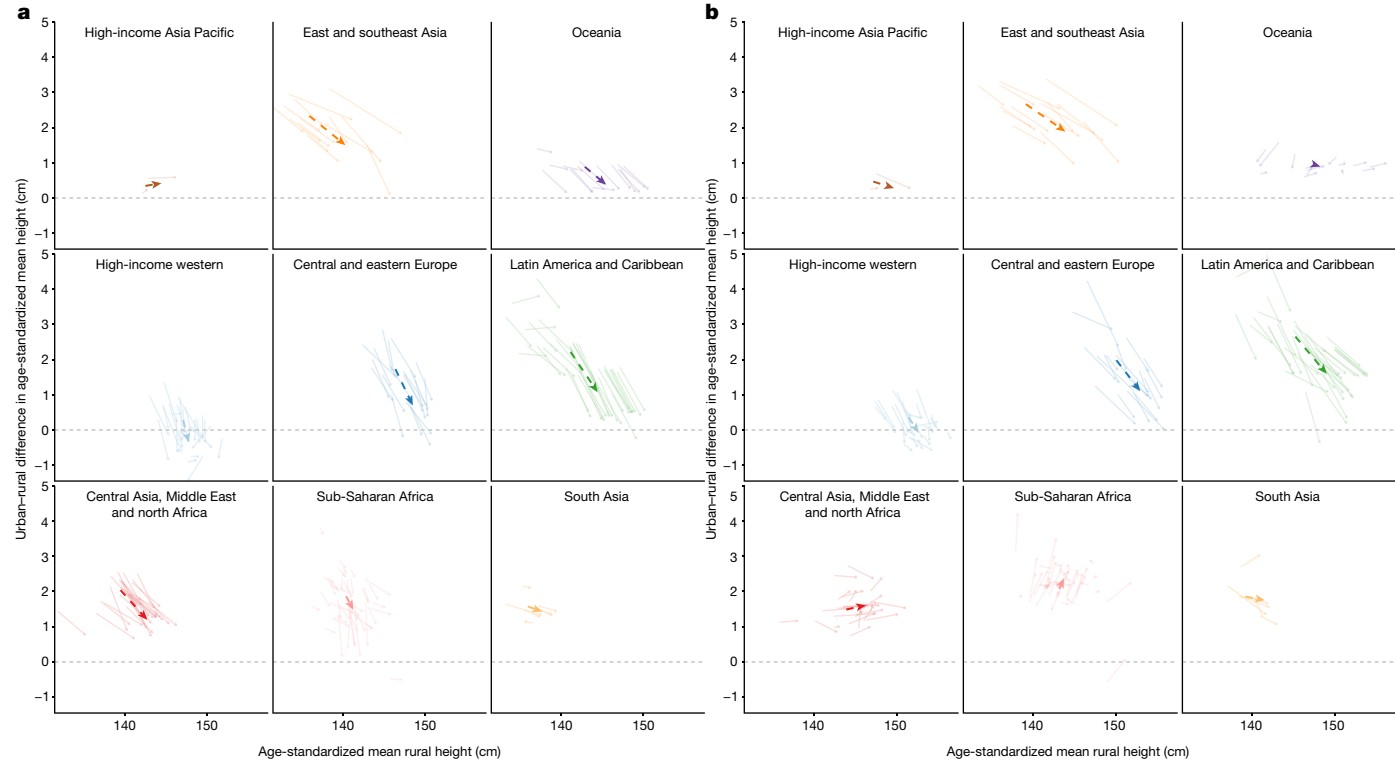

**Fig. 1 | Change in the urban–rural height difference from 1990 to 2020.**
**a**,**b**, Change in the urban–rural difference in age-standardized mean height in relation to the change in age-standardized mean rural height in girls (**a**) and boys (**b**). Each solid arrow in lighter shade shows one country beginning in 1990 and ending in 2020. The dashed arrows in darker shade show the regional averages, calculated as the unweighted arithmetic mean of the values for all countries in each region along the horizontal and vertical axes. For the urban–rural difference, a positive number shows a higher urban mean height and a

negative number shows higher rural mean height. See Extended Data Fig. 2 for urban–rural differences in age-standardized mean height and their change over time shown as maps, together with uncertainties in the estimates. See Supplementary Fig. 4a for results at ages 5, 10, 15 and 19 years. We did not estimate the difference between rural and urban height for countries classified as entirely urban (Bermuda, Kuwait, Nauru and Singapore) or entirely rural (Tokelau).

ages increases the lifelong risk of overweight and obesity and several non-communicable diseases, and might contribute to poor educational outcomes[5,6].

We used 2,325 population-based studies that measured height and weight in 71 million participants in 194 countries (Extended Data Fig. 1 and Supplementary Table 2). We used these data in a Bayesian hier-archical meta-regression model to estimate mean height and BMI of children and adolescents aged 5–19 years by rural and urban place of residence, year and age for 200 countries. Details of data sources and statistical methods are provided in the Methods. Our results represent the height and BMI for children and adolescents of the same age over time (that is, successive cohorts) in rural and urban areas of each country, and the difference between the two. For presentation, we summarize the 15 age-specific estimates, for single years of age from 5 to 19, through age standardization, which puts each country-year's child and adolescent population on the same age distribution and enables comparisons to be made over time and across countries. We also show results, graphically and numerically, for index ages of 5, 10, 15 and 19 years in the Supplementary Information.

In 1990, school-aged boys and girls who lived in cities had a height advantage (that is, were taller) compared with their rural counterparts. The exception was in high-income countries, where the urban height advantage was either negligible (<1.2 cm for age-standardized mean height; posterior probability (PP) for children living in urban areas being taller ranging from 0.51 to >0.99) or there was a small rural advantage (for example, Belgium, the Netherlands and the United Kingdom) (PP for children in rural areas being taller ranging from 0.53 to >0.95 where there was a rural height advantage) (Fig. 1 and Extended Data Fig. 2). The

largest height differences between children and adolescents in cities and rural areas in 1990 occurred in some countries in Latin America (for example, Mexico, Guatemala, Panama and Peru), east and southeast Asia (China, Indonesia and Vietnam), central and eastern Europe (Bulgaria, Hungary and Romania) and sub-Saharan Africa (Democratic Republic of Congo (DR Congo) and Rwanda). The urban height advantage in boys and girls in the named countries ranged from 2.4 to 5.0 cm, and the PP of children living in urban areas being taller than children living in rural areas was >0.99 (see Supplementary Table 3 for country-specific numerical values of height in children living in rural versus urban areas, their difference and the corresponding credible intervals (CrIs)).

The urban–rural height gap in the late twentieth century among low-income and middle-income countries was determined by how much children and adolescents in cities and rural areas had approached as opposed to fallen behind their peers in high-income countries, where there was little difference between urban and rural height. In countries such as Bulgaria, Hungary and Romania, the height of children and adolescents living in urban areas approached that of high-income coun-tries, whereas children and adolescents in rural areas lagged behind, leading to a relatively large gap. In much of sub-Saharan Africa and south Asia, the height of children and adolescents lagged behind their peers in high-income countries regardless of where they lived, such that the urban–rural gap was relatively small. In a third group of low-income or middle-income countries that included Indonesia, Vietnam, Panama, Peru, DR Congo and Rwanda, children living in urban areas remained shorter than in high-income countries, but children from rural areas lagged even further behind, such that the urban–rural gap became large.

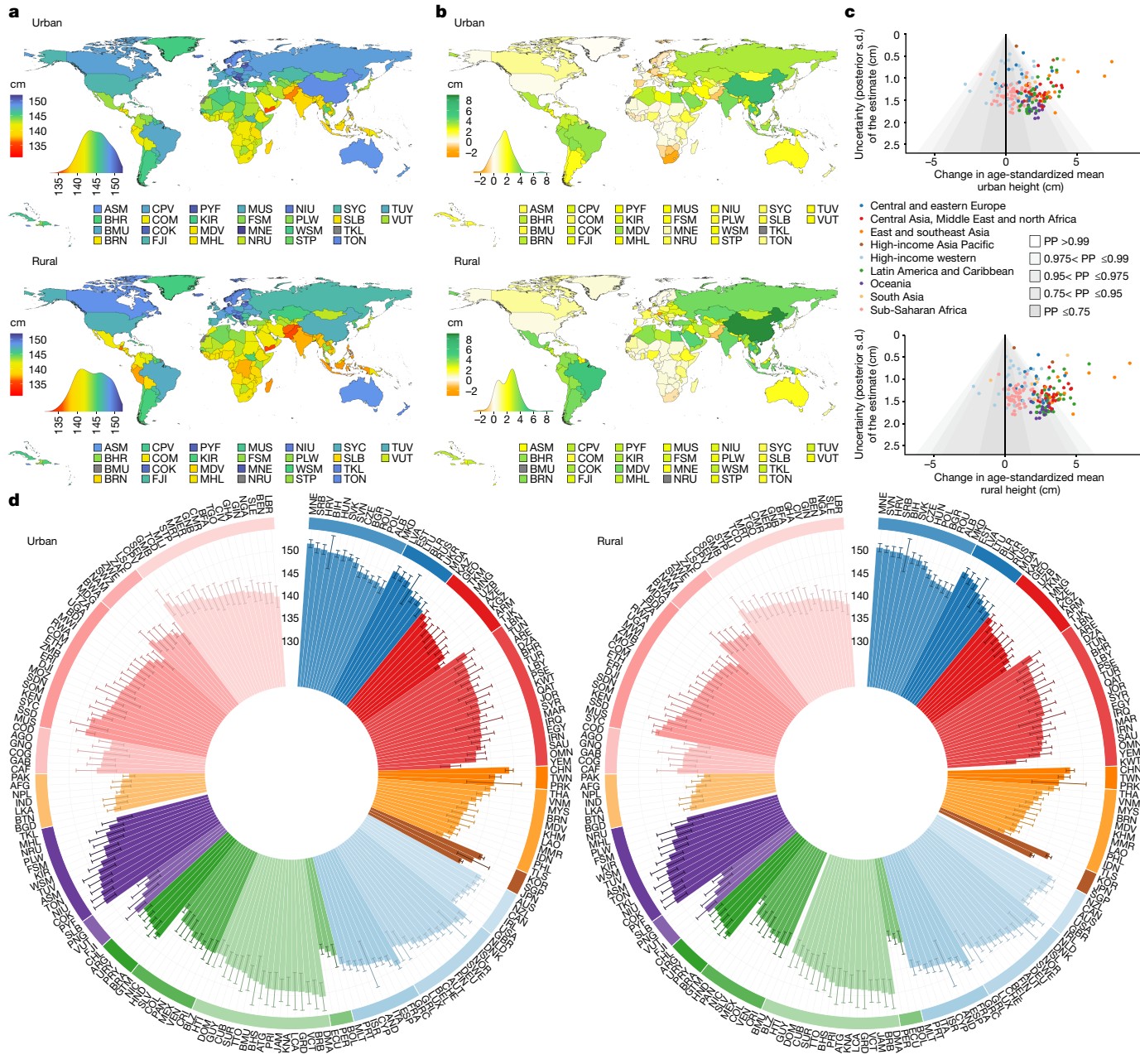

**Fig. 2** | See next page for caption.

By 2020, the urban height advantage in school ages became smaller in much of the world. In many high-income western countries and some central European countries, it disappeared or reversed into a small (typically <1 cm) urban disadvantage (Fig. 1 and Extended Data Figs. 2 and 8). Countries with substantial convergence over these three decades were in central and eastern Europe (for example, Croatia), Latin America and the Caribbean (for example, Argentina, Brazil, Chile and Paraguay), east and southeast Asia (for example, Taiwan) and for girls in central Asia (for example, Kazakhstan and Uzbekistan). The urban height advantage in the named countries declined by around 1–2.5 cm from 1990 to 2020 (the PP of urban–rural height difference having declined ≥0.90 for the named countries). In many other middle-income countries (for example, China, Romania and Vietnam), the urban–rural height gaps declined, but children and adolescents living in cities remained taller than their rural counterparts (by 1.7–2.5 cm in the named countries for boys and girls; the PP of children in cities being taller than children in rural areas in 2020 >0.99). The exception to this

convergence was for boys in most countries in sub-Saharan Africa and some countries in Oceania, south Asia and the region of central Asia, Middle East and north Africa, where the urban height advantage slightly increased over these three decades. The largest increase in the urban height advantage for boys occurred in countries in east Africa such as Ethiopia (0.9 cm larger height gap in 2020 than 1990; 95% CrI −0.9 to 2.9, and PP of an increase of 0.86), Rwanda (1.0 cm larger gap, 95% CrI −0.7 to 3.0, and PP 0.88) and Uganda (1.1 cm larger gap, 95% CrI of −0.6 to 3.1, and PP 0.89). For girls, the urban–rural gap remained largely unchanged in many countries in sub-Saharan Africa and south Asia.

In middle-income countries and emerging economies (newly high-income and industrialized countries) where the height of children and adolescents residing in rural areas converged to those in cities, successive cohorts of children and adolescents living in rural areas outpaced their urban counterparts in becoming taller and attained heights that urban children in the same countries had done decades earlier: growing to heights closer to those seen in high-income countries

**Fig. 2 | Urban and rural height in 2020 and the change from 1990 to 2020 for girls. a**, Age-standardized mean height in 2020 by urban and rural place of residence for girls. The density plots show the distribution of estimates across countries. **b**, Age-standardized change in mean height from 1990 to 2020 by urban and rural place of residence for girls. The density plots show the distribution of estimates across countries. **c**, Change in mean height from 1990 to 2020 in relation to the uncertainty of the change measured by posterior standard deviation. Each point in the scatter plots shows one country. Shaded areas approximately show the PP of an estimated change being a true increase or decrease. The PP of a decrease is one minus that of an increase. If an increase in mean height is statistically indistinguishable from a decrease, the PP of an increase and a decrease is 0.50. PPs closer to 0.50 indicate more uncertainty, whereas those towards 1 indicate more certainty of change. **d**, Age-standardized mean height in 2020 for all countries. The height of each column is the posterior mean estimate shown together with its 95% CrI. Countries are ordered by region and super-region. See Extended Data Fig. 4 for a map of PP of the estimated change. See Supplementary Fig. 5 for results at ages 5, 10, 15 and 19 years. See Supplementary Table 3 for numerical results, including CrIs, as age-standardized and at ages 5, 10, 15 and 19 years. We did not estimate mean rural height in countries classified as entirely urban (Bermuda, Kuwait, Nauru and Singapore), mean urban height in countries classified as entirely rural (Tokelau) or their change over time in these countries, as indicated in grey. Countries are labelled using their International Organization for Standardization (ISO) 3166-1 alpha-3 codes. Afghanistan, AFG; Albania, ALB; Algeria, DZA; American Samoa, ASM; Andorra, AND; Angola, AGO; Antigua and Barbuda, ATG; Argentina, ARG; Armenia, ARM; Australia, AUS; Austria, AUT; Azerbaijan, AZE; Bahamas, BHS; Bahrain, BHR; Bangladesh, BGD; Barbados, BRB; Belarus, BLR; Belgium, BEL; Belize, BLZ; Benin, BEN; Bermuda, BMU; Bhutan, BTN; Bolivia, BOL; Bosnia and Herzegovina, BIH; Botswana, BWA; Brazil, BRA; Brunei Darussalam, BRN; Bulgaria, BGR; Burkina Faso, BFA; Burundi, BDI; Cabo Verde, CPV; Cambodia, KHM; Cameroon, CMR; Canada, CAN; Central African Republic, CAF; Chad, TCD; Chile, CHL; China, CHN; Colombia, COL; Comoros, COM; Congo, COG; Cook Islands, COK; Costa Rica, CRI; Cote d'Ivoire, CIV; Croatia, HRV; Cuba, CUB; Cyprus, CYP; Czechia, CZE; Denmark, DNK; Djibouti, DJI; Dominica, DMA; Dominican Republic, DOM; DR Congo, COD; Ecuador, ECU; Egypt, EGY; El Salvador, SLV; Equatorial Guinea, GNQ; Eritrea, ERI; Estonia, EST; Eswatini, SWZ; Ethiopia, ETH; Fiji, FJI; Finland, FIN; France, FRA; French Polynesia, PYF; Gabon, GAB; Gambia, GMB; Georgia, GEO; Germany, DEU; Ghana, GHA; Greece, GRC; Greenland, GRL; Grenada, GRD; Guatemala, GTM; Guinea Bissau, GNB; Guinea, GIN; Guyana, GUY; Haiti, HTI; Honduras, HND; Hungary, HUN; Iceland, ISL; India, IND; Indonesia, IDN; Iran, IRN; Iraq, IRQ; Ireland, IRL; Israel, ISR; Italy, ITA; Jamaica, JAM; Japan, JPN; Jordan, JOR; Kazakhstan, KAZ; Kenya, KEN; Kiribati, KIR; Kuwait, KWT; Kyrgyzstan, KGZ; Lao PDR, LAO; Latvia, LVA; Lebanon, LBN; Lesotho, LSO; Liberia, LBR; Libya, LBY; Lithuania, LTU; Luxembourg, LUX; Madagascar, MDG; Malawi, MWI; Malaysia, MYS; Maldives, MDV; Mali, MLI; Malta, MLT; Marshall Islands, MHL; Mauritania, MRT; Mauritius, MUS; Mexico, MEX; Micronesia (Federated States of), FSM; Moldova, MDA; Mongolia, MNG; Montenegro, MNE; Morocco, MAR; Mozambique, MOZ; Myanmar, MMR; Namibia, NAM; Nauru, NRU; Nepal, NPL; Netherlands, NLD; New Zealand, NZL; Nicaragua, NIC; Niger, NER; Nigeria, NGA; Niue, NIU; North Korea, PRK; North Macedonia, MKD; Norway, NOR; Occupied Palestinian Territory, PSE; Oman, OMN; Pakistan, PAK; Palau, PLW; Panama, PAN; Papua New Guinea, PNG; Paraguay, PRY; Peru, PER; Philippines, PHL; Poland, POL; Portugal, PRT; Puerto Rico, PRI; Qatar, QAT; Romania, ROU; Russian Federation, RUS; Rwanda, RWA; Saint Kitts and Nevis, KNA; Saint Lucia, LCA; Samoa, WSM; Sao Tome and Principe, STP; Saudi Arabia, SAU; Senegal, SEN; Serbia, SRB; Seychelles, SYC; Sierra Leone, SLE; Singapore, SGP; Slovakia, SVK; Slovenia, SVN; Solomon Islands, SLB; Somalia, SOM; South Africa, ZAF; South Korea, KOR; South Sudan, SSD; Spain, ESP; Sri Lanka, LKA; Saint Vincent and the Grenadines, VCT; Sudan, SDN; Suriname, SUR; Sweden, SWE; Switzerland, CHE; Syrian Arab Republic, SYR; Taiwan, TWN; Tajikistan, TJK; Tanzania, TZA; Thailand, THA; Timor-Leste, TLS; Togo, TGO; Tokelau, TKL; Tonga, TON; Trinidad and Tobago, TTO; Tunisia, TUN; Turkey, TUR; Turkmenistan, TKM; Tuvalu, TUV; Uganda, UGA; Ukraine, UKR; United Arab Emirates, ARE; United Kingdom, GBR; United States of America, USA; Uruguay, URY; Uzbekistan, UZB; Vanuatu, VUT; Venezuela, VEN; Vietnam, VNM; Yemen, YEM; Zambia, ZMB.

(Figs. 2 and 3). Successive cohorts of children and adolescents residing in rural areas in sub-Saharan Africa did not experience the accelerated height gain seen in cohorts in rural areas of middle-income countries. Notably, in the case of boys living in sub-Saharan Africa, there was no gain, or possibly a decrease, in height, which in turn led to a persistence or even widening of the urban–rural gap. As a result of these global trends, by 2020, the largest urban–rural gaps in height were seen in Andean and central Latin America (for example, Bolivia, Panama and Peru, by up to 4.7 cm (95% CrI 4.0–5.5 cm) for boys and 3.8 cm (95% CrI 3.3–4.3 cm) for girls) and, especially for boys, in sub-Saharan Africa (for example, DR Congo, Ethiopia, Mozambique and Rwanda, by up to 4.2 cm (95% CrI 2.7–5.7 cm)).

The urban–rural BMI difference was relatively small throughout these three decades, <1.4 kg m$^{-2}$ in all countries and years and <1.1 kg m$^{-2}$ in all but nine countries, for age-standardized mean BMI (Fig. 4 and Extended Data Figs. 3 and 9). In 1990, the urban–rural BMI gap was largest in sub-Saharan Africa (for example, Ethiopia, Kenya, Malawi, South Africa and Zimbabwe) and south Asia (for example, Bangladesh and India), followed by parts of Latin America (for example, Mexico and Peru). The urban–rural BMI gap in the two sexes in the named countries ranged from 0.4 to 1.2 kg m$^{-2}$, and the PP of children and adolescents living in urban areas having a higher BMI than those in rural areas was ≥0.89. At that time, girls and/or boys in rural areas of some of these countries had mean BMI levels that were close to, and in some ages below, the thresholds of being underweight (>1 s.d. below the median of the World Health Organization (WHO) reference population).

From 1990 to 2020, the BMI of successive cohorts of children and adolescents in both urban and rural areas increased in all but a few mostly high-income countries (for example, Denmark, Italy and Spain) (Figs. 5 and 6). There was heterogeneity in low-income and middle-income countries in how much the BMI increased in cities compared with rural areas. In the majority of countries in sub-Saharan Africa and south Asia, the BMI of successive cohorts of children and adolescents increased more in rural areas than in cities, leading to a closing of the urban–rural difference. The urban–rural BMI gap declined by up to 0.65 kg m$^{-2}$ for both girls and boys, and the PP that the urban–rural BMI difference declined from 1990 to 2020 ranged from 0.52 to 0.95. In both sub-Saharan Africa and south Asia, these changes shifted the mean BMI of boys and girls in rural areas out of the range for being underweight. Moreover, in many countries in sub-Saharan Africa, this shift continued beyond the median of the WHO reference population and in some cases approached the threshold for being overweight (>1 s.d. above the median of the WHO reference population). The opposite, a larger increase in urban BMI, happened in most other low-income and middle-income countries, leading to a slightly larger urban BMI excess in 2020 than in 1990. High-income countries and those in central and eastern Europe experienced a mix of increasing and decreasing urban BMI excess, but remained within a small range (−0.3 to 0.6 kg m$^{-2}$ for almost all countries) over the entire period of analysis. At the regional level, the urban–rural BMI difference changed by <0.25 kg m$^{-2}$ in these regions.

The urban height advantage was larger in boys than girls in most countries (Supplementary Fig. 3). Urban excess BMI was larger in boys than girls in only about one-half of the countries. For the other half, mostly in high-income western countries and those in sub-Saharan Africa, urban excess BMI was larger in girls than boys. The urban height advantage was slightly larger at 5 years of age than at 19 years of age in most low-income and middle-income countries, especially for girls, but there was little difference across ages in high-income regions and in central and eastern Europe (Supplementary Fig. 4).

Since the introduction of modern sanitation in the nineteenth century, cities provided substantial nutritional and health advantages in high-income and subsequently low-income and middle-income countries[19]. Our results show that in the twenty-first century, during school

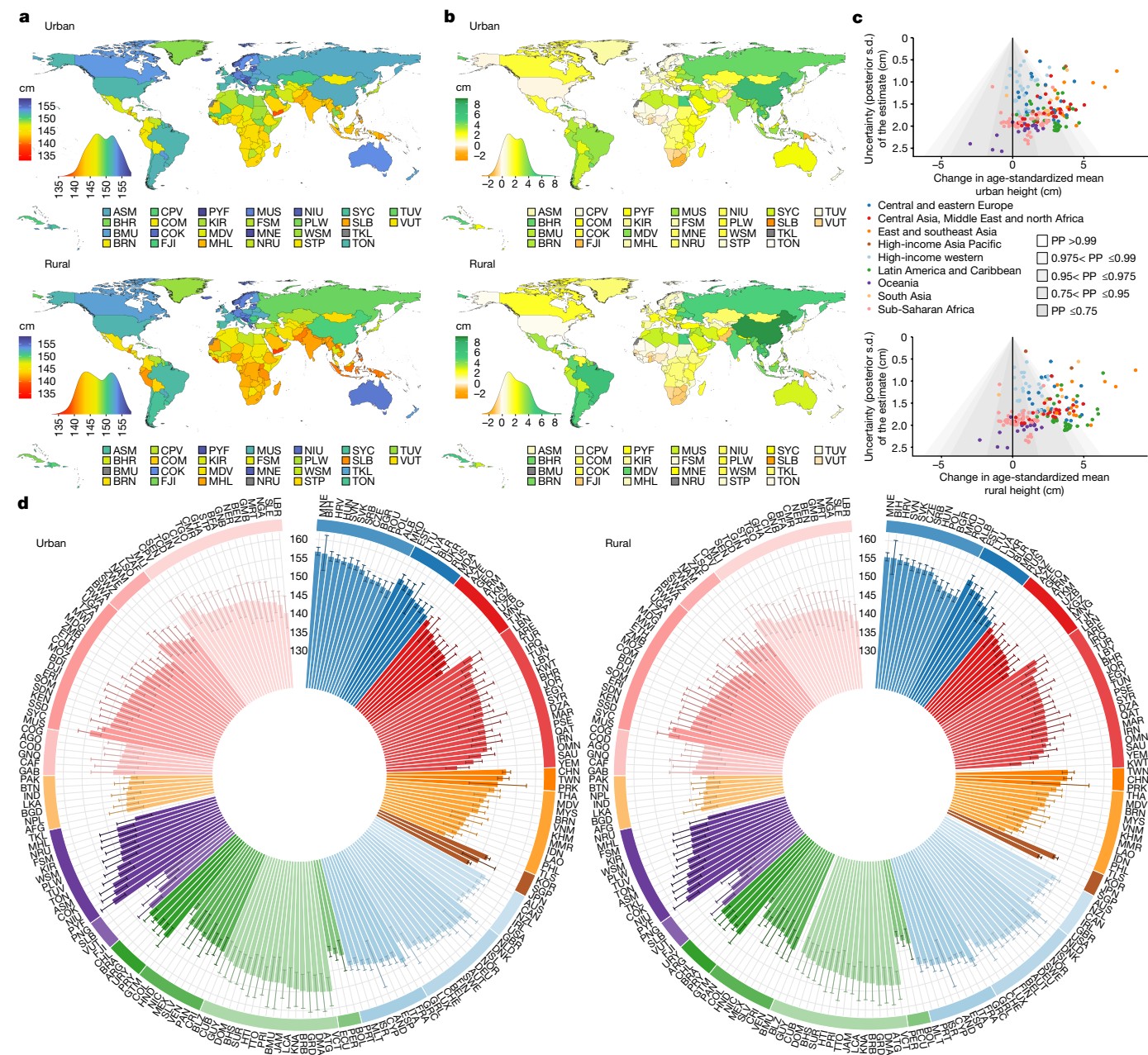

**Fig. 3 | Urban and rural height in 2020 and change from 1990 to 2020 for boys. a–d**, See the caption for Fig. 2 for descriptions of the contents of the figure and for definitions. We did not estimate mean rural height in countries classified as entirely urban (Bermuda, Kuwait, Nauru and Singapore), mean urban height in countries classified as entirely rural (Tokelau) or their change over time, as indicated in grey.

ages, these advantages have disappeared in high-income countries and diminished in middle-income countries and emerging economies in Asia, Latin America and the Caribbean, and parts of Middle East and north Africa. Specifically, in these settings, successive cohorts of school-aged children and adolescents living in cities were outpaced by those in rural areas in terms of height gain but gained slightly more weight by 2020, typically in the unhealthy range (Fig. 7). This contrasted with the poorest region in the world: sub-Saharan Africa. In this region, the urban height advantage persisted or even expanded, whereas rural mean BMI went beyond remedying underweight and surpassed the median of the WHO reference population in 2020, hence consolidating the urban advantage. South Asia had a mixed pattern of urban versus rural trends from 1990 to 2020, with children and adolescents in rural areas gaining both more height and more weight for their height than those in cities. Notably, our

results also show that differences in height and BMI between urban and rural populations within most countries are smaller than the differences across countries, even those in the same region.

We also found that the urban–rural BMI gap, although dynamic, changed much less than the BMI of either subgroup of the population and less than commonly assumed when discussing the role of cities in the obesity epidemic[8,10,12,13,15,16]. Urban–rural BMI differences were especially small in high-income countries, which is consistent with evidence from a few countries that show diets and behaviours are affected more by household socioeconomic status than whether children and adolescents live in cities or rural areas[29,30]. Urban BMI excess increased slightly more in middle-income countries in east and southeast Asia, Latin America and the Caribbean, and Middle East and north Africa, a trend that was the opposite of the convergence in BMI of

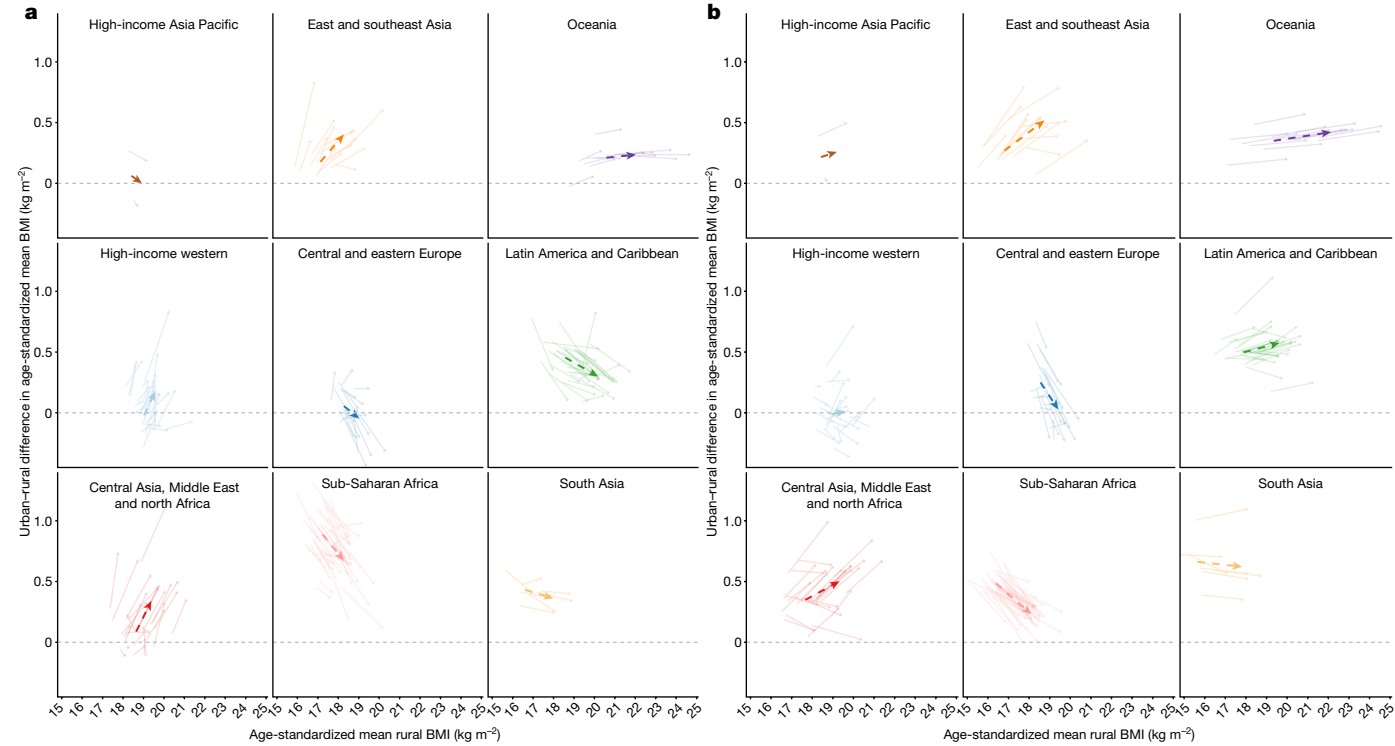

**Fig. 4 | Change in the urban–rural BMI difference from 1990 to 2020.**
**a,b**, Change in urban–rural difference in age-standardized mean BMI for girls
(**a**) and boys (**b**) in relation to change in age-standardized mean rural BMI. See
the caption for Fig. 1 for a description of the contents of this figure. See Extended
Data Fig. 3 for urban–rural differences in age-standardized mean BMI and their

change over time shown as maps, together with uncertainties in the estimates.
See Supplementary Fig. 4b for results at ages 5, 10, 15 and 19 years. We did not
estimate the difference between rural and urban BMI for countries classified as
entirely urban (Bermuda, Kuwait, Nauru and Singapore) or entirely rural
(Tokelau).

adults in these same regions[21]. Additional analyses of data collected by
the NCD Risk Factor Collaboration (NCD-RisC) for young adults (20–29
and 30–39 years) showed that the shift from a small divergent trend to
convergence of BMI between urban and rural areas happens in young
adulthood (Extended Data Figs. 6 and 7), a period during which there is
substantial, but variable, weight gain among population subgroups[31].
These shifts in trends from adolescence to young adulthood might be
a result of changes in diet and energy expenditure that accompany
changes in household structure, social and economic roles and the
living environment[32–34].

Long-term follow-up studies have shown that children and adoles-
cents do not achieve their height potential if they do not consume suf-
ficient and diverse nutritious foods or if they are exposed to repeated
or persistent infections, which result in loss of nutrients[2]. Studies that
use data on household socioeconomic and environmental factors have
indicated that these physiological determinants of height are them-
selves affected by income, education, quality of the living environment
and access to healthcare in rural as well as urban areas[35]. This evidence
indicates that the relatively small urban–rural height differentials in
high-income countries may be because of a greater abundance of
nutritious foods, including some fortified foods, better education
and healthcare and greater ability to finance programmes that promote
healthy growth in countries with greater per-capita income and better
infrastructure. Variations across these countries in the urban–rural
height gap within this small range may be due to the extent of socioeco-
nomic inequalities and poverty, differences in the availability and cost
of nutritious foods between cities and rural areas and whether there
are specific programmes (for example, food assistance or school food
programmes) that improve nutrition in disadvantaged groups[30,36,37]. The
more marked changes in height in urban versus rural areas took place
in middle-income countries and emerging economies. Case studies in

some countries where the heights of children and adolescents living in
rural and urban areas converged show that the convergence was partly
due to using the growth in national income towards programmes and
services that helped close gaps in nutrition, sanitation and healthcare
between different areas and social groups[38–40]. In countries in central
and eastern Europe, transition to a market economy and increases in
trade may have reduced the disparity in access to, and seasonality of,
healthy foods between urban and rural areas[41], and partly underlie the
convergence of height seen in our results. By contrast, case studies in
some countries have shown that where economic growth was accom-
panied by large inequalities in income, nutrition and/or services, the
urban advantage persisted[42–44].

The notable exception in the global trends was sub-Saharan Africa,
where a stagnation or reversal of height gain in rural areas led to the
persistence or widening of urban–rural height differences, whereas the
opposite happened for BMI (Fig. 7). Case studies of specific countries
have indicated that unfavourable trends in nutrition in rural Africa,
where the majority of the poorest people in the world live, started from
macroeconomic shocks in the late twentieth century[45] and subsequent
agriculture, trade and development policies that limited improvements
in income and services, and emphasized agricultural exports over local
food security and diversity[45]. These macroeconomic factors in turn
led to less diverse diets, with higher caloric intake rather than a shift
to protein-rich and nutrient-rich foods (for example, animal products,
seafood, fruits and vegetables)[46–48]. Moreover, the slow expansion of
infrastructure and services in rural areas restricted improvements in
other determinants of healthy growth, such as clean water, sanitation
and health care[49].

Several other factors may have had a secondary role in the observed
trends in height and BMI and their difference in rural and urban areas.
First, weight gain during childhood may reduce the age of puberty

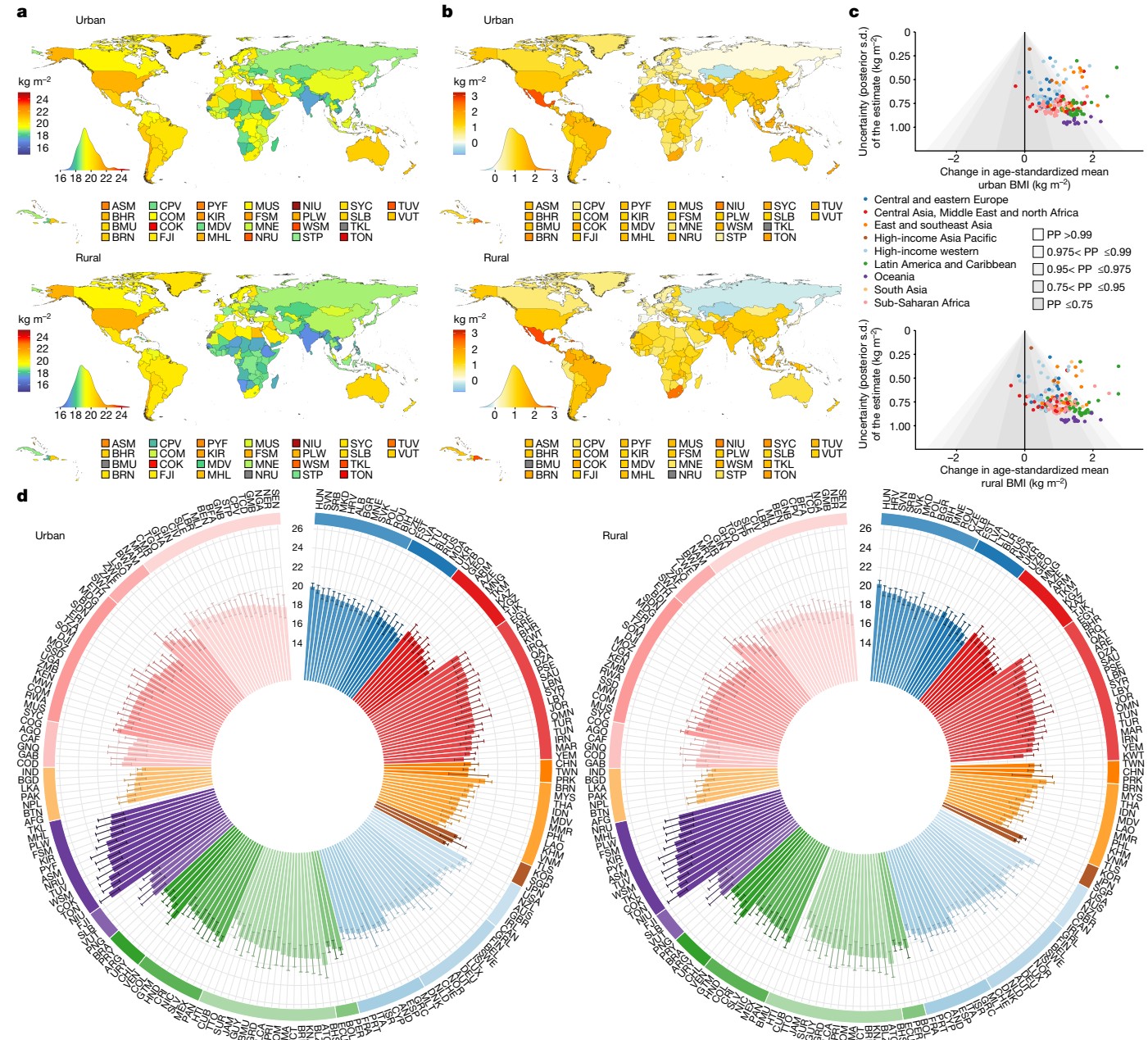

**Fig. 5 | Urban and rural BMI in 2020 and change from 1990 to 2020 for girls. a–d**, See the caption for Fig. 2 for descriptions of the contents of the figure and for definitions. See Extended Data Fig. 5 for a map of PP of the estimated change. See Supplementary Fig. 6 for results at ages 5, 10, 15 and 19 years. See Supplementary Table 4 for numerical results, including CrIs, as age-standardized and at ages 5, 10, 15 and 19 years. We did not estimate mean rural BMI in countries classified as entirely urban (Singapore, Bermuda and Nauru), mean urban BMI in areas classified as entirely countries (Tokelau) or their change over time, as indicated in grey.

onset, which in turn may limit height gain during adolescence[50,51]. No comparable global data currently exist on age at menarche and timing of pubertal growth, even at the national level. Second, rural-to-urban migration and reclassification of previously rural areas to urban as they grow and industrialize may account for some of the observed population-level trends. However, migration tends to be less common in childhood and adolescence than in adulthood in most countries. Finally, improvements in survival among children aged under 5 years in rural areas, particularly low-birthweight children, may have influenced the height and weight of those who survive beyond 5 years of age. However, current data on changes in child survival in rural and urban areas in sub-Saharan Africa are limited and inconclusive in terms of whether mortality declined faster in rural or urban areas[52,53].

As attention in global health turns to children and adolescents, there is a need to consider and evaluate how growth and development in these formative ages may be affected both by social and economic policies that influence household income and poverty and by programmes that affect nutrition, health services, infrastructure and living environments in rural and urban areas. The need to identify, implement and evaluate policies and programmes that improve growth and development outcomes is particularly relevant as the increase in poverty and the cost of food, especially of nutrient-rich foods, as a result of the macroeconomic changes resulting from the COVID-19 pandemic and the war in Ukraine, may hinder further gains or even set back healthy growth and development in children and adolescents.

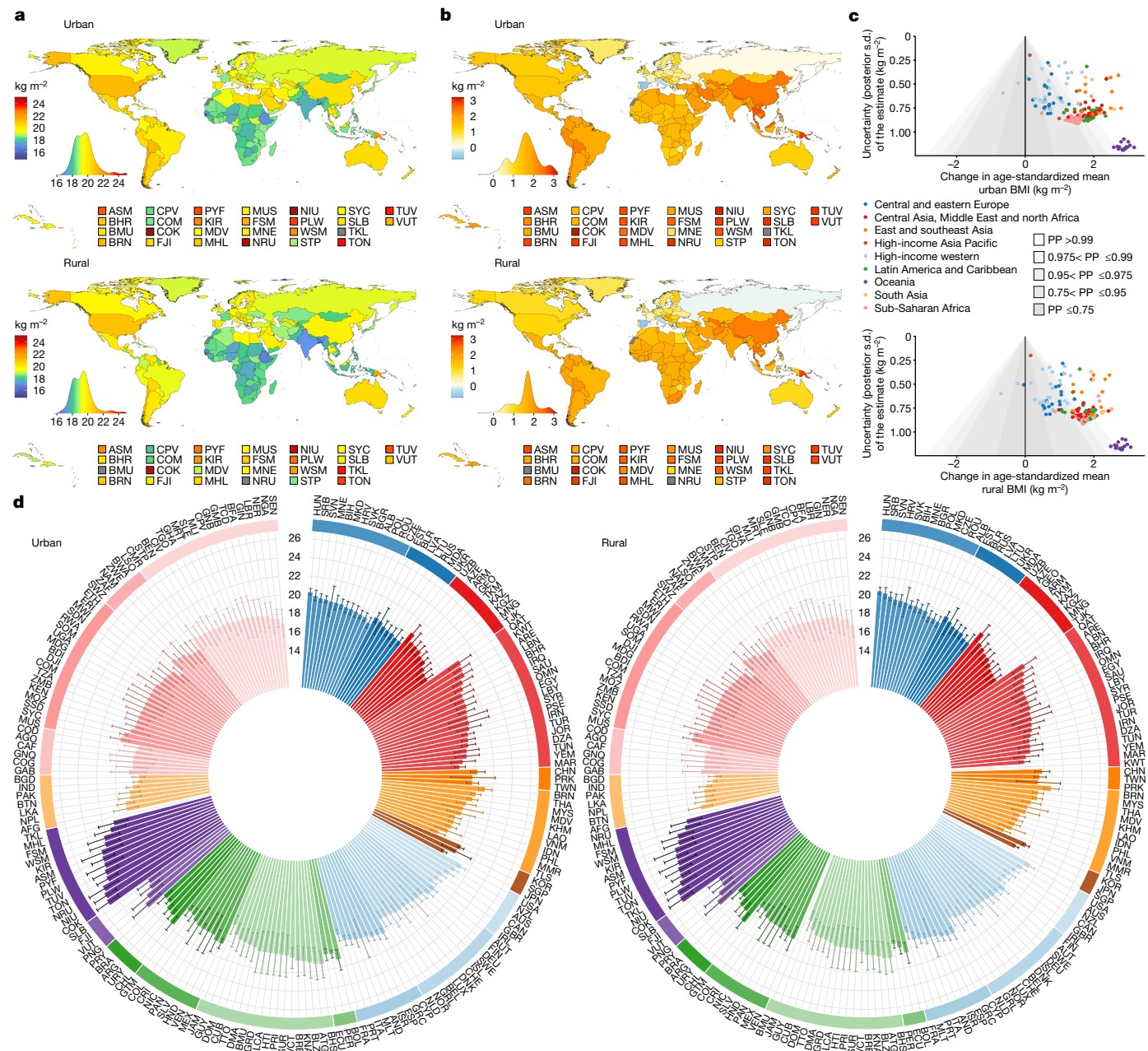

**Fig. 6 | Urban and rural BMI in 2020 and change from 1990 to 2020 for boys. a−d**, See the caption for Fig. 2 for descriptions of the contents of the figure and for definitions. We did not estimate mean rural BMI in countries classified as entirely urban (Singapore, Bermuda and Nauru), mean urban BMI in countries classified as entirely rural (Tokelau) or their change over time, as indicated in grey.

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

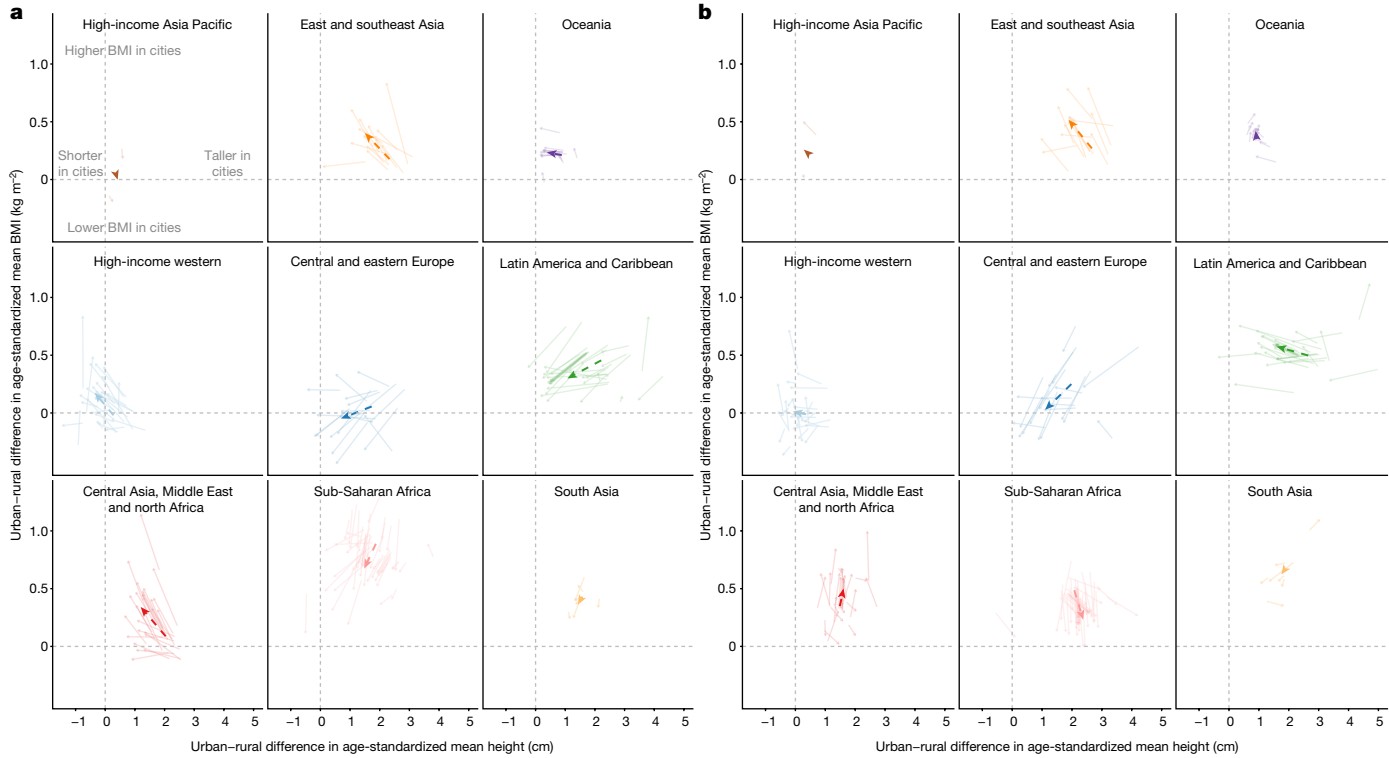

**Fig. 7 | Change in the urban–rural height and BMI difference from 1990 to 2020. a,b,** Change in the urban–rural difference in age-standardized mean height and the urban–rural difference in age-standardized mean BMI in girls (**a**) and boys (**b**). See the caption for Fig. 1 for a description of the contents of

this figure. See Supplementary Fig. 4c for results at ages 5, 10, 15 and 19 years. We did not estimate the difference between rural and urban height and BMI for countries classified as entirely urban (Bermuda, Kuwait, Nauru and Singapore) or entirely rural (Tokelau).

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

NCD Risk Factor Collaboration (NCD-RisC)

Anu Mishra[1,776], Bin Zhou[1,776], Andrea Rodriguez-Martinez[1,776], Honor Bixby[2,3,776], Rosie K. Singleton[1], Rodrigo M. Carrillo-Larco[1], Kate E. Sheffer[1], Christopher J. Paciorek[4], James E. Bennett[1], Victor Lhoste[1], Maria L. C. Iurilli[1], Mariachiara Di Cesare[3], James Bentham[5], Nowell H. Phelps[1], Marisa K. Sophiea[1], Gretchen A. Stevens[6], Goodarz Danaei[7], Melanie J. Cowan[6], Stefan Savin[6], Leanne M. Riley[6], Edward W. Gregg[1], Wichai Aekplakorn[8], Noor Ani Ahmad[9], Jennifer L. Baker[10], Adela Chirita-Emandi[11], Farshad Farzadfar[12], Günther Fink[13,14], Mirjam Heinen[15], Nayu Ikeda[16], Andre P. Kengne[17], Young-Ho Khang[18], Tiina Laatikainen[19,20], Avula Laxmaiah[21], Jun Ma[22], Michele Monroy-Valle[23], Malay K. Mridha[24], Cristina P. Padez[25], Andrew Reynolds[26], Maroje Sorić[27,28], Gregor Starc[28], James P. Wirth[29], Leandra Abarca-Gómez[30], Ziad A. Abdeen[31], Shynar Abdrakhmanova[32], Suhaila Abdul Ghaffar[9], Hanan F. Abdul Rahim[33], Zulfiya Abdurrahmonova[34], Niveen M. Abu-Rmeileh[35], Jamila Abubakar Garba[36], Benjamin Acosta-Cazares[37], Ishag Adam[38], Marzena Adamczyk[39], Robert J. Adams[40], Seth Adu-Afarwuah[41], Kaosar Afsana[24], Shoaib Afzal[42,43], Valirie N. Agbor[44], Imelda A. Agdeppa[45], Javad Aghazadeh-Attari[46], Hassan Aguenaou[47], Carlos A. Aguilar-Salinas[48], Charles Agyemang[49], Mohamad Hasnan Ahmad[9], Ali Ahmadi[50], Naser Ahmadi[12], Nastaran Ahmadi[51], Imran Ahmed[52], Soheir H. Ahmed[53], Wolfgang Ahrens[54], Gulmira Aitmurzaeva[55], Kamel Ajlouni[56], Hazzaa M. Al-Hazzaa[57], Badreya Al-Lahou[58], Rajaa Al-Raddadi[59], Huda M. Al Hourani[60], Nawal M. Al Qaoud[61], Monira Alarouj[62], Fadia AlBuharian[63], Shahla AlDhukair[64], Maryam A. Aldwairji[61], Sylvia Alexius[65], Mohamed M. Ali[6], Abdullah Alkandari[62], Ala'a Alkerwi[66], Buthaina M. Alkhatib[60], Kristine Allin[10], Mar Alvarez-Pedrerol[67], Eman Aly[68], Deepak N. Amarapurkar[69], Pilar Amiano Etxezarreta[70], John Amoah[71], Norbert Amougou[72], Philippe Amouyel[73,74], Lars Bo Andersen[75], Sigmund A. Andersen[76], Odysseas Androutsos[77], Lars Ängquist[42], Ranjit Mohan Anjana[78], Alireza Ansari-Moghaddam[79], Elena Anufrieva[80], Hajer Aounallah-Skhiri[81], Joana Araújo[82], Inger Ariansen[83], Tahir Aris[9], Raphael E. Arku[84], Nimmathota Arlappa[21], Krishna K. Aryal[85], Nega Aseffa[86], Thor Aspelund[87], Felix K. Assah[88], Batyrbek Assembekov[89], Maria Cecília F. Assunção[90], May Soe Aung[91], Juha Auvinen[92,93], Mária Avdičová[94], Shina Avi[95,96], Ana Azevedo[97], Mohsen Azimi-Nezhad[98], Fereidoun Azizi[99], Mehrdad Azmin[12], Bontha V. Babu[100], Maja Bæksgaard Jørgensen[101], Azli Baharudin[9], Suhad Bahijri[59], Marta Bakacs[102], Nagalla Balakrishna[21], Yulia Balanova[103], Mohamed Bamoshmoosh[104], Maciej Banach[105], José R. Banegas[106], Joanna Baran[107], Rafał Baran[107], Carlo M. Barbagallo[108], Valter Barbosa Filho[109], Alberto Barceló[110], Maja Baretić[111], Amina Barkat[112], Joaquin Barnoya[113], Lena Barrera[114], Marta Barreto[115,116], Aluisio J. D. Barros[90], Mauro Virgílio Gomes Barros[117], Anna Bartosiewicz[107], Abdul Basit[118], Joao Luiz D. Bastos[119], Iqbal Bata[120], Anwar M. Batieha[121], Aline P. Batista[122], Rosangela L. Batista[123], Zhamilya Battakova[7], Louise A. Baur[124], Pascal M. Bayauli[125], Robert Beaglehole[126], Silvia Bel-Serrat[127], Antonisamy Belavendra[128], Habiba Ben Romdhane[129], Judith Benedics[130], Mikhail Benet[131], Gilda Estela Benitez Rolandi[132], Elling Bere[133], Ingunn Holden Bergh[83], Yemane Berhane[134], Salim Berkinbayev[135], Antonio Bernabe-Ortiz[136], Gailute Bernotiene[137], Ximena Berrios Carrasola[138], Heloísa Bettiol[139], Manfred E. Beutel[140], Augustin F. Beybey[88], Jorge Bezerra[117], Aroor Bhagyalaxmi[141], Sumit Bharadwaj[142], Santosh K. Bhargava[143], Hongsheng Bi[144], Yufang Bi[145], Daniel Bia[146], Katia Biasch[147], Elysée Claude Bika Lele[148], Mukharram M. Bikbov[149], Bihungum Bista[150], Dusko J. Bjelica[151], Anne A. Bjerregaard[10], Peter Bjerregaard[152], Espen Bjertness[53], Marius B. Bjertness[53], Cecilia Björkelund[153], Katia V. Bloch[154], Anneke Blokstra[155], Moran Blychfeld Magnazu[156,157], Simona Bo[158], Martin Bobak[159], Lynne M. Boddy[160], Bernhard O. Boehm[161], Jolanda M. A. Boer[155], Jose G. Boggia[146], Elena Bogova[162], Carlos P. Boissonnet[163], Stig E. Bojesen[43,42], Marialaura Bonaccio[164], Vanina Bongard[165], Alice Bonilla-Vargas[90], Matthias Bopp[166], Herman Borghs[167], Pascal Bovet[168,169], Khadichamo Boymatova[170], Lien Braeckevelt[171], Lutgart Braeckman[172], Marjolijn C. E. Bragt[173], Imperia Brajkovich[174], Francesco Branca[6], Juergen Breckenkamp[175], João Breda[176], Hermann Brenner[177], Lizzy M. Brewster[49], Garry R. Brian[178], Yajaira Briceño[179], Lacramioara Brinduse[180], Miguel Brito[181], Sinead Brophy[182], Johannes Brug[155], Graziella Bruno[158], Anna Bugge[183], Frank Buntinx[167], Marta Buoncristiano[15], Genc Burazeri[184], Con Burns[185], Antonio Cabrera de León[186], Joseph Cacciottolo[187], Hui Cai[188], Roberta B. Caixeta[189], Tilema Cama[190], Christine Cameron[191], José Camolas[192], Günay Can[193], Ana Paula C. Cândido[194], Felicia Cañete[132], Mario V. Capanzana[45], Naděžda Čapková[195], Eduardo Capuano[196], Rocco Capuano[196], Vincenzo Capuano[196], Marloes Cardol[197], Viviane C. Cardoso[139], Axel C. Carlsson[198], Esteban Carmuega[199], Joana Carvalho[200], José A. Casajús[201], Felipe F. Casanueva[202], Maribel Casas[67], Ertugrul Celikcan[203], Laura Censi[204], Marvin Cervantes-Loaiza[30], Juraci A. Cesar[205], Snehalatha Chamukuttan[206], Angelique Chan[207], Queenie Chan[1], Himanshu K. Chaturvedi[208], Nish Chaturvedi[159], Norsyamlina Che Abdul Rahim[9], Miao Li Chee[209], Chien-Jen Chen[210], Fangfang Chen[211], Huashuai Chen[212], Shuohua Chen[213], Zhengming Chen[44], Ching-Yu Cheng[214], Yiling J. Cheng[215], Bahman Cheraghian[216], Angela Chetrit[217], Ekaterina Chikova-Iscener[218], Mai J. M. Chinapaw[219], Anne Chinnock[220], Arnaud Chiolero[221], Shu-Ti Chiou[222], María-Dolores Chirlaque[223], Belong Cho[224], Kaare Christensen[225], Diego G. Christofaro[226], Jerzy Chudek[227], Renata Cifkova[228,229], Michelle Cilia[230], Eliza Cinteza[180], Massimo Cirillo[231], Frank Claessens[167], Janine Clarke[232], Els Clays[172], Emmanuel Cohen[72], Laura-María Compañ-Gabucio[233], Hans Concin[234], Susana C. Confortin[123], Cyrus Cooper[235], Tara C. Coppinger[185], Eva Corpeleijn[197], Lilia Yadira Cortés[236], Simona Costanzo[164], Dominique Cottel[237], Chris Cowell[124], Cora L. Craig[191], Amelia C. Crampin[238], Amanda J. Cross[1], Ana B. Crujeiras[239], Juan J. Cruz[106], Tamás Csányi[240], Semánová Csilla[241], Alexandra M. Cucu[242,243], Liufu Cui[213], Felipe V. Cureau[244], Sarah Cuschieri[187], Ewelina Czenczek-Lewandowska[107], Graziella D'Arrigo[245], Eleonora d'Orsi[119], Liliana Dacica[246], Jean Dallongeville[237], Albertino Damasceno[247], Camilla T. Damsgaard[42], Rachel Dankner[217], Thomas M. Dantoft[10], Parasmani Dasgupta[248], Saeed Dastgiri[249], Luc Dauchet[73,74], Kairat Davletov[89], Maria Alice Altenburg de Assis[119], Guy De Backer[172], Dirk De Bacquer[172], Amalia De Curtis[164], Patrícia de Fragas Hinnig[15], Giovanni de Gaetano[164], Stefan De Henauw[172], Pilar De Miguel-Etayo[239,201], Paula Duarte de Oliveira[90], David De Ridder[250], Karin De Ridder[251], Susanne R. de Rooij[252,49], Delphine De Smedt[172], Mohan Deepa[78], Alexander D. Deev[253], Vincent DeGennaro Jr[254], Hélène Delisle[255],

Francis Delpeuch[256], Stefaan Demarest[251], Elaine Dennison[235], Katarzyna Dereń[107], Valérie Deschamps[257], Meghnath Dhimal[150], Augusto Di Castelnuovo[258], Juvenal Soares Dias-da-Costa[259], María Elena Díaz-Sánchez[260], Alejandro Diaz[261], Pedro Díaz Fernández[262], María Pilar Díez Ripollés[263], Zivka Dika[27], Shirin Djalalinia[264], Visnja Djordjic[265], Ha T. P. Do[266], Annette J. Dobson[267], Liria Domínguez[268], Maria Benedetta Donati[164], Chiara Donfrancesco[269], Guanghui Dong[270], Yanhui Dong[22], Silvana P. Donoso[271], Angela Döring[272], Maria Dorobantu[180], Ahmad Reza Dorosty[273], Kouamelan Doua[274], Nico Dragano[275], Wojciech Drygas[276,105], Jia Li Duan[277], Charmaine A. Duante[45], Priscilla Duboz[278], Vesselka L. Duleva[218], Virginija Dulskiene[137], Samuel C. Dumith[205], Anar Dushpanova[279,280], Azhar Dyussupova[281], Vilnis Dzerve[282], Elzbieta Dziankowska-Zaborszczyk[105], Guadalupe Echeverría[138], Ricky Eddie[283], Ebrahim Eftekhar[284], Eruke E. Egbagbe[285], Robert Eggertsen[153], Sareh Eghtesad[273], Gabriele Eiben[286], Ulf Ekelund[76], Mohammad El-Khateeb[56], Laila El Ammari[287], Jalila El Ati[288], Denise Eldemire-Shearer[289], Marie Eliasen[10], Paul Elliott[1], Ronit Endevelt[156,290], Reina Engle-Stone[291], Rajiv T. Erasmus[292], Raimund Erbel[293], Cihangir Erem[294], Gul Ergor[295], Louise Eriksen[152], Johan G. Eriksson[296], Jorge Escobedo-de la Peña[37], Saeid Eslami[297], Ali Esmaeili[298], Alun Evans[299], David Faeh[166], Ildar Fakhradiyev[135], Albina A. Fakhretdinova[149], Caroline H. Fall[235], Elnaz Faramarzi[300], Mojtaba Farjam[301], Victoria Farrugia Sant'Angelo[230], Mohammad Reza Fattahi[302], Asher Fawzad[303], Wafaie W. Fawzi[7], Edit Feigl[102], Francisco J. Felix-Redondo[304], Trevor S. Ferguson[289], Romulo A. Fernandes[226], Daniel Fernández-Bergés[305], Daniel Ferrante[306], Thomas Ferrao[232], Gerson Ferrari[307], Marika Ferrari[204], Marco M. Ferrario[308], Catterina Ferreccio[138], Haroldo S. Ferreira[309], Eldridge Ferrer[45], Jean Ferrieres[165], Thamara Hubler Figueiró[119], Anna Fijalkowska[310], Mauro Fisberg[311], Krista Fischer[312], Leng Huat Foo[313], Maria Forsner[314], Heba M. Fouad[68], Damian L. Francis[289], Maria do Carmo Franco[315], Zlatko Fras[316], Guillermo Frontera[317], Flavio D. Fuchs[318], Sandra C. Fuchs[319], Isti I. Fujiati[320], Yuki Fujita[321], Matsuda Fumihiko[322], Viktoriya Furdela[323], Takuro Furusawa[322], Zbigniew Gaciong[324], Mihai Gafencu[11], Manuel Galán Cuesta[325], Andrzej Galbarczyk[326], Henrike Galenkamp[49], Daniela Galeone[327], Myriam Galván[204], Fabio Galvano[213], Jingli Gao[213], Pei Gao[22], Manoli Garcia-de-la-Hera[223], María José García Mérida[262], Marta García Solano[329], Dickman Gareta[124], Sarah P. Garnett[124], Jean-Michel Gaspoz[331], Magda Gasull[223], Adroaldo Cesar Araujo Gaya[319], Anelise Reis Gaya[319], Andrea Gazzinelli[332], Ulrike Gehring[333], Harald Geiger[234], Johanna M. Geleijnse[334], Ronnie George[315], Ebrahim Ghaderi[12], Ali Ghanbari[12], Erfan Ghasemi[12], Oana-Florentina Gheorghe-Fronea[180], Alessandro Gialluisi[308], Simona Giampaoli[269], Francesco Gianfagna[308,258], Christian Gieger[272], Tiffany K. Gill[337], Jonathan Giovannelli[73,74], Glen Gironella[45], Aleksander Giwercman[338], Konstantinos Gkiouras[339], Natalya Glushkova[280,89], Natalja Gluškova[340], Ramesh Godara[341], Justyna Godos[328], Sibel Gogen[203], Marcel Goldberg[342,343], David Goltzman[2], Georgina Gómez[220], Jesús Humberto Gómez Gómez[344], Luis F. Gomez[236], Santiago F. Gómez[345,346], Aleksandra Gomula[347], Bruna Gonçalves Cordeiro da Silva[90], Helen Gonçalves[90], Mauer Gonçalves[348], Ana D. González-Alvarez[349], David A. Gonzalez-Chica[337], Esther M. González-Gil[201], Marcela Gonzalez-Gross[350], Margot González-Leon[37], Juan P. González-Rivas[351], Clicerio González-Villalpando[352], María-Elena González-Villalpando[353], Angel R. Gonzalez[354], Frederic Gottrand[73], Antonio Pedro Graça[355], Sidsel Graff-Iversen[83], Dušan Grafnetter[356], Aneta Grajda[357], Maria G. Grammatikopoulou[358], Ronald D. Gregor[2], Maria João Gregório[355], Else Karin Grøholt[83], Anders Grøntved[225], Giuseppe Grosso[328], Gabriella Gruden[158], Dongfeng Gu[359], Viviana Guajardo[360], Emanuela Gualdi-Russo[361], Pilar Guallar-Castillón[106], Andrea Gualtieri[362], Elias F. Gudmundsson[363], Vilmundur Gudnason[87], Ramiro Guerrero[364], Idris Guessous[250], Andre L. Guimaraes[365], Martin C. Gulliford[366], Johanna Gunnlaugsdottir[363], Marc J. Gunter[367], Xiu-Hua Guo[368], Yin Guo[369], Prakash C. Gupta[370], Rajeev Gupta[371], Oye Gureje[372], Enrique Gutiérrez González[329], Laura Gutierrez[373], Felix Gutzwiller[166], Xinyi Gwee[214], Seongjun Ha[374], Farzad Hadaegh[375], Charalambos A. Hadjigeorgiou[76], Rosa Haghshenas[12], Hamid Hakimi[298], Jytte Halkjær[377], Ian R. Hambleton[378], Behrooz Hamzeh[379], Willem A. Hanekom[380], Dominique Hange[153], Abu A. M. Hanif[24], Sari Hantunen[19], Jie Hao[369], Carla Menêses Hardman[381], Rachakulla Hari Kumar[21], Tina Harmer Lassen[101], Javad Harooni[382], Seyed Mohammad Hashemi-Shahri[79], Maria Hassapidou[383], Jun Hata[384], Teresa Haugsgjerd[385], Alison J. Hayes[124], Jiang He[386], Yuan He[387], Yuna He[388], Regina Heidinger-Felsö[389], Margit Heier[272], Tatjana Hejgaard[390], Marleen Elisabeth Hendriks[391], Rafael dos Santos Henrique[381], Ana Henriques[82], Leticia Hernandez Cadena[352], Sauli Herrala[92], Marianella Herrera-Cuenca[174], Victor W. Herrera[392], Isabelle Herter-Aeberli[393], Karl-Heinz Herzig[93,92], Ramin Heshmat[394], Allan G. Hill[235], Sai Yin Ho[395], Suzanne C. Ho[396], Michael Hobbs[397], Doroteia A. Höfelmann[398], Michelle Holdsworth[256], Reza Homayounfar[399], Clara Homs[400,401], Wilma M. Hopman[402], Andrea R. V. R. Horimoto[139], Claudia M. Hormiga[403], Bernardo L. Horta[90], Leila Houti[404], Christina Howitt[378], Thein Thein Htay[405], Aung Soe Htet[53], Maung Maung Than Htike[406], Yonghua Hu[22], José María Huerta[223], Ilpo Tapani Huhtaniemi[1], Laetitia Huiart[407], Constanta Huidumac Petrescu[242], Martijn Huisman[408], Abdullatif Husseini[35], Chinh Nguyen Huu[266], Inge Huybrechts[367], Nahla Hwalla[409], Jolanda Hyska[184], Licia Iacoviello[164,308], Ellina M. Iakupova[279], Jesús M. Ibarluzea[223], Mohsen M. Ibrahim[410], Norazizah Ibrahim Wong[9], M. Arfan Ikram[411], Carmen Iñiguez[412], Violeta Iotova[413], Vilma E. Irazola[373], Takafumi Ishida[414], Godsent C. Isiguzo[415], Muhammad Islam[416], Sheikh Mohammed Shariful Islam[417], Duygu Islek[418], Ivaila Y. Ivanova-Pandourska[419], Masanori Iwasaki[20], Tuija Jääskeläinen[20], Rod T. Jackson[126], Jeremy M. Jacobs[421], Michel Jadou[422], Tazeen Jafar[207], Bakary Jallow[423], Kenneth James[289], Kazi M. Jamil[424], Konrad Jamrozik[337,777], Anna Jansson[425], Imre Janszky[426], Edward Janus[427], Juel Jarani[428], Marjo-Riitta Jarvelin[1,93], Grazyna Jasienska[326], Ana Jelaković[111], Bojan Jelaković[27], Garry Jennings[429], Chao Qiang Jiang[430], Ramon O. Jimenez[431], Karl-Heinz Jöckel[293], Michel Joffres[432], Jari J. Jokelainen[20], Jost B. Jonas[433], Jitendra Jonnagaddala[434], Torben Jørgensen[10], Pradeep Joshi[435], Josipa Josipović[111], Farahnaz Joukar[436], Jacek J. Jóźwiak[437], Debra S. Judge[397], Anne Juolevi[20], Gregor Jurak[28], Iulia Jurca Simina[11], Vesna Juresa[27], Rudolf Kaaks[177], Felix O. Kaducu[438], Anthony Kafatos[439], Mónika Kaj[440], Eero O. Kajantie[20], Natia Kakutia[441], Daniela Kállayová[280], Zhanna Kalmatayeva[280], Ofra Kalter-Leibovici[217], Yves Kameli[256], Freja B. Kampmann[10], Kodanda R. Kanala[443], Srinivasan Kannan[444], Efthymios Kapantais[445], Eva Karaglani[446], Argyro Karakosta[447], Line L. Kårhus[10], Khem B. Karki[448], Philippe B. Katchunga[449], Marzieh Katibeh[450],

Joanne Katz[451], Peter T. Katzmarzyk[452], Jussi Kauhanen[19], Prabhdeep Kaur[453],
Maryam Kavousi[411], Gyulli M. Kazakbaeva[149], François F. Kaze[88], Calvin Ke[454], Ulrich Keil[455],
Lital Keinan Boker[456], Sirkka Keinänen-Kiukaanniemi[92], Roya Kelishadi[457], Cecily Kelleher[127],
Han C. G. Kemper[219], Maryam Keramati[297], Alina Kerimkulova[458], Mathilde Kersting[459],
Timothy Key[44], Yousef Saleh Khader[121], Arsalan Khaledifar[460], Davood Khalili[399],
Kay-Tee Khaw[461], Bahareh Kheiri[399], Motahareh Kheradmand[462], Alireza Khosravi[463],
Ilse M. S. L. Khouw[173], Ursula Kiechl-Kohlendorfer[464], Sophia J. Kiechl[465], Stefan Kiechl[464,465],
Japhet Killewo[466], Hyeon Chang Kim[467], Jeongseon Kim[468], Jenny M. Kindblom[153,469],
Andrew Kingston[470], Heidi Klakk[471], Magdalena Klimek[326], Jeannette Klimont[472],
Jurate Klumbiene[137], Michael Knoflach[464], Bhawesh Koirala[473], Elin Kolle[76],
Patrick Kolsteren[172], Jürgen König[474], Raija Korpelainen[93], Paul Korrovits[475],
Magdalena Korzycka[310], Jelena Kos[111], Seppo Koskinen[20], Katsuyasu Kouda[476], Éva Kovács[477],
Viktoria Anna Kovacs[440], Irina Kovalskys[478], Sudhir Kowlessur[479], Slawomir Koziel[347],
Jana Kratenova[195], Wolfgang Kratzer[480], Vilma Kriaucioniene[137], Susi Kriemler[166],
Peter Lund Kristensen[225], Helena Krizan[481], Maria F. Kroker-Lobos[482], Steinar Krokstad[426],
Daan Kromhout[197], Herculina S. Kruger[483,484], Ruan Kruger[483,484], Łukasz Kryst[485],
Ruzena Kubinova[195], Renata Kuciene[137], Urho M. Kujala[486], Enisa Kujundzic[487],
Zbigniew Kulaga[357], Mukhtar Kulimbet[280,89], R. Krishna Kumar[488], Marie Kunešová[489],
Pawel Kurjata[276], Yadlapalli S. Kusuma[490], Vladimir Kutsenko[103], Kari Kuulasmaa[20],
Catherine Kyobutungi[491], Quang Ngoc La[492], Fatima Zahra Laamiri[493],
Carl Lachat[172], Karl J. Lackner[140], Youcef Laid[494], Lachmie Lall[495], Tai Hing Lam[395],
Maritza Landaeta Jimenez[174], Edwige Landais[256], Vera Lanska[356], Georg Lappas[496],
Bagher Larijani[497], Simo Pone Larissa[498], Tint Swe Latt[499], Martino Laurenzi[500],
Laura Lauria[269], Maria Lazo-Porras[136], Gwenaëlle Le Coroller[66], Khanh Le Nguyen Bao[266],
Agnès Le Port[256], Tuyen D. Le[256], Jeannette Lee[214,501], Jeonghee Lee[468], Paul H. Lee[502],
Nils Lehmann[293], Terho Lehtimäki[503,504], Daniel Lemogoum[505], Branimir Leskošek[28],
Justyna Leszczak[107], Katja B. Leth-Møller[10], Gabriel M. Leung[395], Naomi S. Levitt[506],
Yanping Li[7], Merike Liivak[340], Christa L. Lilly[507], Charlie Lim[214,501], Wei-Yen Lim[214,501],
M. Fernanda Lima-Costa[508], Hsien-Ho Lin[221], Xu Lin[509], Yi-Ting Lin[510], Lars Lind[510],
Vijaya Lingam[335], Birgit Linkohr[272], Allan Linneberg[10], Lauren Lissner[153],
Mieczyslaw Litwin[357], Jing Liu[511], Lijuan Liu[369], Wei-Cheng Lo[512], Helle-Mai Loit[340],
Khuong Quynh Long[492], Guadalupe Longo Abril[513], Luis Lopes[200], Marcus V. V. Lopes[119],
Oscar Lopes[514], Esther Lopez-Garcia[106], Tania Lopez[515], Paulo A. Lotufo[139],
José Eugenio Lozano[516], Janice L. Lukrafka[517], Dalia Luksiene[137], Annamari Lundqvist[20],
Nuno Lunet[200], Charles Lunogelo[518], Michala Lustigová[228,195], Edyta Łuszczki[107],
Jean-René M'Buyamba-Kabangu[519], Guansheng Ma[22], Xu Ma[387],
George L. L. Machado-Coelho[122], Aristides M. Machado-Rodrigues[25], Enguerran Macia[278],
Luisa M. Macieira[520], Ahmed A. Madar[53], Anja L. Madsen[10], Gladys E. Maestre[521],
Stefania Maggi[522], Dianna J. Magliano[523], Sara Magnacca[258], Emmanuella Magriplis[524],
Gowri Mahasampath[128], Bernard Maire[256], Marjeta Majer[27], Marcia Makdisse[525],
Päivi Mäki[20], Fatemeh Malekzadeh[273], Reza Malekzadeh[302,273], Rahul Malhotra[207],
Kodavanti Mallikharjuna Rao[21], Sofia K. Malyutina[526], Lynell V. Maniego[45], Yannis Manios[446],
Masimango Imani Manix[527], Jim I. Mann[26], Fariborz Mansour-Ghanaei[436], Taru Manyanga[528],
Enzo Manzato[529], Anie Marcil[232], Paula Margozzini[138], Joany Mariño[530], Anastasia Markaki[531],
Oonagh Markey[532], Eliza Markidou Ioannidou[533], Pedro Marques-Vidal[534,535],
Larissa Pruner Marques[536], Jaume Marrugat[537,538], Yves Martin-Prevel[256],
Rosemarie Martin[539], Reynaldo Martorell[418], Eva Martos[540], Katharina Maruszczak[541],
Stefano Marventano[328], Giovanna Masala[542], Luis P. Mascarenhas[543], Shariq R. Masoodi[544],
Ellisiv B. Mathiesen[545], Prashant Mathur[546], Alicia Matijasevich[139], Piotr Matłosz[107],
Tandi E. Matsha[547], Victor Matsudo[548], Christina Mavrogianni[446], Artur Mazur[107],
Jean Claude N. Mbanya[88], Shelly R. McFarlane[289], Stephen T. McGarvey[549], Martin McKee[550],
Stela McLachlan[551], Rachael M. McLean[26], Scott B. McLean[232], Margaret L. McNairy[552],
Breige A. McNulty[127], Sounnia Mediene Benchekor[404], Jurate Medzioniene[137],
Parinaz Mehdipour[1], Kirsten Mehlig[153], Amir Houshang Mehrparvar[51], Aline Meirhaeghe[553],
Jørgen Meisfjord[83], Christa Meisinger[272], Jesus D. Melgarejo[167], Marina Melkumova[554],
João Mello[319], Fabián Méndez[114], Carlos O. Mendivil[555], Ana Maria B. Menezes[90],
Geetha R. Menon[208], Gert B. M. Mensink[556], Maria Teresa Menzano[327], Indrapal I. Meshram[21],
Diane T. Meto[557], Jie Mi[211], Kim F. Michaelsen[42], Nathalie Michels[172], Kairit Mikkel[312],
Karolina Mitkowska[556], Jody C. Miller[26], Olga Milushkina[558], Cláudia S. Minderico[559],
G. K. Mini[560], Juan Francisco Miquel[138], J. Jaime Miranda[136], Mohammad Reza Mirjalili[51],
Daphne Mirkopoulou[561], Erkin Mirrakhimov[458], Marjeta Mišigoj-Durakovic[27],
Antonio Mistretta[328], Veronica Mocanu[562], Pietro A. Modesti[563], Sahar Saeedi Moghaddam[12],
Bahram Mohajer[12], Mostafa K. Mohamed[564], Shukri F. Mohamed[491], Kazem Mohammad[273],
Mohammad Reza Mohammadi[273], Zahra Mohammadi[273], Noushin Mohammadifard[565],
Reza Mohammadpourhodki[297], Viswanathan Mohan[78], Salim Mohanna[136],
Muhammad Fadhli Mohd Yusoff[9], Iraj Mohebbi[46], Farnam Mohebi[4], Marie Moitry[147,566],
Line T. Møllehave[10], Niels C. Møller[225], Dénes Molnár[389], Amirabbas Momenan[399],
Charles K. Mondo[567], Roger A. Montenegro Mendoza[568], Eric Monterrubio-Flores[352],
Kotsedi Daniel K. Monyeki[569], Jin Soo Moon[224], Mahmood Moosazadeh[462],
Hermine T. Mopa[88], Farhad Moradpour[336], Leila B. Moreira[319], Alain Morejon[570],
Luis A. Moreno[201,239], Francis Morey[571], Karen Morgan[572], Suzanne N. Morin[2],
Erik Lykke Mortensen[42], George Moschonis[573], Alireza Moslem[574], Malgorzata Mossakowska[575],
Aya Mostafa[564], Seyed-Ali Mostafavi[273], Anabela Mota-Pinto[25], Jorge Mota[200],
Mohammad Esmaeel Motlagh[216], Jorge Motta[568], Marcos André Moura-dos-Santos[117],
Yeva Movsesyan[554], Kelias P. Msyamboza[576], Thet Thet Mu[577], Magdalena Muc[25],
Florian Muca[578], Boban Mugoša[487], Maria L. Muiesan[579], Martina Müller-Nurasyid[140],
Thomas Münzel[140], Jaakko Mursu[19], Elaine M. Murtagh[580], Kamarul Imran Musa[313],
Sanja Musić Milanović[481,27], Vera Musil[27], Geofrey Musinguzi[581], Muel Telo M. C. Muyer[582],
Iraj Nabipour[583], Shohreh Naderimagham[12], Gabriele Nagel[584], Farid Najafi[379],
Harunobu Nakamura[476], Hanna Nalecz[310], Jana Námešná[94], Ei Ei K. Nang[214,501],
Vinay B. Nangia[585], Martin Nankap[586], Sameer Narake[370], Paola Nardone[269], Take Naseri[587],
Matthias Nauck[530], William A. Neal[507], Azim Nejatizadeh[284], Chandini Nekkantti[434],
Keiu Nelis[340], Ilona Nenko[326], Martin Neovius[588], Flavio Nervi[138], Tze Pin Ng[214],
Chung T. Nguyen[589], Nguyen D. Nguyen[589], Quang Ngoc Nguyen[591], Michael Y. Ni[395],
Rodica Nicolescu[242], Peng Nie[592], Ramfis E. Nieto-Martínez[593], Yury M. Nikitin[526],
Guang Ning[145], Toshiharu Ninomiya[384], Nobuo Nishi[16], Sania Nishtar[594], Marianna Noale[522],
Oscar A. Noboa[146], Helena Nogueira[25], Maria Nordendahl[314], Børge G. Nordestgaard[43,42],

Davide Noto[108], Natalia Nowak-Szczepanska[347], Mohannad Al Nsour[595], Irfan Nuhoğlu[294],
Baltazar Nunes[115,116], Eha Nurk[340], Fred Nuwaha[581], Moffat Nyirenda[550], Terence W. O'Neill[596],
Dermot O'Reilly[299], Galina Obreja[597], Caleb Ochimana[7], Angélica M. Ochoa-Avilés[271],
Eiji Oda[598], Augustine N. Odili[599], Kyungwon Oh[600], Kumiko Ohara[476], Claes Ohlsson[153,469],
Ryutaro Ohtsuka[601], Örn Olafsson[363], Maria Teresa A. Olinto[319], Isabel O. Oliveira[90],
Mohd Azahadi Omar[9], Saeed M. Omar[602], Altan Onat[603,777], Sok King Ong[604],
N. Charlotte Onland-Moret[333], Lariane M. Ono[398], Pedro Ordunez[189], Rui Ornelas[605],
Ana P. Ortiz[606], Pedro J. Ortiz[136], Merete Osler[10], Clive Osmond[235], Sergej M. Ostojic[265],
Afshin Ostovar[607], Johanna A. Otero[608], Kim Overvad[450], Ellis Owusu-Dabo[609],
Fred Michel Paccaud[169], Ioannis Pagkalos[383], Elena Pahomova[282], Karina Mary de Paiva[119],
Andrzej Pająk[326], Alberto Palloni[610], Luigi Palmieri[269], Wen-Harn Pan[210],
Songhomitra Panda-Jonas[611], Arvind Pandey[264], Francesco Panza[612], Mariela Paoli[179],
Sousana K. Papadopoulou[383], Dimitrios Papandreou[613], Rossina G. Pareja[268],
Soon-Woo Park[614], Suyeon Park[600], Winsome R. Parnell[26], Mahboubeh Parsaeian[273],
Ionela M. Pascanu[615], Patrick Pasquet[72], Nikhil D. Patel[616], Marcos Pattussi[259],
Halyna Pavlyshyn[323], Raimund Pechlaner[464], Ivan Pećin[111], Mangesh S. Pednekar[370],
João M. Pedro[617], Nasheeta Peer[618], Sergio Viana Peixoto[508], Markku Peltonen[20],
Alexandre C. Pereira[139], Marco A. Peres[619], Cynthia M. Pérez[606], Valentina Peterkova[162],
Annette Peters[272], Astrid Petersmann[530], Janina Petkeviciene[137], Ausra Petrauskiene[137],
Olga Petrovna Kovtun[80], Emanuela Pettenuzzo[620], Niloofar Peykari[264], Norbert Pfeiffer[140],
Modou Cheyassin Phall[221], Son Thai Pham[621], Rafael N. Pichardo[622], Daniela Pierannunzio[269],
Iris Pigeot[54], Hynek Pikhart[159], Aida Pilav[623], Lorenza Pilotto[624], Francesco Pistelli[625],
Freda Pitakaka[626], Aleksandra Piwonska[276], Andreia N. Pizarro[200], Pedro Plans-Rubió[627],
Alina G. Platonova[628], Bee Koon Poh[629], Hermann Pohlabeln[54], Nadija S. Polka[628],
Raluca M. Pop[615], Stevo R. Popovic[151], Miquel Porta[538], Georg Posch[234], Anil Poudyal[150],
Dimitrios Poulimeneas[383], Hamed Pouraram[273], Farhad Pourfarzi[630], Akram Pourshams[273],
Hossein Poustchi[273], Rajendra Pradeepa[78], Alison J. Price[550], Jacqueline F. Price[551],
Antonio Prista[631], Rui Providencia[159], Jardena J. Puder[534], Iveta Pudule[632], Maria Puiu[11],
Margus Punab[475], Muhammed S. Qadir[533], Radwan F. Qasrawi[31], Mostafa Qorbani[534],
Hedley K. Quintana[568], Pedro J. Quiroga-Padilla[555], Tran Quoc Bao[635], Stefan Rach[54],
Ivana Radic[265], Ricardas Radisauskas[137], Salar Rahimikazerooni[302], Mahfuzar Rahman[636],
Mahmudur Rahman[637], Olli Raitakari[638], Manu Raj[488], Tamerlan Rajabov[639],
Sherali Rakhmatulloev[34], Ivo Rakovac[15], Sudha Ramachandra Rao[453],
Ambady Ramachandran[206], Otim P. C. Ramadan[640], Virgílio V. Ramires[641],
Jacqueline Ramke[126], Elisabete Ramos[97], Rafel Ramos[642], Lekhraj Rampal[643],
Sanjay Rampal[644], Lalka S. Rangelova[218], Vayia Rarra[645], Ramon A. Rascon-Pacheco[37],
Cassiano Ricardo Rech[119], Josep Redon[412], Paul Ferdinand M. Reganit[646],
Valéria Regecová[647], Jane D. P. Renner[648], Judit A. Repasy[389], Cézane P. Reuter[648],
Luis Revilla[515], Abbas Rezaianzadeh[302], Yeunsook Rho[374], Lourdes Ribas-Barba[649],
Robespierre Ribeiro[650,777], Elio Riboli[1], Adrian Richter[530], Fernando Rigo[651], Attilio Rigotti[138],
Natascia Rinaldo[361], Tobias F. Rinke de Wit[652], Ana I. Rito[115], Raphael M. Ritti-Dias[653],
Juan A. Rivera[352], Reina G. Roa[654], Louise Robinson[470], Cynthia Robitaille[555],
Romana Roccaldo[204], Daniela Rodrigues[25], Fernando Rodríguez-Artalejo[106], María del Cristo
Rodriguez-Perez[262], Laura A. Rodríguez-Villamizar[656], Andrea Y. Rodríguez[657],
Ulla Roggenbuck[293], Peter Rohloff[658], Fabian Rohner[659], Rosalba Rojas-Martinez[352],
Nipa Rojroongwasinkul[6], Dora Romaguera[239], Elisabetta L. Romeo[660], Rafaela V. Rosario[661],
Annika Rosengren[153,469], Ian Rouse[662], Vanessa Rouzier[663], Joel G. R. Roy[232],
Maira H. Ruano[664], Adolfo Rubinstein[373], Frank J. Rühli[166], Jean-Bernard Ruidavets[165],
Blanca Sandra Ruiz-Betancourt[37], Maria Ruiz-Castell[66], Emma Ruiz Moreno[665],
Iuliia A. Rusakova[149], Kenisha Russell Jonsson[425], Paola Russo[666], Petra Rust[474],
Marcin Rutkowski[667], Marge Saamel[340], Charumathi Sabanayagam[209], Hamideh Sabbaghi[399],
Elena Sacchini[362], Harshpal S. Sachdev[668], Alireza Sadjadi[273], Ali Reza Safarpour[302],
Sare Safi[399], Saeid Safiri[300], Mohammad Hossien Saghi[574], Olfa Saidi[129], Nader Saki[216],
Sanja Šalaj[27], Benoit Salanave[257], Eduardo Salazar Martinez[352], Calogero Saleva[542],
Diego Salmerón[23], Veikko Salomaa[20], Jukka T. Salonen[296], Massimo Salvetti[579],
Margarita Samoutini[669], Jose Sánchez-Abanto[670], Inés Sánchez Rodríguez[344], Sandjaja[671],
Susana Sans[672], Loreto Santa Marina[673], Ethel Santacruz[132], Diana A. Santos[559],
Ina S. Santos[90], Lèlita C. Santos[520], Maria Paula Santos[200], Osvaldo Santos[674], Rute Santos[200],
Tamara R. Santos[675], Jouko L. Saramies[676], Luis B. Sardinha[559], Nizal Sarrafzadegan[565],
Thirunavukkarasu Sathish[418], Kai-Uwe Saum[177], Savvas Savva[376], Mathilde Savy[256],
Norie Sawada[677], Mariana Sbaraini[319], Marcia Scazufca[678], Beatriz D. Schaan[319],
Angelika Schaffrath Rosario[556], Herman Schargrodsky[679], Anja Schienkiewitz[556],
Karin Schindler[680], Sabine Schipf[530], Carsten O. Schmidt[530], Ida Maria Schmidt[681],
Andrea Schneider[272], Peter Schnohr[43], Ben Schöttker[177], Sara Schramm[293],
Stine Schramm[225], Helmut Schröder[223], Constance Schultsz[682], Matthias B. Schulze[683],
Aletta E. Schutte[434,684], Sylvain Sebert[93], Moslem Sedaghattalab[382], Rusidah Selamat[9],
Vedrana Sember[28], Abhijit Sen[685], Idowu O. Senbanjo[686], Sadaf G. Sepanlou[273],
Guillermo Sequera[132], Luis Serra-Majem[687], Jennifer Servais[232], Ľudmila Ševčíková[688],
Svetlana Shalnova[103], Teresa Shamah-Levy[352], Seyed Morteza Shamshirgaran[98],
Coimbatore Subramaniam Shanthirani[78], Maryam Sharafkhah[273], Sanjib K. Sharma[473],
Jonathan E. Shaw[523], Amaneh Shayanrad[273], Ali Akbar Shayesteh[216], Lela Shengelia[441],
Zumin Shi[34], Kenji Shibuya[366], Hana Shimizu-Furusawa[689], Tal Shimony[456], Rahman Shiri[690],
Namuna Shrestha[85], Khairil Si-Ramlee[604], Alfonso Siani[666], Rosalynn Siantar[209],
Abla M. Sibai[409], Labros S. Sidossis[691], Natalia Silitrari[692], Antonio M. Silva[123],
Caroline Ramos de Moura Silva[117], Diego Augusto Santos Silva[119], Kelly S. Silva[119],
Xueling Sim[214,501], Mary Simon[206], Judith Simons[434], Leon A. Simons[434], Agneta Sjöberg[153],
Michael Sjöström[588,777], Natalia A. Skoblina[558], Gry Skodje[694], Tatyana Slazhnyova[32],
Jolanta Slowikowska-Hilczer[105], Przemysław Slusarczyk[575], Liam Smeeth[550],
Hung-Kwan So[395], Fernanda Cunha Soares[117], Grzegorz Sobek[107], Eugène Sobngwi[88],
Morten Sodemann[225], Stefan Söderberg[314], Moesijanti Y. E. Soekatri[695],
Agustinus Soemantri[696,777], Reecha Sofat[159], Vincenzo Solfrizzi[697],
Mohammad Hossein Somi[300], Emily Sonestedt[338], Yi Song[22], Sajid Soofi[52],
Thorkild I. A. Sørensen[42], Elin P. Sørgjerd[426], Charles Sossa Jérome[698],
Victoria E. Soto-Rojas[364], Aïcha Soumaré[699], Alfonso Sousa-Poza[700], Slavica Sovic[27],
Bente Sparboe-Nilsen[701], Karen Sparrenberger[319], Phoebe R. Spencer[397], Angela Spinelli[269],
Igor Spiroski[702,703], Jan A. Staessen[167], Hanspeter Stamm[704], Kaspar Staub[166], Bill Stavreski[429],
Jostein Steene-Johannessen[76], Peter Stehle[705], Aryeh D. Stein[418], George S. Stergiou[447],

Jochanan Stessman[421], Ranko Stevanović[481], Jutta Stieber[272,777], Doris Stöckl[272], Jakub Stokwiszewski[706], Ekaterina Stoyanova[707], Gareth Stratton[182], Karien Stronks[49], Maria Wany Strufaldi[315], Lela Sturua[441], Ramón Suárez-Medina[260], Machi Suka[708], Chien-An Sun[709], Liang Sun[509], Johan Sundström[510], Yn-Tz Sung[396], Jordi Sunyer[67], Paibul Suriyawongpaisal[8], Nabil William G. Sweis[710], Boyd A. Swinburn[126], Rody G. Sy[646], René Charles Sylva[711], Moyses Szklo[451], Lucjan Szponar[706], Lorraine Tabone[230], E. Shyong Tai[214,501], Konstantinos D. Tambalis[447], Mari-Liis Tammeso[312], Abdonas Tamosiunas[137], Eng Joo Tan[712], Xun Tang[22], Maya Tanrygulyyeva[713], Frank Tanser[714], Yong Tao[22], Mohammed Rasoul Tarawneh[715], Jakob Tarp[450], Carolina B. Tarqui-Mamani[670], Radka Taxová Braunerová[489], Anne Taylor[337], Julie Taylor[159], Félicité Tchibindat[716], Saskia Te Velde[717], William R. Tebar[226], Grethe Tell[385], Tania Tello[136], Yih Chung Tham[209], K. R. Thankappan[341], Holger Theobald[198], Nihal Thomas[128], Barbara Thorand[272], Betina H. Thuesen[10], Ľubica Tichá[688], Erik J. Timmermans[718], Dwi H. Tjandrarini[719], Anne Tjonneland[377], Hanna K. Tolonen[20], Janne S. Tolstrup[152], Murat Topbas[294], Roman Topór-Mądry[326], Liv Elin Torheim[701], María José Tormo[720], Michael J. Tornaritis[376], Maties Torrent[721], Laura Torres-Collado[223], Stefania Toselli[722], Giota Touloumi[447], Pierre Traissac[256], Thi Tuyet-Hanh Tran[492], Mark S. Tremblay[723], Areti Triantafyllou[339], Dimitrios Trichopoulos[7,777], Antonia Trichopoulou[724], Oanh T. H. Trinh[590], Atul Trivedi[725], Yu-Hsiang Tsao[726], Lechaba Tshepo[727], Maria Tsigga[383], Panagiotis Tsintavis[383], Shoichiro Tsugane[677], John Tuitele[728,729], Azalia M. Tuliakova[149], Marshall K. Tulloch-Reid[289], Fikru Tullu[730], Tomi-Pekka Tuomilehto[20], Maria L. Turley[731], Gilad Twig[217,732], Per Tynelius[588], Evangelia Tzala[1], Themistoklis Tzotzas[445], Christophe Tzourio[699], Peter Ueda[588], Eunice Ugel[733], Flora A. M. Ukoli[734], Hanno Ulmer[464], Belgin Unal[295], Zhamyila Usupova[55], Hannu M. T. Uusitalo[735], Nalan Uysal[736], Justina Vaitkeviciute[137], Gonzalo Valdivia[138], Susana Valle[744], Damaskini Valvi[738], Rob M. van Dam[739], Bert-Jan van den Born[49], Johan Van der Heyden[251], Yvonne T. van der Schouw[333], Koen Van Herck[172], Wendy Van Lippevelde[172], Hoang Van Minh[492], Natasja M. Van Schoor[408], Irene G. M. van Valkengoed[49], Dirk Vanderschueren[172], Diego Vanuzzo[624], Anette Varbo[43,42], Gregorio Varela-Moreiras[740], Luz Nayibe Vargas[236], Patricia Varona-Pérez[260], Senthil K. Vasan[235], Daniel G. Vasques[319], Tomas Vega[516], Toomas Veidebaum[340], Gustavo Velasquez-Melendez[332], Biruta Velika[632], Maïté Verloigne[172], Giovanni Veronesi[308], W. M. Monique Verschuren[155], Cesar G. Victora[90], Giovanni Viegi[741], Leyre Viet[155], Frøydis N. Vik[133], Monica Vilar[742], Salvador Villalpando[352], Jesus Vioque[743], Jyrki K. Virtanen[19], Sophie Visvikis-Siest[744], Bharathi Viswanathan[168], Mihaela Vladulescu[745], Tiina Vlasoff[746], Dorja Vocanec[27], Peter Vollenweider[534,535], Henry Völzke[530], Ari Voutilainen[19], Martine Vrijheid[67], Tanja G. M. Vrijkotte[252,49], Alisha N. Wade[747], Thomas Waldhör[680], Janette Walton[185], Elvis O. A. Wambiya[491], Wan Mohamad Wan Bebakar[313], Wan Nazaimoon Wan Mohamud[748], Rildo de Souza Wanderley Júnior[117], Ming-Dong Wang[655], Ningli Wang[369], Qian Wang[749], Xiangjun Wang[750], Ya Xing Wang[368], Ying-Wei Wang[751], S. Goya Wannamethee[159], Nicholas Wareham[461], Adelheid Weber[130], Karen Webster-Kerr[752], Niels Wedderkopp[225], Daniel Weghuber[541], Wenbin Wei[368], Aneta Weres[107], Bo Werner[753], Leo D. Westbury[235], Peter H. Whincup[754], Kremlin Wickramasinghe[15], Kurt Widhalm[680], Indah S. Widyahening[755], Andrzej Więcek[227], Philipp S. Wild[140], Rainford J. Wilks[289], Johann Willeit[464], Peter Willeit[464], Julianne Williams[15], Tom Wilsgaard[545], Rusek Wojciech[756], Bogdan Wojtyniak[706], Kathrin Wolf[272], Roy A. Wong-McClure[30], Andrew Wong[159], Emily B. Wong[380], Jyh Eiin Wong[629], Tien Yin Wong[207], Jean Woo[396], Mark Woodward[1,434], Frederick C. Wu[596], Hon-Yen Wu[757], Jianfeng Wu[144], Li Juan Wu[368], Shouling Wu[213], Justyna Wyszyńska[107], Haiquan Xu[758], Liang Xu[759], Nor Azwany Yaacob[313], Uruwan Yamborisut[8], Weili Yan[760], Ling Yang[44], Xiaoguang Yang[388], Yang Yang[750], Nazan Yardim[203], Tabara Yasuharu[322], Martha Yépez García[742], Panayiotis K. Yiallouros[761], Agneta Yngve[510], Moein Yoosefi[12], Akihiro Yoshihara[762], Qi Sheng You[368], San-Lin You[709], Novie O. Younger-Coleman[289], Yu-Ling Yu[167], Yunjiang Yu[763], Safiah Md Yusof[764], Ahmad Faudzi Yusoff[9], Luciana Zaccagni[361], Vassilis Zafiropulos[765], Ahmad A. Zainuddin[9], Seyed Rasoul Zakavi[297], Farhad Zamani[766], Sabina Zambon[529], Antonis Zampelas[524], Hana Zamrazilová[489], Maria Elisa Zapata[199], Abdul Hamid Zargar[767], Ko Ko Zaw[499], Ayman A. Zayed[710], Tomasz Zdrojewski[667], Magdalena Żegleń[768], Kristyna Zejglicova[195], Tajana Zeljkovic Vrkic[111], Yi Zeng[22,769], Luxia Zhang[770], Zhen-Yu Zhang[167], Dong Zhao[511], Ming-Hui Zhao[770], Wenhua Zhao[388], Yanitsa V. Zhecheva[419], Shiqi Zhen[771], Yingfeng Zheng[207], Bekbolat Zholdin[772], Maigeng Zhou[388], Dan Zhu[773], Marie Zins[342,343], Emanuel Zitt[234], Yanina Zocalo[146], Nada Zoghlami[81], Julio Zuñiga Cisneros[568], Monika Zuziak[774], Zulfiqar A. Bhutta[416,52], Robert E. Black[775] & Majid Ezzati[1,41] ✉

[1]Imperial College London, London, UK. [2]McGill University, Montreal, Québec, Canada. [3]University of Essex, Colchester, UK. [4]University of California Berkeley, Berkeley, CA, USA. [5]University of Kent, Canterbury, UK. [6]World Health Organization, Geneva, Switzerland. [7]Harvard T. H. Chan School of Public Health, Boston, MA, USA. [8]Mahidol University, Nakhon Pathom, Thailand. [9]Ministry of Health, Kuala Lumpur, Malaysia. [10]Bispebjerg and Frederiksberg Hospital, Copenhagen, Denmark. [11]Victor Babes University of Medicine and Pharmacy, Timisoara, Romania. [12]Non-Communicable Diseases Research Center, Tehran, Iran. [13]Swiss Tropical and Public Health Institute, Basel, Switzerland. [14]University of Basel, Basel, Switzerland. [15]World Health Organization Regional Office for Europe, Copenhagen, Denmark. [16]National Institutes of Biomedical Innovation, Health and Nutrition, Tokyo, Japan. [17]South African Medical Research Council, Cape Town, South Africa. [18]Seoul National University College of Medicine, Seoul, Republic of Korea. [19]University of Eastern Finland, Kuopio, Finland. [20]Finnish Institute for Health and Welfare, Helsinki, Finland. [21]ICMR–National Institute of Nutrition, Hyderabad, India. [22]Peking University, Beijing, China. [23]Universidad de San Carlos, Guatemala City, Guatemala. [24]BRAC James P. Grant School of Public Health, Dhaka, Bangladesh. [25]University of Coimbra, Coimbra, Portugal. [26]University of Otago, Dunedin, New Zealand. [27]University of Zagreb, Zagreb, Croatia. [28]University of Ljubljana, Ljubljana, Slovenia. [29]GroundWork, Geneva, Switzerland. [30]Caja Costarricense de Seguro Social, San José, Costa Rica. [31]Al-Quds University, East Jerusalem, State of Palestine. [32]National Center of Public Health, Astana, Kazakhstan. [33]Qatar University, Doha, Qatar. [34]Ministry of Health and Social Protection, Dushanbe, Tajikistan. [35]Birzeit University, Birzeit, State of Palestine. [36]Usmanu Danfodiyo University Teaching Hospital, Sokoto, Nigeria. [37]Instituto Mexicano del Seguro Social, Mexico City, Mexico. [38]Qassim University, Unaizah, Saudi Arabia. [39]RehaKlinika, Rzeszów, Poland. [40]Flinders University, Adelaide, South Australia, Australia. [41]University of Ghana, Accra, Ghana. [42]University of Copenhagen, Copenhagen, Denmark. [43]Copenhagen University Hospital, Copenhagen, Denmark. [44]University of Oxford, Oxford, UK. [45]Food and Nutrition Research Institute, Taguig, The Philippines. [46]Urmia University of Medical Sciences, Urmia, Iran. [47]Ibn Tofail University, Kénitra, Morocco. [48]Instituto Nacional de Ciencias Médicas y Nutrición, Mexico City, Mexico. [49]University of Amsterdam, Amsterdam, The Netherlands. [50]Modeling in Health Research Center, Shahrekord, Iran. [51]Shahid Sadoughi University of Medical Sciences, Yazd, Iran. [52]The Aga Khan University, Karachi, Pakistan. [53]University of Oslo, Oslo, Norway. [54]Leibniz Institute for Prevention Research and Epidemiology–BIPS, Bremen, Germany. [55]Republican Center for Health Promotion, Bishkek, Kyrgyzstan. [56]National Center for Diabetes, Endocrinology and Genetics, Amman, Jordan. [57]Princess Nourah bint Abdulrahman University, Riyadh, Saudi Arabia. [58]Kuwait Institute for Scientific Research, Kuwait City, Kuwait. [59]King Abdulaziz University, Jeddah, Saudi Arabia. [60]The Hashemite University, Zarqa, Jordan. [61]Ministry of Health, Kuwait City, Kuwait. [62]Dasman Diabetes Institute, Kuwait City, Kuwait. [63]Aldara Hospital and Medical Center, Riyadh, Saudi Arabia. [64]King Abdullah International Medical Research Center, Riyadh, Saudi Arabia. [65]Universidade Federal da Integração Latino-Americana, Foz do Iguaçu, Brazil. [66]Luxembourg Institute of Health, Strassen, Luxembourg. [67]Barcelona Institute for Global Health CIBERESP, Barcelona, Spain. [68]World Health Organization Regional Office for the Eastern Mediterranean, Cairo, Egypt. [69]Bombay Hospital and Medical Research Centre, Mumbai, India. [70]Departamento de Salud del Gobierno Vasco, San Sebastián, Spain. [71]Ghana Health Service, Kintampo, Ghana. [72]UMR CNRS-MNHN 7206, Paris, France. [73]University of Lille, Lille, France. [74]Lille University Hospital, Lille, France. [75]Western Norway University of Applied Sciences, Sogndal, Norway. [76]Norwegian School of Sport Sciences, Oslo, Norway. [77]University of Thessaly, Trikala, Greece. [78]Madras Diabetes Research Foundation, Chennai, India. [79]Zahedan University of Medical Sciences, Zahedan, Iran. [80]Yekaterinburg State Medical Academy, Yekaterinburg, Russia. [81]National Institute of Public Health, Tunis, Tunisia. [82]Institute of Public Health of the University of Porto, Porto, Portugal. [83]Norwegian Institute of Public Health, Oslo, Norway. [84]University of Massachusetts Amherst, Amherst, MA, USA. [85]Public Health Promotion and Development Organization, Kathmandu, Nepal. [86]Haramaya University, Dire Dawa, Ethiopia. [87]University of Iceland, Reykjavik, Iceland. [88]University of Yaoundé 1, Yaoundé, Cameroon. [89]Asfendiyarov Kazakh National Medical University, Almaty, Kazakhstan. [90]Federal University of Pelotas, Pelotas, Brazil. [91]University of Medicine 1, Yangon, Myanmar. [92]Oulu University Hospital, Oulu, Finland. [93]University of Oulu, Oulu, Finland. [94]Regional Authority of Public Health, Banska Bystrica, Slovakia. [95]Tel Aviv University, Tel Aviv, Israel. [96]Hebrew University of Jerusalem, Jerusalem, Israel. [97]University of Porto Medical School, Porto, Portugal. [98]Neyshabur University of Medical Sciences, Neyshabur, Iran. [99]Research Institute for Endocrine Sciences, Tehran, Iran. [100]Indian Council of Medical Research, New Delhi, India. [101]National Institute of Public Health, Copenhagen, Denmark. [102]National Institute of Pharmacy and Nutrition, Budapest, Hungary. [103]National Medical Research Centre for Therapy and Preventive Medicine, Moscow, Russia. [104]University of Science and Technology, Sana'a, Yemen. [105]Medical University of Lodz, Lodz, Poland. [106]Universidad Autónoma de Madrid CIBERESP, Madrid, Spain. [107]University of Rzeszów, Rzeszów, Poland. [108]University of Palermo, Palermo, Italy. [109]Federal Institute of Education, Science and Technology of Ceara, Ceara, Brazil. [110]University of Miami, Miami, FL, USA. [111]University Hospital Center Zagreb, Zagreb, Croatia. [112]Mohammed V University, Rabat, Morocco. [113]Unidad de Cirugia Cardiovascular, Guatemala City, Guatemala. [114]Universidad del Valle, Cali, Colombia. [115]National Institute of Health Doutor Ricardo Jorge, Lisbon, Portugal. [116]NOVA University Lisbon, Lisbon, Portugal. [117]University of Pernambuco, Recife, Brazil. [118]Baqai Institute of Diabetology and Endocrinology, Karachi, Pakistan. [119]Federal University of Santa Catarina, Florianópolis, Brazil. [120]Dalhousie University, Halifax, Nova Scotia, Canada. [121]Jordan University of Science and Technology, Irbid, Jordan. [122]Universidade Federal de Ouro Preto, Ouro Preto, Brazil. [123]Federal University of Maranhão, São Luís, Brazil. [124]University of Sydney, Sydney, New South Wales, Australia. [125]Cliniques Universitaires de Kinshasa, Kinshasa, Democratic Republic of the Congo. [126]University of Auckland, Auckland, New Zealand. [127]University College Dublin, Dublin, Ireland. [128]Christian Medical College, Vellore, India. [129]University Tunis El Manar, Tunis, Tunisia. [130]Federal Ministry of Social Affairs, Health, Care and Consumer Protection, Vienna, Austria. [131]Cafam University Foundation, Bogotá, Colombia. [132]Ministerio de Salud Pública y Bienestar Social, Asunción, Paraguay. [133]University of Agder, Kristiansand, Norway. [134]Addis Continental Institute of Public Health, Addis Ababa, Ethiopia. [135]Kazakh National Medical University, Almaty, Kazakhstan. [136]Universidad Peruana Cayetano Heredia, Lima, Peru. [137]Lithuanian University of Health Sciences, Kaunas, Lithuania. [138]Pontificia Universidad Católica de Chile, Santiago, Chile. [139]University of São Paulo, São Paulo, Brazil. [140]Johannes Gutenberg University, Mainz, Germany. [141]B. J. Medical College, Ahmedabad, India. [142]Chirayu Medical College, New Delhi, India. [143]Sunder Lal Jain Hospital, Delhi, India. [144]Shandong University of Traditional Chinese Medicine, Jinan, China. [145]Shanghai Jiao-Tong University School of Medicine, Shanghai, China. [146]Universidad de la República, Montevideo, Uruguay. [147]University of Strasbourg, Strasbourg, France. [148]Institute of Medical Research and Medicinal Plant Studies, Yaoundé, Cameroon. [149]Ufa Eye Research Institute, Ufa, Russia. [150]Nepal Health Research Council, Kathmandu, Nepal. [151]University of Montenegro, Niksic, Montenegro. [152]University of Southern Denmark, Copenhagen, Denmark. [153]University of Gothenburg, Gothenburg, Sweden. [154]Universidade Federal do Rio de Janeiro, Rio de Janeiro, Brazil. [155]National Institute for Public Health and the Environment, Bilthoven, The Netherlands. [156]University of Haifa, Haifa, Israel. [157]Ministry of Health, Ramat Gan, Israel. [158]University of Turin, Turin, Italy. [159]University College London, London, UK. [160]Liverpool John Moores University, Liverpool, UK. [161]Nanyang Technological University, Singapore, Singapore. [162]National Medical Research Center for Endocrinology, Moscow, Russia. [163]Centro de Educación Médica e Investigaciones Clínicas, Buenos Aires, Argentina. [164]IRCCS Neuromed, Pozzilli, Italy. [165]Toulouse University School of Medicine, Toulouse, France. [166]University of Zurich, Zurich, Switzerland. [167]KU Leuven, Leuven, Belgium. [168]Ministry of Health, Victoria, Seychelles. [169]Unisanté, Lausanne, Switzerland. [170]World Health Organization Country Office in Tajikistan, Dushanbe, Tajikistan. [171]Flemish Agency for Care and Health, Brussels, Belgium. [172]Ghent University, Ghent, Belgium. [173]FrieslandCampina, Amersfoort, The Netherlands. [174]Universidad Central de Venezuela, Caracas, Venezuela. [175]Bielefeld University, Bielefeld, Germany. [176]World Health Organization Athens Quality of Care Office, Athens, Greece. [177]German Cancer Research Center, Heidelberg, Germany. [178]The Fred Hollows Foundation, Auckland, New Zealand. [179]University of the Andes, Mérida, Venezuela. [180]Carol Davila University of Medicine and Pharmacy, Bucharest, Romania.

[181]Instituto Politécnico de Lisboa, Lisbon, Portugal. [182]Swansea University, Swansea, UK. [183]University College Copenhagen, Copenhagen, Denmark. [184]Institute of Public Health, Tirana, Albania. [185]Munster Technological University, Cork, Ireland. [186]Universidad de La Laguna, Santa Cruz de Tenerife, Spain. [187]University of Malta, Msida, Malta. [188]Vanderbilt University, Nashville, TN, USA. [189]Pan American Health Organization, Washington, DC, USA. [190]Ministry of Health, Tongatapu, Tonga. [191]Canadian Fitness and Lifestyle Research Institute, Ottawa, Ontario, Canada. [192]Hospital Santa Maria, Lisbon, Portugal. [193]Istanbul University–Cerrahpasa, Istanbul, Turkey. [194]Universidade Federal de Juiz de Fora, Juiz de Fora, Brazil. [195]National Institute of Public Health, Prague, Czech Republic. [196]Gaetano Fucito Hospital, Mercato San Severino, Italy. [197]University of Groningen, Groningen, The Netherlands. [198]Karolinska Institutet, Huddinge, Sweden. [199]Centro de Estudios Sobre Nutrición Infantil, Buenos Aires, Argentina. [200]University of Porto, Porto, Portugal. [201]University of Zaragoza, Zaragoza, Spain. [202]Santiago de Compostela University, Santiago de Compostela, Spain. [203]Ministry of Health, Ankara, Turkey. [204]Council for Agricultural Research and Economics, Rome, Italy. [205]Federal University of Rio Grande, Rio Grande, Brazil. [206]India Diabetes Research Foundation, Chennai, India. [207]Duke–NUS Medical School, Singapore, Singapore. [208]ICMR–National Institute of Medical Statistics, New Delhi, India. [209]Singapore Eye Research Institute, Singapore, Singapore. [210]Academia Sinica, Taipei, Taiwan. [211]Capital Institute of Pediatrics, Beijing, China. [212]Xiangtan University, Xiangtan, China. [213]Kailuan General Hospital, Tangshan, China. [214]National University of Singapore, Singapore, Singapore. [215]US Centers for Disease Control and Prevention, Atlanta, GA, USA. [216]Ahvaz Jundishapur University of Medical Sciences, Ahvaz, Iran. [217]The Gertner Institute for Epidemiology and Health Policy Research, Ramat Gan, Israel. [218]National Centre of Public Health and Analyses, Sofia, Bulgaria. [219]Amsterdam UMC Public Health Research Institute, Amsterdam, The Netherlands. [220]Universidad de Costa Rica, San José, Costa Rica. [221]University of Fribourg, Fribourg, Switzerland. [222]National Taiwan University, Taipei, Taiwan. [223]CIBERESP, Madrid, Spain. [224]Seoul National University, Seoul, Republic of Korea. [225]University of Southern Denmark, Odense, Denmark. [226]Universidade Estadual Paulista, Presidente Prudente, Brazil. [227]Medical University of Silesia, Katowice, Poland. [228]Charles University, Prague, Czech Republic. [229]Thomayer Hospital, Prague, Czech Republic. [230]Primary Health Care, Floriana, Malta. [231]University of Salerno, Fisciano, Italy. [232]Statistics Canada, Ottawa, Ontario, Canada. [233]Alicante Institute for Health and Biomedical Research, Alicante, Spain. [234]Agency for Preventive and Social Medicine, Bregenz, Austria. [235]University of Southampton, Southampton, UK. [236]Pontificia Universidad Javeriana, Bogotá, Colombia. [237]Institut Pasteur de Lille, Lille, France. [238]Malawi Epidemiology and Intervention Research Unit, Lilongwe, Malawi. [239]CIBEROBN, Madrid, Spain. [240]Hungarian University of Sports Science, Budapest, Hungary. [241]University of Debrecen, Debrecen, Hungary. [242]National Institute of Public Health, Bucharest, Romania. [243]University of Medicine and Pharmacy, Bucharest, Romania. [244]Universidade Federal do Rio Grande do Norte, Natal, Brazil. [245]National Research Council, Reggio Calabria, Italy. [246]Eftimie Murgu University Resita, Resita, Romania. [247]Eduardo Mondlane University, Maputo, Mozambique. [248]Indian Statistical Institute, Kolkata, India. [249]Tabriz Health Services Management Research Center, Tabriz, Iran. [250]Geneva University Hospitals, Geneva, Switzerland. [251]Sciensano, Brussels, Belgium. [252]University Medical Centers, Amsterdam, The Netherlands. [253]National Research Centre for Preventive Medicine, Moscow, Russia. [254]Innovating Health International, Port-au-Prince, Haiti. [255]University of Montreal, Montreal, Quebec, Canada. [256]French National Research Institute for Sustainable Development, Montpellier, France. [257]French Public Health Agency, St Maurice, France. [258]Mediterranea Cardiocentro, Naples, Italy. [259]Universidade do Vale do Rio dos Sinos, São Leopoldo, Brazil. [260]National Institute of Hygiene, Epidemiology and Microbiology, Havana, Cuba. [261]National Council of Scientific and Technical Research, Buenos Aires, Argentina. [262]Servicio Canario de la Salud del Gobierno de Canarias, Santa Cruz de Tenerife, Spain. [263]Consejería de Salud del Gobierno de La Rioja, Logroño, Spain. [264]Ministry of Health and Medical Education, Tehran, Iran. [265]University of Novi Sad, Novi Sad, Serbia. [266]National Institute of Nutrition, Hanoi, Vietnam. [267]University of Queensland, Brisbane, Queensland, Australia. [268]Instituto de Investigación Nutricional, Lima, Peru. [269]Istituto Superiore di Sanità, Rome, Italy. [270]Sun Yat-sen University, Guangzhou, China. [271]Universidad de Cuenca, Cuenca, Ecuador. [272]Helmholtz Zentrum München, Munich, Germany. [273]Tehran University of Medical Sciences, Tehran, Iran. [274]Ministère de la Santé et de l'Hygiène Publique, Abidjan, Côte d'Ivoire. [275]University Hospital Düsseldorf, Düsseldorf, Germany. [276]National Institute of Cardiology, Warsaw, Poland. [277]Beijing Center for Disease Prevention and Control, Beijing, China. [278]IRL 3189 ESS, Marseille, France. [279]Scuola Superiore Sant'Anna, Pisa, Italy. [280]Al-Farabi Kazakh National University, Almaty, Kazakhstan. [281]Semey Medical University, Semey, Kazakhstan. [282]University of Latvia, Riga, Latvia. [283]Ministry of Health and Medical Services, Gizo, Solomon Islands. [284]Hormozgan University of Medical Sciences, Bandar Abbas, Iran. [285]University of Benin, Benin City, Nigeria. [286]University of Skövde, Skövde, Sweden. [287]Ministry of Health, Rabat, Morocco. [288]National Institute of Nutrition and Food Technology, Tunis, Tunisia. [289]The University of the West Indies, Kingston, Jamaica. [290]Ministry of Health, Jerusalem, Israel. [291]University of California Davis, Davis, CA, USA. [292]University of Stellenbosch, Cape Town, South Africa. [293]University of Duisburg-Essen, Essen, Germany. [294]Karadeniz Technical University, Trabzon, Turkey. [295]Dokuz Eylul University, Izmir, Turkey. [296]University of Helsinki, Helsinki, Finland. [297]Mashhad University of Medical Sciences, Mashhad, Iran. [298]Rafsanjan University of Medical Sciences, Rafsanjan, Iran. [299]Queen's University Belfast, Belfast, UK. [300]Tabriz University of Medical Sciences, Tabriz, Iran. [301]Fasa University of Medical Sciences, Fasa, Iran. [302]Shiraz University of Medical Sciences, Shiraz, Iran. [303]Baqai Medical University, Karachi, Pakistan. [304]Centro de Salud Villanueva Norte, Badajoz, Spain. [305]Hospital Don Benito-Villanueva de la Serena, Badajoz, Spain. [306]Ministry of Health, Buenos Aires, Argentina. [307]Universidad de Santiago de Chile, Santiago, Chile. [308]University of Insubria, Varese, Italy. [309]Federal University of Alagoas, Alagoas, Brazil. [310]Institute of Mother and Child, Warsaw, Poland. [311]Hospital Infantil Sabará, São Paulo, Brazil. [312]University of Tartu, Tartu, Estonia. [313]Universiti Sains Malaysia, Kelantan, Malaysia. [314]Umeå University, Umeå, Sweden. [315]Federal University of São Paulo, São Paulo, Brazil. [316]University Clinical Centre Ljubljana, Ljubljana, Slovenia. [317]Hospital Universitario Son Espases, Palma, Spain. [318]Hospital de Clinicas de Porto Alegre, Porto Alegre, Brazil. [319]Universidade Federal do Rio Grande do Sul, Porto Alegre, Brazil. [320]Universitas Sumatera Utara, Medan, Indonesia. [321]Kindai University, Osaka-Sayama, Japan. [322]Kyoto University, Kyoto, Japan. [323]I. Horbachevsky Ternopil National Medical University, Ternopil, Ukraine. [324]Medical University of Warsaw, Warsaw, Poland. [325]Consejería de Sanidad del Gobierno de Cantabria, Santander, Spain. [326]Jagiellonian University Medical College, Kraków, Poland. [327]Ministero della Salute DG Prevenzione Sanitaria, Rome, Italy. [328]University of Catania, Catania, Italy. [329]Agencia Española de Seguridad Alimentaria y Nutrición, Madrid, Spain. [330]Africa Health Research Institute, Mtubatuba, South Africa. [331]Geneva University Medical School, Geneva, Switzerland. [332]Universidade Federal de Minas Gerais, Belo Horizonte, Brazil. [333]Utrecht University, Utrecht, The Netherlands. [334]Wageningen University, Wageningen, The Netherlands. [335]Medical Research Foundation, Chennai, India. [336]Kurdistan University of Medical Sciences, Sanandaj, Iran. [337]University of Adelaide, Adelaide, South Australia, Australia. [338]Lund University, Lund, Sweden. [339]Aristotle University of Thessaloniki, Thessaloniki, Greece. [340]National Institute for Health Development, Tallinn, Estonia. [341]Central University of Kerala, Kasaragod, India. [342]Institut National de la Santé et de la Recherche Médicale, Paris, France. [343]Paris University, Paris, France. [344]Instituto Murciano de Investigación Biosanitaria Virgen de la Arrixaca, Murcia, Spain. [345]Gasol Foundation, Sant Boi de Llobregat, Spain. [346]University of Lleida, Sant Boi de Llobregat, Spain. [347]PASs Hirszfeld Institute of Immunology and Experimental Therapy, Wroclaw, Poland. [348]University Agostinho Neto, Luanda, Angola. [349]Kansas State University, Manhattan, KS, USA. [350]Universidad Politécnica de Madrid, Madrid, Spain. [351]International Clinical Research Center, Brno, Czech Republic. [352]National Institute of Public Health, Cuernavaca, Mexico. [353]Centro de Estudios en Diabetes A.C., Mexico City, Mexico. [354]Universidad Autónoma de Santo Domingo, Santo Domingo, Dominican Republic. [355]Ministry of Health, Lisbon, Portugal. [356]Institute for Clinical and Experimental Medicine, Prague, Czech Republic. [357]Children's Memorial Health Institute, Warsaw, Poland. [358]University of Thessaly, Larissa, Greece. [359]National Center of Cardiovascular Diseases, Beijing, China. [360]International Life Science Institute, Buenos Aires, Argentina. [361]University of Ferrara, Ferrara, Italy. [362]Authority Sanitaria San Marino, San Marino, San Marino. [363]Icelandic Heart Association, Kopavogur, Iceland. [364]Universidad Icesi, Cali, Colombia. [365]State University of Montes Claros, Montes Claros, Brazil. [366]King's College London, London, UK. [367]International Agency for Research on Cancer, Lyon, France. [368]Capital Medical University, Beijing, China. [369]Capital Medical University Beijing Tongren Hospital, Beijing, China. [370]Healis-Sekhsaria Institute for Public Health, Navi Mumbai, India. [371]Eternal Heart Care Centre and Research Institute, Jaipur, India. [372]University of Ibadan, Ibadan, Nigeria. [373]Institute for Clinical Effectiveness and Health Policy, Buenos Aires, Argentina. [374]National Health Insurance Service, Wonju, Republic of Korea. [375]Prevention of Metabolic Disorders Research Center, Tehran, Iran. [376]Research and Education Institute of Child Health, Nicosia, Cyprus. [377]Danish Cancer Society Research Center, Copenhagen, Denmark. [378]The University of the West Indies, Cave Hill, Barbados. [379]Kermanshah University of Medical Sciences, Kermanshah, Iran. [380]Africa Health Research Institute, Durban, South Africa. [381]Federal University of Pernambuco, Recife, Brazil. [382]Yasuj University of Medical Sciences, Yasuj, Iran. [383]International Hellenic University, Thessaloniki, Greece. [384]Kyushu University, Fukuoka, Japan. [385]University of Bergen, Bergen, Norway. [386]Tulane University, New Orleans, LA, USA. [387]National Research Institute for Health and Family Planning, Beijing, China. [388]Chinese Center for Disease Control and Prevention, Beijing, China. [389]University of Pécs, Pécs, Hungary. [390]Danish Health Authority, Copenhagen, Denmark. [391]Joep Lange Institute, Amsterdam, The Netherlands. [392]Universidad Autónoma de Bucaramanga, Bucaramanga, Colombia. [393]ETH Zurich, Zurich, Switzerland. [394]Chronic Diseases Research Center, Tehran, Iran. [395]University of Hong Kong, Hong Kong, China. [396]The Chinese University of Hong Kong, Hong Kong, China. [397]University of Western Australia, Perth, Western Australia, Australia. [398]Universidade Federal do Paraná, Curitiba, Brazil. [399]Shahid Beheshti University of Medical Sciences, Tehran, Iran. [400]Gasol Foundation, Barcelona, Spain. [401]University Ramon Llull, Sant Boi de Llobregat, Spain. [402]Kingston Health Sciences Centre, Kingston, Ontario, Canada. [403]Fundación Oftalmológica de Santander, Bucaramanga, Colombia. [404]University Oran 1, Oran, Algeria. [405]Independent Public Health Specialist, Nay Pyi Taw, Myanmar. [406]Ministry of Health and Sports, Nay Pyi Taw, Myanmar. [407]Santé publique France, Saint-Maurice, France. [408]VU University Medical Center, Amsterdam, The Netherlands. [409]American University of Beirut, Beirut, Lebanon. [410]Cairo University, Cairo, Egypt. [411]Erasmus Medical Center Rotterdam, Rotterdam, The Netherlands. [412]University of Valencia, Valencia, Spain. [413]Medical University Varna, Varna, Bulgaria. [414]The University of Tokyo, Tokyo, Japan. [415]Alex Ekwueme Federal University Teaching Hospital, Abakaliki, Nigeria. [416]The Hospital for Sick Children, Toronto, Ontario, Canada. [417]Deakin University, Geelong, Victoria, Australia. [418]Emory University, Atlanta, GA, USA. [419]Bulgarian Academy of Sciences, Sofia, Bulgaria. [420]Tokyo Metropolitan Institute of Gerontology, Tokyo, Japan. [421]Hadassah University Medical Center, Jerusalem, Israel. [422]Université Catholique de Louvain, Brussels, Belgium. [423]Gambia National Nutrition Agency, Banjul, The Gambia. [424]Kuwait Institute for Scientific Research, Safat, Kuwait. [425]Public Health Agency of Sweden, Solna, Sweden. [426]Norwegian University of Science and Technology, Trondheim, Norway. [427]University of Melbourne, Melbourne, Victoria, Australia. [428]Sports University of Tirana, Tirana, Albania. [429]Heart Foundation, Melbourne, Victoria, Australia. [430]Guangzhou 12th Hospital, Guangzhou, China. [431]Universidad Eugenio Maria de Hostos, Santo Domingo, Dominican Republic. [432]Simon Fraser University, Burnaby, British Columbia, Canada. [433]Institute of Molecular and Clinical Ophthalmology Basel, Basel, Switzerland. [434]University of New South Wales, Sydney, New South Wales, Australia. [435]World Health Organization Country Office, Delhi, India. [436]Guilan University of Medical Sciences, Rasht, Iran. [437]University of Opole, Opole, Poland. [438]Gulu University, Gulu, Uganda. [439]University of Crete, Heraklion, Greece. [440]Hungarian School Sport Federation, Budapest, Hungary. [441]National Center for Disease Control and Public Health, Tbilisi, Georgia. [442]Ministry of Health, Bratislava, Slovakia. [443]Sri Venkateswara University, Tirupati, India. [444]Sree Chitra Tirunal Institute for Medical Sciences and Technology, Trivandrum, India. [445]Hellenic Medical Association for Obesity, Athens, Greece. [446]Harokopio University, Athens, Greece. [447]National and Kapodistrian University of Athens, Athens, Greece. [448]Maharajgunj Medical Campus, Kathmandu, Nepal. [449]Université Officielle de Bukavu, Bukavu, Democratic Republic of the Congo. [450]Aarhus University, Aarhus, Denmark. [451]Johns Hopkins Bloomberg School of Public Health, Baltimore, MD, USA. [452]Pennington Biomedical Research Center, Baton Rouge, LA, USA. [453]National Institute of Epidemiology, Chennai, India. [454]University of Toronto, Toronto, Ontario, Canada. [455]University of Münster, Münster, Germany. [456]Israel Center for Disease Control, Ramat Gan, Israel. [457]Research Institute for Primordial Prevention of Non-communicable Disease, Isfahan, Iran. [458]Kyrgyz State Medical Academy, Bishkek, Kyrgyzstan. [459]Research Institute of Child Nutrition, Dortmund, Germany. [460]Shahrekord University of Medical Sciences, Shahrekord, Iran. [461]University of Cambridge, Cambridge, UK. [462]Mazandaran University of Medical Sciences, Sari, Iran. [463]Hypertension Research Center, Isfahan, Iran. [464]Medical University of Innsbruck, Innsbruck, Austria. [465]VASCage, Innsbruck, Austria. [466]Muhimbili University of

Health and Allied Sciences, Dar es Salaam, Tanzania. [467]Yonsei University College of Medicine, Seoul, Republic of Korea. [468]National Cancer Center, Goyang-si, Republic of Korea. [469]Sahlgrenska University Hospital, Gothenburg, Sweden. [470]Newcastle University, Newcastle, UK. [471]University College South Denmark, Haderslev, Denmark. [472]Statistics Austria, Vienna, Austria. [473]B. P. Koirala Institute of Health Sciences, Dharan, Nepal. [474]University of Vienna, Vienna, Austria. [475]Tartu University Clinics, Tartu, Estonia. [476]Kansai Medical University, Hirakata, Japan. [477]District Department of State Public Health Service, Hildburghausen, Germany. [478]Pontificia Universidad Católica Argentina, Buenos Aires, Argentina. [479]Ministry of Health and Wellness, Port Louis, Mauritius. [480]University Hospital Ulm, Ulm, Germany. [481]Croatian Institute of Public Health, Zagreb, Croatia. [482]Institute of Nutrition of Central America and Panama, Guatemala City, Guatemala. [483]North-West University, Potchefstroom, South Africa. [484]South African Medical Research Council, Potchefstroom, South Africa. [485]University of Physical Education, Kraków, Poland. [486]University of Jyväskylä, Jyväskylä, Finland. [487]Institute of Public Health, Podgorica, Montenegro. [488]Amrita Institute of Medical Sciences, Cochin, India. [489]Institute of Endocrinology, Prague, Czech Republic. [490]All India Institute of Medical Sciences, New Delhi, India. [491]African Population and Health Research Center, Nairobi, Kenya. [492]Hanoi University of Public Health, Hanoi, Vietnam. [493]Hassan First University of Settat, Settat, Morocco. [494]Ministry of Health, Algiers, Algeria. [495]Ministry of Health, Georgetown, Guyana. [496]Sahlgrenska Academy, Gothenburg, Sweden. [497]Endocrinology and Metabolism Research Center, Tehran, Iran. [498]Clinical Research Education, Networking & Consultancy, Douala, Cameroon. [499]University of Public Health, Yangon, Myanmar. [500]Centro Studi Epidemiologici di Gubbio, Gubbio, Italy. [501]National University Health System, Singapore, Singapore. [502]University of Leicester, Leicester, UK. [503]Tampere University Hospital, Tampere, Finland. [504]Tampere University, Tampere, Finland. [505]University of Douala, Douala, Cameroon. [506]University of Cape Town, Cape Town, South Africa. [507]West Virginia University, Morgantown, WV, USA. [508]Oswaldo Cruz Foundation Rene Rachou Research Institute, Belo Horizonte, Brazil. [509]Shanghai Institute of Nutrition and Health of Chinese Academy of Sciences, Shanghai, China. [510]Uppsala University, Uppsala, Sweden. [511]Capital Medical University Beijing An Zhen Hospital, Beijing, China. [512]Taipei Medical University, Taipei, Taiwan. [513]Servicio Andaluz de Salud, Sevilla, Spain. [514]Sports Medical Center of Minho, Braga, Portugal. [515]Universidad San Martín de Porres, Lima, Peru. [516]Consejería de Sanidad Junta de Castilla y León, Valladolid, Spain. [517]Universidade Federal de Ciências da Saúde de Porto Alegre, Porto Alegre, Brazil. [518]Ilembula Lutheran Hospital, Ilembula, Tanzania. [519]University of Kinshasa Hospital, Kinshasa, Democratic Republic of the Congo. [520]Coimbra University Hospital Center, Coimbra, Portugal. [521]University of Texas Rio Grande Valley, Harlingen, TX, USA. [522]Institute of Neuroscience of the National Research Council, Padua, Italy. [523]Baker Heart and Diabetes Institute, Melbourne, Victoria, Australia. [524]Agricultural University of Athens, Athens, Greece. [525]Academia VBHC, São Paulo, Brazil. [526]SB RAS Federal Research Center Institute of Cytology and Genetics, Novosibirsk, Russia. [527]Université Catholique de Bukavu, Bukavu, Democratic Republic of the Congo. [528]University of Northern British Columbia, Prince George, British Columbia, Canada. [529]University of Padua, Padua, Italy. [530]University Medicine Greifswald, Greifswald, Germany. [531]Hellenic Mediterranean University, Siteia, Greece. [532]Loughborough University, Loughborough, UK. [533]Ministry of Health, Nicosia, Cyprus. [534]Lausanne University Hospital, Lausanne, Switzerland. [535]University of Lausanne, Lausanne, Switzerland. [536]Secretaria de Estado da Saúde de Santa Catarina, Florianópolis, Brazil. [537]CIBERCV, Barcelona, Spain. [538]Institut Hospital del Mar d'Investigacions Mèdiques, Barcelona, Spain. [539]Mary Immaculate College, Limerick, Ireland. [540]Hungarian Society of Sports Medicine, Budapest, Hungary. [541]Paracelsus Medical University, Salzburg, Austria. [542]Institute for Cancer Research, Prevention and Clinical Network, Florence, Italy. [543]Universidade Estadual do Centro-Oeste, Guarapuava, Brazil. [544]Sher-i-Kashmir Institute of Medical Sciences, Srinagar, India. [545]UiT The Arctic University of Norway, Tromsø, Norway. [546]ICMR–National Centre for Disease Informatics and Research, Bengaluru, India. [547]Sefako Makgatho Health Sciences University, Pretoria, South Africa. [548]Centro de Estudos do Laboratório de Aptidão Física de São Caetano do Sul, São Paulo, Brazil. [549]Brown University, Providence, RI, USA. [550]London School of Hygiene & Tropical Medicine, London, UK. [551]University of Edinburgh, Edinburgh, UK. [552]Weill Cornell Medicine, New York City, NY, USA. [553]Institut National de la Santé et de la Recherche Médicale, Lille, France. [554]Arabkir Medical Centre–Institute of Child and Adolescent Health, Yerevan, Armenia. [555]Universidad de los Andes, Bogotá, Colombia. [556]Robert Koch Institute, Berlin, Germany. [557]University of Abidjan, Abidjan, Côte d'Ivoire. [558]Pirogov Russian National Research Medical University, Moscow, Russia. [559]Universidade de Lisboa, Lisbon, Portugal. [560]Saveetha Dental Colleges & Hospitals, Chennai, India. [561]Democritus University, Alexandroupolis, Greece. [562]Grigore T Popa University of Medicine and Pharmacy, Iasi, Romania. [563]Università degli Studi di Firenze, Florence, Italy. [564]Ain Shams University, Cairo, Egypt. [565]Isfahan Cardiovascular Research Center, Isfahan, Iran. [566]Strasbourg University Hospital, Strasbourg, France. [567]Mulago Hospital, Kampala, Uganda. [568]Instituto Conmemorativo Gorgas de Estudios de la Salud, Panama City, Panama. [569]University of Limpopo, Sovenga, South Africa. [570]University of Medical Sciences of Cienfuegos, Cienfuegos, Cuba. [571]Ministry of Health and Wellness, Belmopan, Belize. [572]Royal College of Surgeons in Ireland, Dublin, Ireland. [573]La Trobe University, Melbourne, Victoria, Australia. [574]Sabzevar University of Medical Sciences, Sabzevar, Iran. [575]International Institute of Molecular and Cell Biology, Warsaw, Poland. [576]World Health Organization Country Office, Lilongwe, Malawi. [577]Department of Public Health, Nay Pyi Taw, Myanmar. [578]Albanian Sports Science Association, Tirana, Albania. [579]University of Brescia, Brescia, Italy. [580]University of Limerick, Limerick, Ireland. [581]Makerere University School of Public Health, Kampala, Uganda. [582]University de Kinshasa, Kinshasa, Democratic Republic of the Congo. [583]Bushehr University of Medical Sciences, Bushehr, Iran. [584]Ulm University, Ulm, Germany. [585]Suraj Eye Institute, Nagpur, India. [586]UNICEF, Yaoundé, Cameroon. [587]Ministry of Health, Apia, Samoa. [588]Karolinska Institutet, Stockholm, Sweden. [589]National Institute of Hygiene and Epidemiology, Hanoi, Vietnam. [590]University of Medicine and Pharmacy, Ho Chi Minh City, Vietnam. [591]Hanoi Medical University, Hanoi, Vietnam. [592]Xi'an Jiaotong University, Xi'an, China. [593]LifeDoc Health, Memphis, TN, USA. [594]Heartfile, Islamabad, Pakistan. [595]Eastern Mediterranean Public Health Network, Amman, Jordan. [596]University of Manchester, Manchester, UK. [597]State University of Medicine and Pharmacy, Chisinau, Moldova. [598]Tachikawa General Hospital, Nagaoka, Japan. [599]University of Abuja College of Health Sciences, Abuja, Nigeria. [600]Korea Centers for Disease Control and Prevention, Cheongju-si, Republic of Korea. [601]Japan Wildlife Research Center, Tokyo, Japan. [602]Gadarif University, Gadarif, Sudan. [603]Istanbul University, Istanbul, Turkey. [604]Ministry of Health, Bandar Seri Begawan, Brunei. [605]University of Madeira, Funchal, Portugal. [606]University of Puerto Rico, San Juan, Puerto Rico. [607]Osteoporosis Research Center, Tehran, Iran. [608]Universidad de Santander, Bucaramanga, Colombia. [609]Kwame Nkrumah University of Science and Technology, Kumasi, Ghana. [610]University of Wisconsin-Madison, Madison, WI, USA. [611]Privatpraxis Prof Jonas und Dr Panda-Jonas, Heidelberg, Germany. [612]IRCCS Ente Ospedaliero Specializzato in Gastroenterologia S. de Bellis, Bari, Italy. [613]Zayed University, Abu Dhabi, United Arab Emirates. [614]Catholic University of Daegu, Daegu, Republic of Korea. [615]University of Medicine, Pharmacy, Science and Technology of Târgu Mures, Târgu Mures, Romania. [616]Jivandeep Hospital, Anand, India. [617]Centro de Investigação em Saúde de Angola, Caxito, Angola. [618]South African Medical Research Council, Durban, South Africa. [619]National Dental Care Centre Singapore, Singapore, Singapore. [620]University Hospital of Varese, Varese, Italy. [621]Vietnam National Heart Institute, Hanoi, Vietnam. [622]Clínica de Medicina Avanzada Dr. Abel González, Santo Domingo, Dominican Republic. [623]University of Sarajevo, Sarajevo, Bosnia and Herzegovina. [624]Cardiovascular Prevention Centre Udine, Udine, Italy. [625]University of Pisa, Pisa, Italy. [626]Ministry of Health and Medical Services, Honiara, Solomon Islands. [627]Public Health Agency of Catalonia, Barcelona, Spain. [628]O. M. Marzeyev Institute for Public Health of the National Academy of the Medical Sciences of Ukraine, Kyiv, Ukraine. [629]Universiti Kebangsaan Malaysia, Kuala Lumpur, Malaysia. [630]Ardabil University of Medical Sciences, Ardabil, Iran. [631]Universidade Pedagógica, Maputo, Mozambique. [632]Centre for Disease Prevention and Control, Riga, Latvia. [633]Sulaimani Polytechnic University, Sulaymaniyah, Iraq. [634]Alborz University of Medical Sciences, Karaj, Iran. [635]Ministry of Health, Hanoi, Vietnam. [636]Pure Earth, Dhaka, Bangladesh. [637]Institute of Epidemiology Disease Control and Research, Dhaka, Bangladesh. [638]University of Turku, Turku, Finland. [639]UNICEF, Baku, Azerbaijan. [640]World Health Organization Country Office, Juba, South Sudan. [641]Instituto Federal Riograndense, Rio Grande, Brazil. [642]Institut Universitari d'Investigació en Atenció Primària Jordi Gol, Girona, Spain. [643]Universiti Putra Malaysia, Serdang, Malaysia. [644]University of Malaya, Kuala Lumpur, Malaysia. [645]Sotiria Hospital, Athens, Greece. [646]University of the Philippines, Manila, The Philippines. [647]Slovak Academy of Sciences, Bratislava, Slovakia. [648]University of Santa Cruz do Sul, Santa Cruz do Sul, Brazil. [649]Nutrition Research Foundation, Barcelona, Spain. [650]Minas Gerais State Secretariat for Health, Belo Horizonte, Brazil. [651]CS S. Agustín Ibsalut, Palma, Spain. [652]Amsterdam Institute for Global Health and Development, Amsterdam, The Netherlands. [653]Universidade Nove de Julho, São Paulo, Brazil. [654]Ministerio de Salud, Panama City, Panama. [655]Public Health Agency of Canada, Ottawa, Ontario, Canada. [656]Universidad Industrial de Santander, Bucaramanga, Colombia. [657]Ministry of Health and Social Protection, Bogotá, Colombia. [658]Wuqu' Kawoq, Tecpan, Guatemala. [659]GroundWork, Fläsch, Switzerland. [660]Associazione Calabrese di Epatologia, Reggio Calabria, Italy. [661]University of Minho, Braga, Portugal. [662]Fiji National University, Suva, Fiji. [663]GHESKIO Clinics, Port-au-Prince, Haiti. [664]Universidad de San Carlos, Quetzaltenango, Guatemala. [665]National Center for Epidemiology CIBERESP, Madrid, Spain. [666]Institute of Food Sciences of the National Research Council, Avellino, Italy. [667]Medical University of Gdansk, Gdansk, Poland. [668]Sitaram Bhartia Institute of Science and Research, New Delhi, India. [669]Kindergarten of Avlonari, Evia, Greece. [670]National Institute of Health, Lima, Peru. [671]Ministry of Health, Jakarta, Indonesia. [672]Catalan Department of Health, Barcelona, Spain. [673]Biodonostia Health Research Institute, San Sebastián, Spain. [674]Instituto de Saúde Ambiental, Lisbon, Portugal. [675]Federal University of Alagoas, Maceió, Brazil. [676]South Karelia Social and Health Care District, Lappeenranta, Finland. [677]National Cancer Center, Tokyo, Japan. [678]University of São Paulo Clinics Hospital, São Paulo, Brazil. [679]Hospital Italiano de Buenos Aires, Buenos Aires, Argentina. [680]Medical University of Vienna, Vienna, Austria. [681]Rigshospitalet, Copenhagen, Denmark. [682]Academic Medical Center of University of Amsterdam, Amsterdam, The Netherlands. [683]German Institute of Human Nutrition Potsdam-Rehbruecke, Nuthetal, Germany. [684]The George Institute for Global Health, Sydney, New South Wales, Australia. [685]Center for Oral Health Services and Research Mid-Norway, Trondheim, Norway. [686]Lagos State University College of Medicine, Lagos, Nigeria. [687]University of Las Palmas de Gran Canaria, Las Palmas de Gran Canaria, Spain. [688]Comenius University, Bratislava, Slovakia. [689]Teikyo University, Tokyo, Japan. [690]Finnish Institute of Occupational Health, Helsinki, Finland. [691]Rutgers University, New Brunswick, NJ, USA. [692]National Agency for Public Health, Chisinau, Moldova. [693]St Vincent's Hospital, Sydney, New South Wales, Australia. [694]Nes Municipality, Årnes, Norway. [695]Health Polytechnic Jakarta II Institute, Jakarta, Indonesia. [696]Diponegoro University, Semarang, Indonesia. [697]University of Bari, Bari, Italy. [698]Institut Régional de Santé Publique, Ouidah, Benin. [699]University of Bordeaux, Bordeaux, France. [700]University of Hohenheim, Stuttgart, Germany. [701]Oslo Metropolitan University, Oslo, Norway. [702]Institute of Public Health, Skopje, North Macedonia. [703]Ss. Cyril and Methodius University, Skopje, North Macedonia. [704]Lamprecht und Stamm Sozialforschung und Beratung AG, Zurich, Switzerland. [705]Bonn University, Bonn, Germany. [706]National Institute of Public Health–National Institute of Hygiene, Warsaw, Poland. [707]Kalina Malina Kindergarten, Pazardjik, Bulgaria. [708]The Jikei University School of Medicine, Tokyo, Japan. [709]Fu Jen Catholic University, Taipei, Taiwan. [710]University of Jordan, Amman, Jordan. [711]National Statistical Office, Praia, Cabo Verde. [712]Monash University, Melbourne, Victoria, Australia. [713]Scientific Research Institute of Maternal and Child Health, Ashgabat, Turkmenistan. [714]University of Lincoln, Lincoln, UK. [715]Ministry of Health, Amman, Jordan. [716]UNICEF, Niamey, Niger. [717]University of Applied Sciences Utrecht, Utrecht, The Netherlands. [718]University Medical Center Utrecht, Utrecht, The Netherlands. [719]National Research and Innovation Agency, Jakarta, Indonesia. [720]Health Service, Murcia, Spain. [721]Institut d'Investigacio Sanitaria Illes Balears, Menorca, Spain. [722]University of Bologna, Bologna, Italy. [723]Children's Hospital of Eastern Ontario Research Institute, Ottawa, Ontario, Canada. [724]Hellenic Health Foundation, Athens, Greece. [725]Government Medical College, Bhavnagar, India. [726]Institute of Epidemiology and Preventive Medicine, Taipei, Taiwan. [727]Sefako Makgatho Health Sciences University, Ga-Rankuwa, South Africa. [728]Department of Health, Faga'alu, American Samoa. [729]LBJ Hospital, Faga'alu, American Samoa. [730]Addis Ababa University, Addis Ababa, Ethiopia. [731]Ministry of Health, Wellington, New Zealand. [732]Israel Defense Forces Medical Corps, Tel HaShomer, Israel. [733]Universidad Centro–Occidental Lisandro Alvarado, Barquisimeto, Venezuela. [734]Meharry Medical College, Nashville, TN, USA. [735]University of Tampere Tays Eye Center, Tampere, Finland. [736]Sabiha Gokcen Ilkokulu, Ankara, Turkey. [737]Polytechnic Institute of Porto, Porto, Portugal. [738]Icahn School of Medicine at Mount Sinai, New York City, NY, USA. [739]George Washington University, Washington, DC, USA. [740]Universidad CEU San Pablo, Madrid, Spain. [741]Institute of Clinical Physiology of National Research Council, Pisa, Italy. [742]Universidad San Francisco de Quito, Quito, Ecuador. [743]University Miguel Hernandez, Alicante, Spain.

[744]Université de Lorraine, Nancy, France. [745]Sunflower Nursery School, Craiova, Romania. [746]North Karelia Center for Public Health, Joensuu, Finland. [747]University of the Witwatersrand, Johannesburg, South Africa. [748]Institute for Medical Research, Kuala Lumpur, Malaysia. [749]Xinjiang Medical University, Urumqi, China. [750]Shanghai Educational Development Co. Ltd, Shanghai, China. [751]Ministry of Health and Welfare, Taipei, Taiwan. [752]Ministry of Health and Wellness, Kingston, Jamaica. [753]Örebro University, Örebro, Sweden. [754]St George's, University of London, London, UK. [755]Universitas Indonesia, Jakarta, Indonesia. [756]Rehamed-Center, Tajęcina, Poland. [757]National Yang Ming Chiao Tung University, Taipei, Taiwan. [758]Institute of Food and Nutrition Development of Ministry of Agriculture and Rural Affairs, Beijing, China. [759]Beijing Institute of Ophthalmology, Beijing, China. [760]Children's Hospital of Fudan University, Shanghai, China. [761]University of Cyprus, Nicosia, Cyprus. [762]Niigata University, Niigata, Japan. [763]South China Institute of Environmental Sciences, Guangzhou, China. [764]International Medical University, Shah Alam, Malaysia. [765]Hellenic Mediterranean University, Heraklion, Greece. [766]Iran University of Medical Sciences, Tehran, Iran. [767]Center for Diabetes and Endocrine Care, Srinagar, India. [768]Jagiellonian University, Kraków, Poland. [769]Duke University, Durham, NC, USA. [770]Peking University First Hospital, Beijing, China. [771]Jiangsu Provincial Center for Disease Control and Prevention, Nanjing, China. [772]West Kazakhstan Medical University, Aktobe, Kazakhstan. [773]Inner Mongolia Medical University, Hohhot, China. [774]Przedszkole No. 81, Warsaw, Poland. [775]Johns Hopkins University, Baltimore, MD, USA. [776]These authors contributed equally: Anu Mishra, Bin Zhou, Andrea Rodriguez-Martinez, Honor Bixby. [777]Deceased: Konrad Jamrozik, Altan Onat, Robespierre Ribeiro, Michael Sjöström, Agustinus Soemantri, Jutta Stieber, Dimitrios Trichopoulos. ✉e-mail: majid.ezzati@imperial.ac.uk

## Methods

We estimated trends in mean height and BMI for children and adolescents aged 5–19 years from 1990 to 2020 by rural and urban place of residence for the 200 countries and territories listed in Supplementary Table 1. We pooled, in a Bayesian meta-regression, repeated cross-sectional population-based data on height and BMI. Our results represent estimates of height and BMI for children and adolescents of the same age over time (that is, for successive cohorts) in rural and urban settings for each country.

### Data sources

We used a database on cardiometabolic risk factors collated by NCD-RisC. Data were obtained from publicly available multi-country and national measurement surveys, for example, Demographic and Health Surveys (DHS), WHO-STEPwise approach to Surveillance (STEPS) surveys, and those identified through the Inter-University Consortium for Political and Social Research, UK Data Service and European Health Interview & Health Examination Surveys Database. With the help of the WHO and its regional and country offices as well as the World Heart Federation, we identified and accessed population-based survey data from national health and statistical agencies. We searched and reviewed published studies as previously detailed[54] and invited eligible studies to join NCD-RisC, as we did with data holders from earlier pooled analyses of cardiometabolic risk factors[55–58]. The NCD-RisC database is continuously updated through all the above routes and through periodic requests to NCD-RisC members to ask them to suggest additional sources in their countries.

We carefully checked that each data source met our inclusion criteria, as listed below. Potential duplicate data sources were first identified by comparing studies from the same country and year, followed by checking with NCD-RisC members that had provided data about whether the sources from the same country and year, with similar samples, were the same or distinct. If two sources were confirmed as duplicates, one was discarded. All NCD-RisC members were also periodically asked to review the list of sources from their country to verify that the included data met the inclusion criteria and were not duplicates.

For each data source, we recorded the study population, the sampling approach, the years of measurement and the measurement methods. Only data that were representative of the population were included. All data sources were assessed in terms of whether they covered the entire country, one or more subnational regions (that is one or more provinces or states, more than three cities, or more than five rural communities), or one or a small number of communities (limited geographical scope not meeting above national or subnational criteria), and whether participants in rural, urban or both areas were included. As stated in the sections on the statistical model, these study-level attributes were used in the Bayesian hierarchical model to estimate mean height and BMI by country, year, sex, age and place of residence using all available data while taking into account differences in the populations from which different studies had sampled. All submitted data were checked by at least two independent individuals. Questions and clarifications were discussed with NCD-RisC members and resolved before data were incorporated into the database.

Anonymized individual data from the studies in the NCD-RisC database were re-analysed according to a common protocol. We calculated the mean height and the mean BMI, and the associated standard errors, by sex, single year of age from 5 to 19 years and rural or urban place of residence. Additionally, for analysis of height, participants aged 20–30 years were included, assigned to their corresponding birth cohort, because mean height in these ages would be at least that when they were aged 19 years given that the decline in height with age begins in the third and fourth decades of life[59]. All analyses incorporated sample weights and complex survey design, when applicable,

in calculating summary statistics. For studies that had used simple random sampling, we calculated the mean as the average of all individuals within the group and the associated standard error (s.d. divided by the square root of sample size); for studies that had used multistage (stratified) sampling, we accounted for survey design features, including clusters, strata and sample weights, to weight each observation by the inverse sampling probability and estimated standard error through Taylor series linearization, as implemented in the R 'survey' package[60]. Computer code was provided to NCD-RisC members who requested assistance. For surveys without information on the place of residence, we calculated summary statistics stratified by age and sex for the entire sample, which represented the population-weighted sum of rural and urban means; data on the share of population in urban and rural areas were from the United Nations Population Division[61].

Additionally, summary statistics for nationally representative data from sources that were identified but not accessed using the above routes were extracted from published reports. Data were also extracted for two STEPS surveys that were not publicly available. We also included data from a previous global-data pooling study[58], when not accessed through the above routes.

### Data inclusion and exclusion

Data sources were included in the NCD-RisC height and weight database if the following criteria were met: measured data on height and weight were available; study participants were 5 years of age or older; data were collected using a probabilistic sampling method with a defined sampling frame; data were from population samples at the national, subnational or community level as defined above; and data were from the countries and territories listed in Supplementary Table 1.

We excluded all data sources that were solely based on self-reported weight and height without a measurement component because these data are subject to biases that vary by geography, time, age, sex and socioeconomic characteristics[62–64]. Owing to these variations, approaches to correcting self-reported data may leave residual bias. We also excluded data sources on population subgroups for which anthropometric status may differ systematically from the general population, including the following: studies that had included or excluded people based on their health status or cardiovascular risk; studies in which participants were only ethnic minorities; specific educational, occupational or socioeconomic subgroups (with the exception noted below); those recruited through health facilities (with the exception noted below); and females aged 15–19 years in surveys that sampled only ever-married women or measured height and weight only among mothers.

We used school-based data in countries and age–sex groups with school enrolment of 70% or higher. We used data for which the sampling frame was health insurance schemes in countries where at least 80% of the population were insured. Finally, we used data collected through general practice and primary care systems in high-income and central European countries with universal insurance because contact with the primary care systems tends to be as good as or better than response rates for population-based surveys.

We excluded participants whose age was <18 years and whose data were not reported by single year of age (<0.01% of all participants) because height and weight may have nonlinear age associations in these ages, especially during growth spurts. We excluded BMI data for females who were pregnant at the time of measurement (<0.01% of all participants). We excluded <0.2% of all participants who had recorded height: <60 cm or >180 cm for ages <10 years; <80 cm or >200 cm for ages 10–14 years; <100 cm or >250 cm for ages ≥15 years, or who had recorded weight: <5 kg or >90 kg for age <10 years; <8 kg or >150 kg for ages 10–14 years; <12 kg or >300 kg for ages ≥15 years, or who had recorded BMI: <6 kg m$^{-2}$ or >40 kg m$^{-2}$ for ages <10 years; <8 kg m$^{-2}$ or >60 kg m$^{-2}$ for ages 10–14 years; <10 kg m$^{-2}$ or >80 kg m$^{-2}$ for ages ≥15 years.

## Conversion of BMI prevalence metrics to mean BMI

In 0.5% of our data points, mostly extracted from published reports or from a previous pooling analysis[58], the mean BMI was not reported but data were available for the prevalence of one or more BMI categories, for example BMI ≥30 kg m$^{-2}$. To use these data, we used previously validated conversion regressions[65] to estimate the missing primary outcome from the available BMI prevalence metric or metrics. Additional details on regression model specifications along with the regression coefficients are reported at https://github.com/NCD-RisC/ncdrisc-methods/.

## Statistical model overview

We used a Bayesian hierarchical meta-regression model to estimate the mean height and BMI by country, year, sex, age and place of residence using the aforementioned data. For presentation, we summarized the 15 age-specific estimates, for single years of age from 5 to 19 years, through age standardization, which puts the child and adolescent population for each country-year on the same age distribution, and hence enables comparisons to be made over time and across countries. We generated age-standardized estimates by taking weighted means of age-specific estimates using age weights from the WHO standard population[66]. We also show results, graphically and numerically, for index ages of 5, 10, 15 and 19 years in the Supplementary Information.

The statistical model is described in detail in statistical papers[67,68], related substantive papers[7,20,21,55–58,65,69] and in the section below on model specification. In summary, the model had a hierarchical structure in which estimates for each country and year were informed by its own data, if available, and by data from other years in the same country and from other countries, especially those in the same region and super-region, with data for similar time periods. The extent to which estimates for each country-year were influenced by data from other years and other countries depended on whether the country had data, the sample size of the data, whether they were national, and the within-country and within-region variability of the available data. For the purpose of hierarchical analysis, countries and territories were organized into 21 regions, mostly based on geography and national income (Supplementary Table 1). Regions were in turn organized into nine super-regions.

We used observation year, that is, the year in which data were collected, as the timescale for the analysis of BMI and birth year as the timescale for the analysis of height, consistent with previous analyses[7,65,70]. Time trends were modelled through a combination of a linear term, to capture gradual long-term change, and a second-order random walk, which allows for nonlinear trends[71], both modelled hierarchically. The age associations of height and BMI were modelled, using cubic splines, to allow for nonlinear changes over age, including periods of rapid and slow rise. Periods of rapid rise representing adolescent growth spurts, which occur earlier in girls than boys[72–74], were reflected in the placement of spline knots for boys and girls, respectively, as detailed in the section on model specification. Spline coefficients were allowed to vary across countries, informed by their own data as well as data from other countries as specified by a hierarchical structure, as previously described[69].

The model also accounted for the possibility that height or BMI in subnational and community samples might differ systematically from nationally representative samples and have larger variation than in national studies. These features were accounted for through the inclusion of fixed-effect and random-effect terms for subnational and community data as detailed in the model specification section below. The fixed effects accounted for systematic differences between subnational or community studies and national studies. The inclusion of random effects allowed national data to have greater influence on the estimates than subnational or community data with similar sample sizes because the subnational and community data have additional variance from the random-effect terms. Both were estimated empirically as a part of model fitting.

Following the approach of previous papers[20,21,67], the model included parameters representing the urban–rural height or BMI difference, which is empirically estimated and allowed to vary by country and year. We further expanded the model to allow urban–rural difference in height or BMI to vary by age, as height or weight with age may vary between children residing in rural versus urban areas. If data for a country-year-age group contained a mix of children living in urban and rural areas but were not stratified by place of residence (21% of all data sources), the estimated height or BMI difference was informed by stratified data from other age groups, years and countries, especially those in the same region with data from similar time periods and/or ages.

## Statistical model specification

As stated earlier, for each data source, we calculated mean height and BMI, together with corresponding standard errors, stratified by sex, age and rural or urban place of residence. For sources that did not stratify the sample on the place of residence, we obtained age-and-sex-stratified data. Each study contributed up to 30 mean BMI data points or 32 mean height data points for each sex, with the exact number depending on how many age groups were represented in the study and whether the study provided data stratified on urban and rural place of residence. The likelihood for an observation at urbanicity level $s$ (urban-only, rural-only or mixed; referred to as stratum hereafter) and age group $h$, with age $z_h$, from study $i$, carried out in country $j$ at time $t$ is as follows:

$$y_{s,h,i} \sim N(a_{j[i]} + b_{j[i]}t_i + u_{j[i],t_i} + \gamma_i(z_h) + X_i\beta + e_i$$
$$+ I_{s,i}[p_{j[i]} + q_{j[i]}t_i + r_{j[i]}z_h + d_i], SD^2_{s,h,i}/n_{s,h,i} + \tau^2),$$

where the country-specific intercept and linear time slope from the $j$th country ($j = 1 ... J$, where $J = 200$ which is the total number of countries in our analysis) are denoted $a_j$ and $b_j$, respectively. We describe the hierarchical model used for the $a$'s and $b$'s in the section 'Linear components of country time trends'. Letting $T = 31$ be the total number of years from 1990 to 2020, the $T$-length vector $u_j$ captures smooth nonlinear change over time in country $j$, as described in the section 'Nonlinear change'. The age effects of the $h$th age group (with age $z_h$) in study $i$ are denoted by $\gamma_i$; we describe the age model in the section 'Age model'. The matrix $X$ contains terms describing whether studies were representative at the national, subnational or community level. In addition, a random effect, $e_i$, is estimated for each study, described in the section 'Study-level term and study-specific random effects'.

**Linear components of country time trends.** The model had a hierarchical structure, whereby studies were nested in countries, which were nested in regions (indexed by $k$), which were nested in super-regions (indexed by $l$), which were all nested in the globe (see Supplementary Table 1 for a list of countries and territories in each region, and regions in each super-region). This structure allowed the model to share information across units to a greater degree when data were non-existent or weakly informative (for example, had a small sample size or were not nationally representative) and, to a lesser extent, in data-rich countries and regions[75].

The $a$ and $b$ terms are country-specific linear intercepts and time slopes with terms at each level of the hierarchy, denoted by the superscripts $c$, $r$, $s$ and $g$, respectively:

$$a_j = a^c_j + a^r_{k[j]} + a^s_{l[k]} + a^g,$$
$$b_j = b^c_j + b^r_{k[j]} + b^s_{l[k]} + b^g,$$
$$a^x_j \sim N(0, \kappa^x_a),$$
$$b^x_j \sim N(0, \kappa^x_b),$$

where $x = \{c, r, s\}$.

The $\kappa$ terms were each assigned a flat prior on the s.d. scale[76]. We also assigned flat priors to $a^g$ and $b^g$.

**Nonlinear change.** Mean BMI or height may change nonlinearly over time[7,54,58,65,70]. We captured smooth nonlinear change in time in urban and rural strata of country $j$ using the vector $u_j$. Just as $a_j$ and $b_j$ are each defined as the sum of country, region, super-region and global components, we defined

$$u_j = u_j^c + u_{k[j]}^r + u_{l[k]}^s + u^g.$$

To allow the model to differentiate between the degrees of nonlinearity that exist at the country, region, super-region and global levels, we assigned the four components of each $u$ a Gaussian autoregressive prior[71,77]. In particular, the $T$ vectors $u_j^c$ ($j=1...J$), $u_k^r$ ($k=1...K$), $u_l^s$ ($l=1...L$) and $u^g$ each have a normal prior with mean zero and precision $\lambda_c P$, $\lambda_r P$, $\lambda_s P$ and $\lambda_g P$, respectively, where the scaled precision matrix $P$ in the Gaussian autoregressive prior penalizes first and second differences as follows:

$$P = \begin{bmatrix} 1 & 0 & 0 & \cdots & 0 \\ -2 & 1 & 0 & \cdots & 0 \\ 1 & -2 & 1 & \cdots & 0 \\ 0 & 1 & -2 & \cdots & 0 \\ 0 & 0 & 1 & \cdots & 0 \\ \vdots & \vdots & \vdots & \ddots & \vdots \\ 0 & 0 & 0 & \cdots & 1 \end{bmatrix} \begin{bmatrix} 1 & -2 & 1 & 0 & 0 & \cdots & 0 \\ 0 & 1 & -2 & 1 & 0 & \cdots & 0 \\ 0 & 0 & 1 & -2 & 1 & \cdots & 0 \\ \vdots & \vdots & \vdots & \vdots & \vdots & \ddots & \vdots \\ 0 & 0 & 0 & 0 & 0 & \cdots & 1 \end{bmatrix}$$

$$= \begin{bmatrix} 1 & -2 & 1 & 0 & 0 & \cdots & 0 \\ -2 & 5 & -4 & 1 & 0 & \cdots & 0 \\ 1 & -4 & 6 & -4 & 1 & \cdots & 0 \\ 0 & 1 & -4 & 6 & -4 & \cdots & 0 \\ 0 & 0 & 1 & -4 & 6 & \cdots & 0 \\ \vdots & \vdots & \vdots & \vdots & \vdots & \ddots & \vdots \\ 0 & 0 & 0 & 0 & 0 & \cdots & 1 \end{bmatrix}.$$

$P$ is multiplied by the estimated precision parameters $\lambda_c$, $\lambda_r$, $\lambda_s$ and $\lambda_g$, thus upweighting or downweighting the strength of its penalties and ultimately determining the degree of smoothing at each level. For each of the four precision parameters, we used a truncated flat prior on the s.d. scale ($1/\sqrt{\lambda}$)[76]. We truncated these priors such that $\log \lambda \le 20$ for each of the four $\lambda$'s. This upper bound is enforced as a computational convenience, whereby models with $\log \lambda > 20$ are treated as equivalent to a model with $\log \lambda = 20$ as they essentially have no extralinear variability in time. In practice, this upper bound had little effect on the parameter estimates. Furthermore, we ordered the $\lambda$'s a priori as follows: $\lambda_c < \lambda_r < \lambda_s < \lambda_g$. This prior constraint conveys the natural expectation that, for example, the global height or BMI trend has less extralinear variability than the trend of any given region, which in turn has less variability than those of constituent countries.

The matrix $P$ has rank $T-2$, corresponding to a flat, improper prior on the mean and the slope of the $u_j^c$'s, the $u_k^r$'s and the $u_l^s$'s and $u^g$, and is not invertible[78]. Thus, we had a proper prior in a reduced-dimension space[71], with the prior expressed as follows:

$$P(u_j^c | \lambda_c) \propto \lambda_c^{\frac{T-2}{2}} \exp\left\{-\frac{\lambda_c}{2} u_j^{c\prime} P u_j^c\right\}.$$

Note that if $u_j^c$ had a non-zero mean, this would introduce non-identifiability with respect to $a_j^c$. By the same token, $b_j^c$ would not be identifiable if $u_j$ had a non-zero time slope, and similarly for the other means and slopes. Thus, to achieve identifiability of the $a$'s, $b$'s, and $u$'s, we constrained the mean and slope of $u^g$ and of each $u^s$, $u^r$ and $u^c$ to be zero. Enforcing orthogonality between the linear and nonlinear portions of the time trends meant that each can be interpreted independently.

For the cases in which we have observations for at least two different time points, this improper prior will not lead to an improper posterior

because the data will provide information about the mean and slope. In order to enforce the desired orthogonality between the linear and nonlinear portions of the model, we constrained the mean and slope of the $u_j^c$'s, $u_k^r$'s and $u_l^s$'s and of $u^g$ to be zero[71].

For the six countries with no height data, and seven countries with no BMI data, we took the Moore–Penrose pseudoinverse of $P$[79], setting to infinity those eigenvalues that correspond to the non-identifiability. This effectively constrained the non-identified portions of the model to zero, as the corresponding variances are set to zero[77]; in this case the Rue and Held correction[71] is not needed. An intermediate case occurs when data are observed for only one time point in a country. In this case, the full conditional precision has rank $T-1$ because the mean but not the linear trend of $u_j^c$ is identified by the data. We therefore constrained the linear trend of $u_j^c$ to zero by taking the generalized inverse of the full conditional precision. We then constrained the mean of $u_j^c$ to zero using the one-dimensional version of the Rue and Held correction[71].

**Age model.** To capture sex-specific patterns of growth, especially adolescent growth spurts, we modelled age using cubic splines. The number and position of the knots of the splines were selected on the basis of a combination of physiological and statistical considerations, as described in a national level analysis[7]. For age group $h$ with age $z_h$, in study $i$, the age effect for height and BMI is given, respectively, as follows:

$$\gamma_i(z_h) = \gamma_{1i} z_h + \gamma_{2i} z_h^2 + \gamma_{3i} z_h^3 + \gamma_{4i}(z_h - k_1)_+^3 + \gamma_{5i}(z_h - k_2)_+^3 \\ + \gamma_{6i}(z_h - k_3)_+^3 + \gamma_{7i}(z_h - k_4)_+^3, \quad \text{(height)}$$

$$\gamma_i(z_h) = \gamma_{1i} z_h + \gamma_{2i} z_h^2 + \gamma_{3i} z_h^3 + \gamma_{4i}(z_h - k_1)_+^3 + \gamma_{5i}(z_h - k_2)_+^3. \quad \text{(BMI)}$$

For height, four spline knots were placed at ages $\{k_1, k_2, k_3, k_4\} = \{8, 10, 12, 14\}$ for girls and at ages $\{k_1, k_2, k_3, k_4\} = \{10, 12, 14, 16\}$ for boys. For BMI, we used two spline knots (at ages 10 and 15 years) because, at the population level, changes in BMI with age are smoother than those in height[7,72,73]. Each of the spline coefficients was allowed to vary across countries, with a hierarchical structure as described in a previous paper[69], using the equation below, where $\psi$ is the global intercept, and $c$, $r$ and $s$ are the country, region and super-region random intercepts, respectively. The $k$th age effect coefficients for study $i$ ($\gamma_{k,i}$) for each age group $h$, with age $z_h$, are given as follows:

$$\gamma_{k,i} = \psi_k + c_{k,j[i]} + r_{k,l[i]} + s_{k,m[i]},$$
$$c_{k,j} \sim N(0, \sigma_{k,c}^2),$$
$$r_{k,l} \sim N(0, \sigma_{k,r}^2),$$
$$s_{k,l} \sim N(0, \sigma_{k,s}^2).$$

A flat improper prior was placed on each of the $\sigma_k$'s.

**Study-level term and study-specific random effects.** Mean height or BMI from individual studies may deviate from the true country-year mean owing to factors associated with sampling, response or measurement. We used a study-level term to help account for potential systematic differences associated with data sources that are representative of subnational and community populations. Our model therefore included time-varying offsets (referred to as fixed effects above) for subnational and community data in the term $X_i\beta$:

$$X_i\beta = \beta_1 I\{X_{j[i],t[i]}^{cvrg} = \text{subnational}\} + \beta_2 I\{X_{j[i],t[i]}^{cvrg} = \text{subnational}\} t_i \\ + \beta_3 I\{X_{j[i],t[i]}^{cvrg} = \text{community}\} + \beta_4 I\{X_{j[i],t[i]}^{cvrg} = \text{community}\} t_i,$$

where $X_{j[i],t[i]}^{cvrg}$ is the indicator for whether the coverage of study $i$, in country $j$ and year $t$, is subnational or community.

Even after accounting for sampling variability, national studies may still not reflect the true mean height or BMI level of a country with perfect accuracy, and subnational and community studies have even larger variability. In study $i$, the study-specific random effect $e_i$ allows all age groups from the same study to have an unusually high or an unusually low mean after conditioning on the other terms in the model. Each $e_i$ is assigned a normal prior with variance depending on whether study $i$ is representative at the national, subnational or community level. Random effects from national studies were constrained to have smaller variance ($v_n$) than random effects of subnational studies ($v_s$), which were in turn constrained to have smaller variance than community studies ($v_c$). To make country-level predictions, we set $e_i = 0$, thus not including random effects arising from imperfections and variations in study design and implementation and from within-country variability of height or BMI means.

**Urban and rural strata.** To model mean height and BMI by urban and rural places of residence, the model included offsets for the two strata. The offsets were captured by country-specific intercept, linear time and age effects, using a centred indicator term ($I_{s,i}$):

$$I_{s,i}[p_{j[i]} + q_{j[i]}t_i + r_{j[i]}z_h + d_i],$$

where $I_{s,i} = -1 + 2X^{\mathrm{urb}}_{s,i}$, with

$$X^{\mathrm{urb}}_{s,i} = \begin{cases} 1, & \text{if stratum } s \text{ contains only urban individuals,} \\ 0, & \text{if stratum } s \text{ contains only rural individuals,} \\ X^{\mathrm{urb}}_{j[i],t[i]}, & \text{if stratum } s \text{ contains a mixture of} \\ & \quad \text{urban and rural individuals.} \end{cases}$$

In other words, for data not stratified by place of residence, the model treated the unstratified mean height or BMI as equivalent to the weighted sum of the (unobserved) urban sample mean height or BMI and rural sample mean height or BMI, with the weights based on the proportion of the population of that country living in urban areas in the year of the survey ($X^{\mathrm{urb}}_{j[i],t[i]}$).

The intercept ($p$) and slope ($q$) terms capture the country-to-country variation in the magnitude of the height or BMI difference between urban and rural populations and how the difference changes over time. The slope ($r$) captures the country-to-country variation in the BMI or height difference between urban and rural populations across age groups. These were specified with the same geographical hierarchy as the country-specific intercepts ($a$) and slopes ($b$) as follows:

$$p_j = p_j^c + p_{k[j]}^r + p_{l[k]}^s + p^{\mathrm{g}},$$
$$q_j = q_j^c + q_{k[j]}^r + q_{l[k]}^s + q^{\mathrm{g}},$$
$$r_j = r_j^c + r_{k[j]}^r + r_{l[k]}^s + r^{\mathrm{g}},$$
$$p_j^x \sim N(0, \kappa_p^x),$$
$$q_j^x \sim N(0, \kappa_q^x),$$
$$r_j^x \sim N(0, \kappa_r^x),$$

where $x = \{c, r, s\}$. The study random effect term $d_i$ incorporates deviations from the country-level urban–rural difference in each study and is analogous to $e_i$.

**Residual age-by-study variability.** The age patterns across communities within a given country may differ from the overall age pattern of that country. This within-study variability cannot be captured by the $e$ terms, which are equal across age-specific observations in each study, so we included an additional variance component for each study, $\tau^2$.

**Model implementation**
All analyses were done separately by sex because age, geographical and temporal patterns of height and BMI differ between girls and boys[7,65]. We fitted the statistical model using Markov chain Monte Carlo (MCMC). We started 35 parallel MCMC runs from randomly generated overdispersed starting values. For computational efficiency, each chain was run for a total of 75,000 iterations. All chains converged to the same target distribution within this number, but due to the overdispersed initial values, the length of burn-in required to converge to the target distribution varied. After the runs were completed, we used trace plots to monitor convergence and to select chains that had completed burn-in within 35,000 iterations. This resulted in 16 chains for boys and 17 for girls for BMI, and 14 chains for boys and 16 for girls for height. Within each of these chains, post-burn-in iterations were thinned by keeping every 10th iteration, which were then combined for all chains and further thinned to a final set of 5,000 draws of the model parameter estimates. We used the posterior distribution of the model parameters to obtain the posterior distributions of our outcomes: mean urban and rural height and BMI, and the urban–rural difference in mean height and BMI. Posterior estimates were made for one-year age groups from 5 to 19 years, as well as for age-standardized outcomes, by year. The reported CrIs represent the 2.5th and the 97.5th percentiles of the posterior distributions. We also report the posterior s.d. of estimates, and PP that the estimated change in height or BMI in rural or urban areas, and in the urban–rural height or BMI difference over time, represents a true increase or decrease.

Convergence was confirmed for the country-sex specific posterior outcomes—namely mean urban height and BMI, mean rural height and BMI and the urban–rural difference in mean height and BMI—for reporting ages (5, 10, 15, 19 years and age-standardized) and years (1990 and 2020) using the R-hat diagnostic[80,81]. For height, the 2.5th to 97.5th percentiles of the R-hats for the reporting ages and years were 0.999–1.010 for girls and 0.999–1.004 for boys. For BMI, the 2.5th to 97.5th percentiles of the R-hats were 0.999–1.004 for girls and 0.999–1.005 for boys.

We applied the pool-adjacent-violators algorithm, a monotonic regression that uses an iterative algorithm based on least squares to fit a free-form line to a sequence of observations such that the fitted line is non-decreasing[82,83], on the posterior height estimates to ensure that the height for each birth cohort increased monotonically with age. In practice, this had little effect on the results, with height at age 19 years adjusted by an average of 0.26 cm or less for both boys and girls. All analyses were conducting using the statistical software R (v.4.1.2)[84].

**Strengths and limitations**
An important strength of our study is its novel scope of presenting consistent and comparable estimates of urban and rural height and BMI among school-aged children and adolescents, which is essential to formulate and evaluate policies that aim to improve health in these formative ages. We used a large number of population-based studies from 194 countries and territories covering around 99% of the population of the world. We maintained a high level of data quality and representativeness through repeated checks of study characteristics against our inclusion and exclusion criteria, and did not use any self-reported data to avoid bias in height and weight. Data were analysed according to a consistent protocol, and the characteristics and quality of data from each country were rigorously verified through repeated checks by NCD-RisC members. We used a statistical model that used all available data and took into account the epidemiological features of height and BMI during childhood and adolescence by using nonlinear time trends and age associations. The model used the available information on the urban–rural difference in height and BMI and estimated the age-varying and time-varying urban–rural difference for all countries and territories hierarchically.

Despite our extensive efforts to identify and access data, some countries had fewer data, especially those in the Caribbean, Polynesia, Micronesia and sub-Saharan Africa. Of the studies used, fewer than half had data for children aged 5–9 years compared to nearly 90% with data for children and adolescents aged 10–19 years. The scarcity of data is reflected in the larger uncertainty of our estimates for these countries and regions, and younger age groups. This reflects the need to systematically include school-aged children in both health and nutrition surveys, and, especially in countries where school enrolment is high, to use schools as a platform for monitoring growth and developmental outcomes for entire national populations and key subgroups such as those in rural and urban areas. Although urban and rural classifications are commonly based on definitions by national statistical offices, classification of cities and rural areas may, appropriately, vary by country according to their demographic characteristics (for example, population size or density), economic activities, administrative structures, infrastructure and environment. Similarly, urbanization takes place through a variety of mechanisms such as changes in fertility in rural and urban areas, migration and reclassification of previously rural areas to urban as they grow and industrialize. Each of these mechanisms may have different implications for nutrition and physical activity, and hence height and/or BMI, and should be a subject of studies that follow individual participants and changes in their place of residence. Finally, there is variation in growth and development of children within rural or urban areas based on household socioeconomic status and community characteristics that affect access to and the quality of nutrition, the living environment and healthcare[35,85,86]. Among these, in some cities, a large number of families live in slums[19,87]. School-aged children and adolescents living in slums have nutrition, environment and healthcare access that is typically worse than other residents of the city, although often better than those in rural areas[19,87–90].

## Reporting summary

Further information on research design is available in the Nature Portfolio Reporting Summary linked to this article.

## Data availability

Estimates of mean BMI and height by country, year, sex, single year of age as well as age-standardized, and place of residence (urban and rural) will be available from https://www.ncdrisc.org in machine-readable numerical format and as visualizations upon publication of the paper. Input data from publicly available sources and contact information for data providers can be downloaded from https://www.ncdrisc.org and Zenodo (https://doi.org/10.5281/zenodo.7355601).

## Code availability

The computer code for the Bayesian hierarchical model and the code used to generate figures in this work will be available at https://www.ncdrisc.org and Zenodo (https://doi.org/10.5281/zenodo.7355601).

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

**Acknowledgements** This study was funded by the UK Medical Research Council (grant number MR/V034057/1), the Wellcome Trust (Pathways to Equitable Healthy Cities grant 209376/Z/17/Z), the AstraZeneca Young Health Programme and the European Commission (STOP project through EU Horizon 2020 research and innovation programme under Grant Agreement 774548). For the purpose of open access, the author has applied a Creative Commons Attribution (CC BY) licence to the Author Accepted Manuscript version arising from this submission. We thank W.Dietz, L.Jaacks and W.Johnson for recommendations of relevant citations. The authors alone are responsible for the views expressed in this Article and they do not necessarily represent the views, decisions, or policies of the institutions with which they are affiliated.

**Author contributions** A.M., B.Z., A.R.M., H.B. and R.K.S. led the data collection and management. A.M., B.Z., A.R.M., H.B., C.J.P., J.E.B. and M.E. developed the statistical method. A.M., B.Z., A.R.M. and H.B. coded the statistical method. A.M. conducted analyses and prepared results. The other authors contributed to study design, and collected, reanalysed, checked and pooled data. M.E., A.M., B.Z., A.R.M. and H.B. wrote the first draft of the report. All other authors commented on the draft report.

**Competing interests** M.E. reports a charitable grant from the AstraZeneca Young Health Programme.

**Additional information**
**Correspondence and requests for materials** should be addressed to Majid Ezzati.

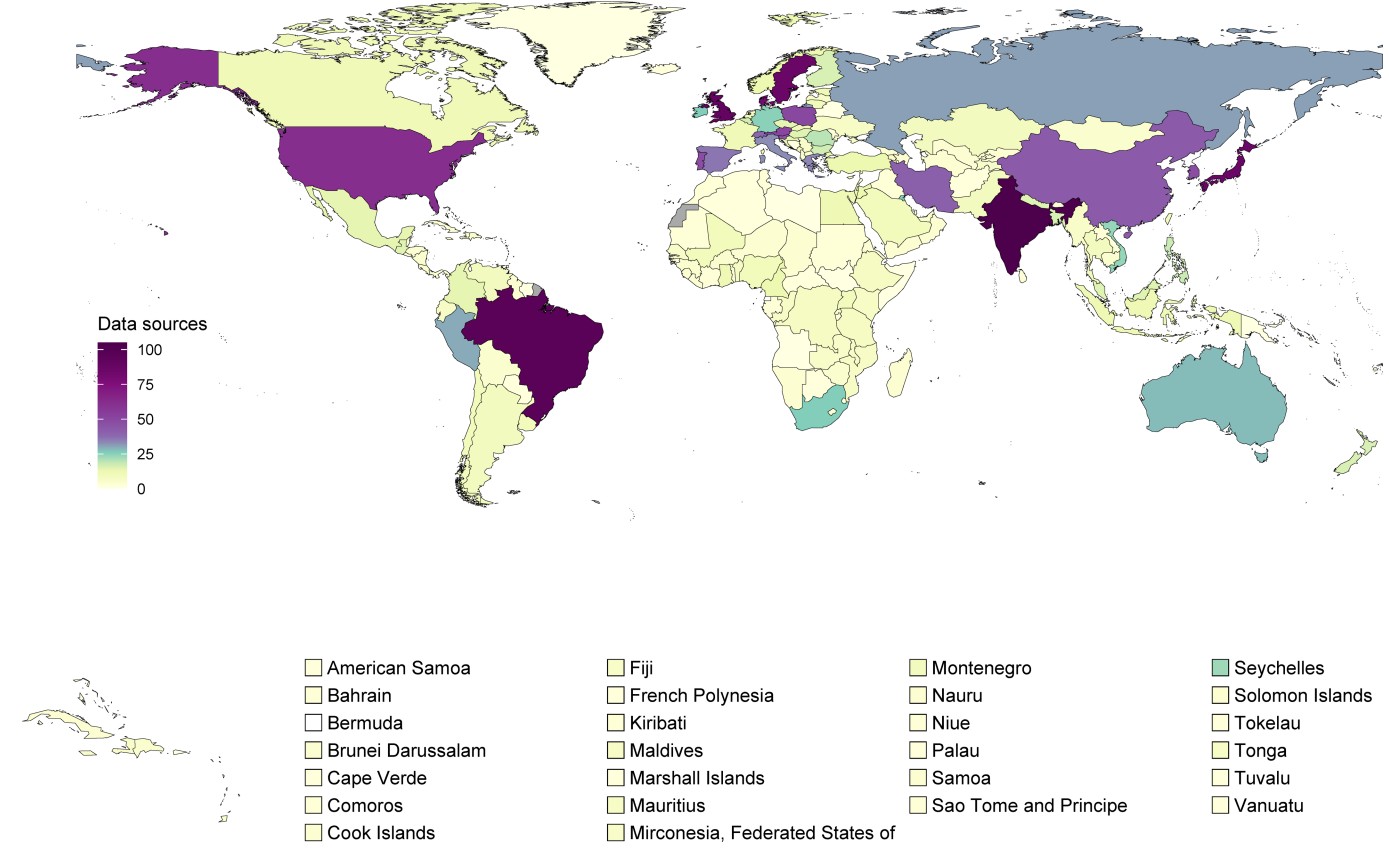

| | | | |
|---|---|---|---|
| ☐ American Samoa | ☐ Fiji | ☐ Montenegro | ☐ Seychelles |
| ☐ Bahrain | ☐ French Polynesia | ☐ Nauru | ☐ Solomon Islands |
| ☐ Bermuda | ☐ Kiribati | ☐ Niue | ☐ Tokelau |
| ☐ Brunei Darussalam | ☐ Maldives | ☐ Palau | ☐ Tonga |
| ☐ Cape Verde | ☐ Marshall Islands | ☐ Samoa | ☐ Tuvalu |
| ☐ Comoros | ☐ Mauritius | ☐ Sao Tome and Principe | ☐ Vanuatu |
| ☐ Cook Islands | ☐ Mirconesia, Federated States of | | |

**Extended Data Fig. 1 | Number of data sources used in the analysis, by country.**

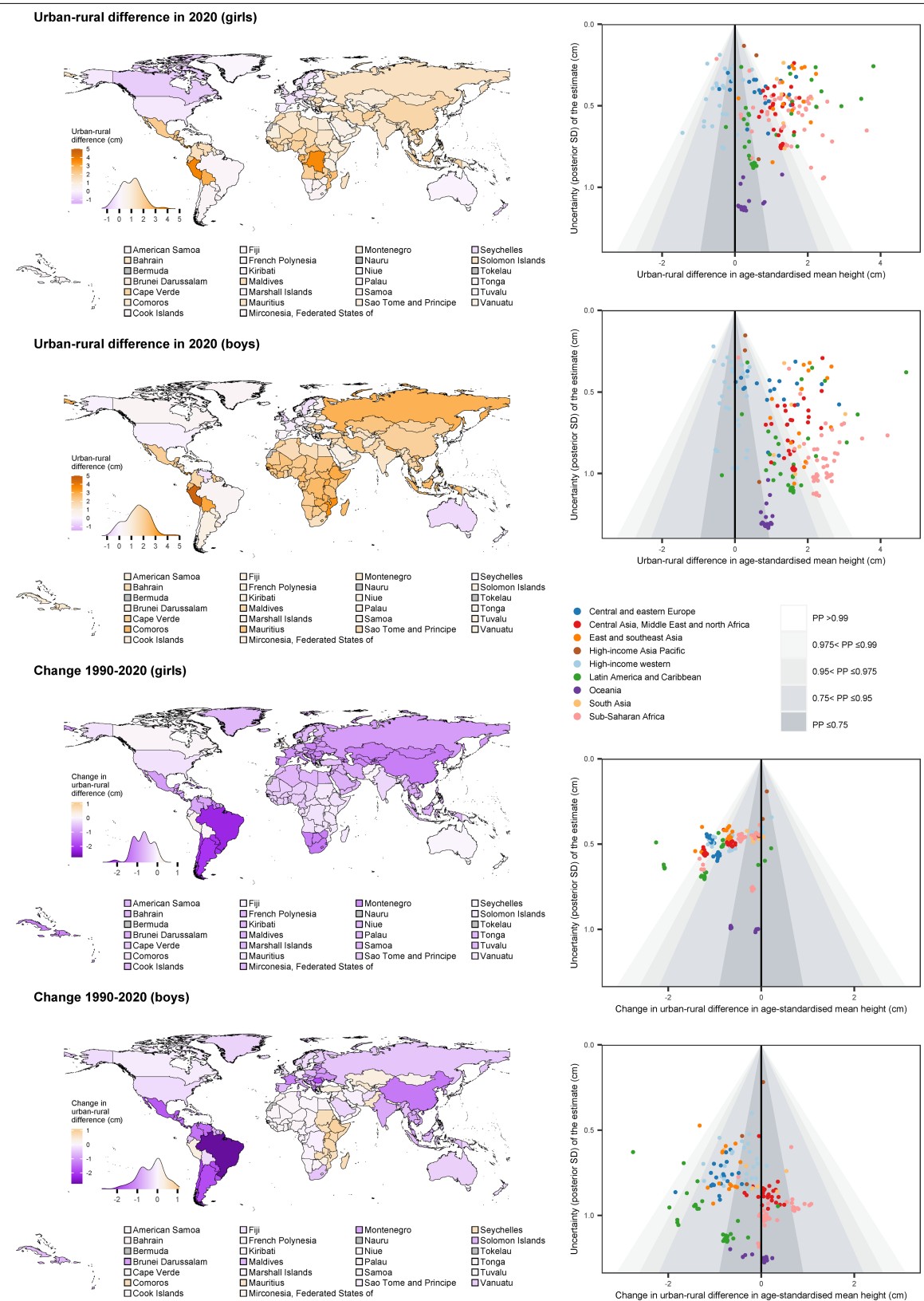

**Extended Data Fig. 2** | See next page for caption.

**Extended Data Fig. 2 | Urban-rural height difference in 2020 and change from 1990 to 2020.** The top two maps show the urban-rural difference in age-standardised mean height in 2020 for girls and boys resepectively. A positive number shows higher urban mean height and a negative number shows higher rural mean height. The bottom two maps show the change from 1990 to 2020. The density plot below each map shows the distribution of estimates across countries. The top two scatter plots show the urban-rural difference in age-standardised mean height in relation to the uncertainty of the difference measured by posterior s.d. The bottom two scatter plots show the change from 1990 to 2020 in urban-rural difference in mean height in relation to the uncertainty of the change measured by posterior s.d. Each point in the scatter plots shows one country. Shaded areas approximately show the posterior probability (PP) of a true difference (top two scatter plots) and of a true increase or decrease in difference (bottom two scatter plots). See Extended Data Fig. 8 for PPs of the urban-rural difference in age-standardised mean height and its change. See Supplementary Fig. 7 for results at ages 5, 10, 15 and 19 years. We did not estimate the difference between rural and urban height for countries classified as entirely urban (Bermuda, Kuwait, Nauru and Singapore) or entirely rural (Tokelau), as indicated in grey.

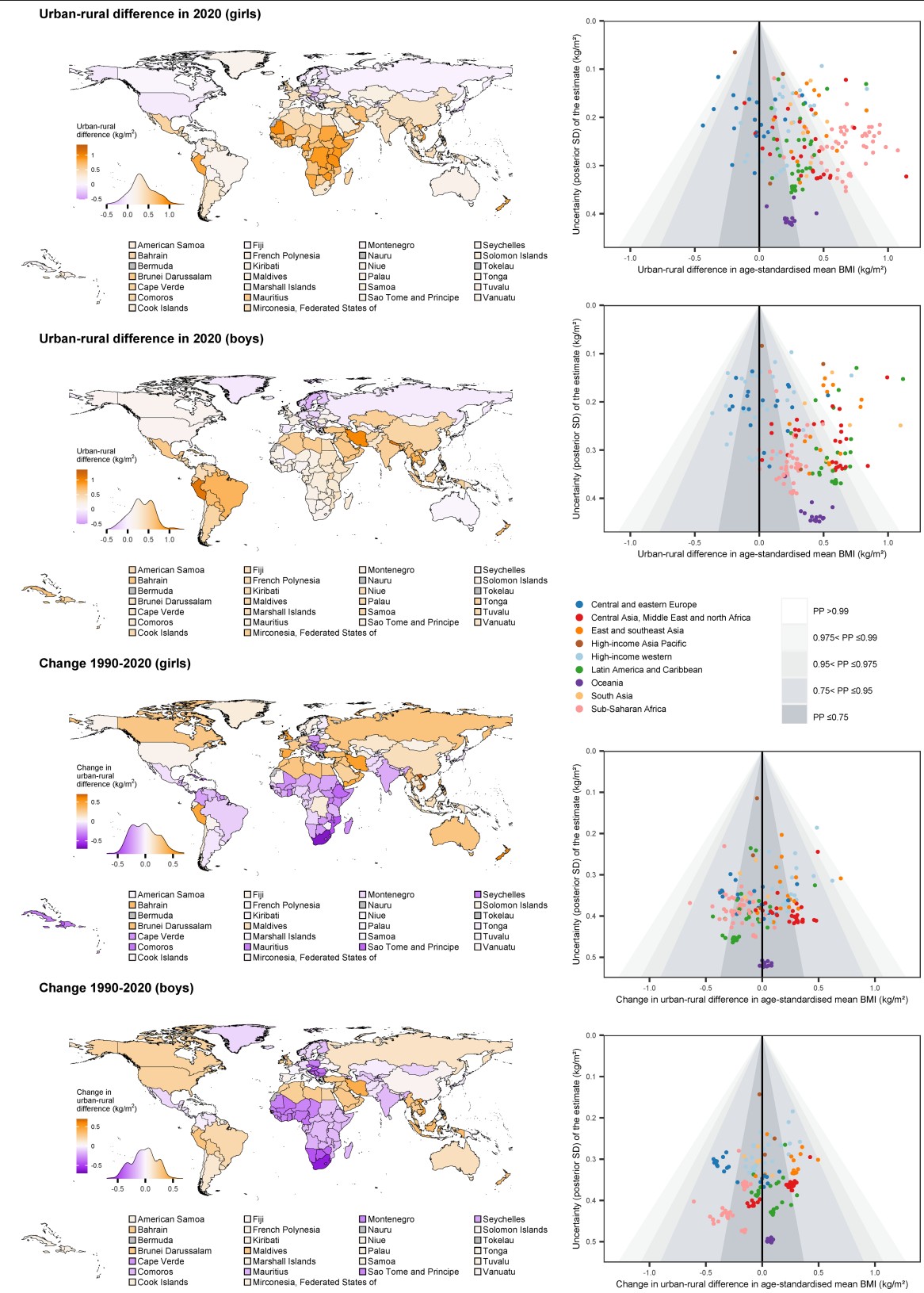

**Extended Data Fig. 3 | Urban-rural body-mass index (BMI) difference in 2020 and change from 1990 to 2020.** See Extended Data Fig. 2 caption for descriptions of the contents of the figure and for definitions. See Extended Data Fig. 9 for PP of the urban-rural difference in age-standardised mean BMI and its change. See Supplementary Fig. 8 for results at ages 5, 10, 15 and 19 years. We did not estimate the difference between rural and urban BMI for countries classified as entirely urban (Bermuda, Kuwait, Nauru and Singapore) or entirely rural (Tokelau), as indicated in grey.

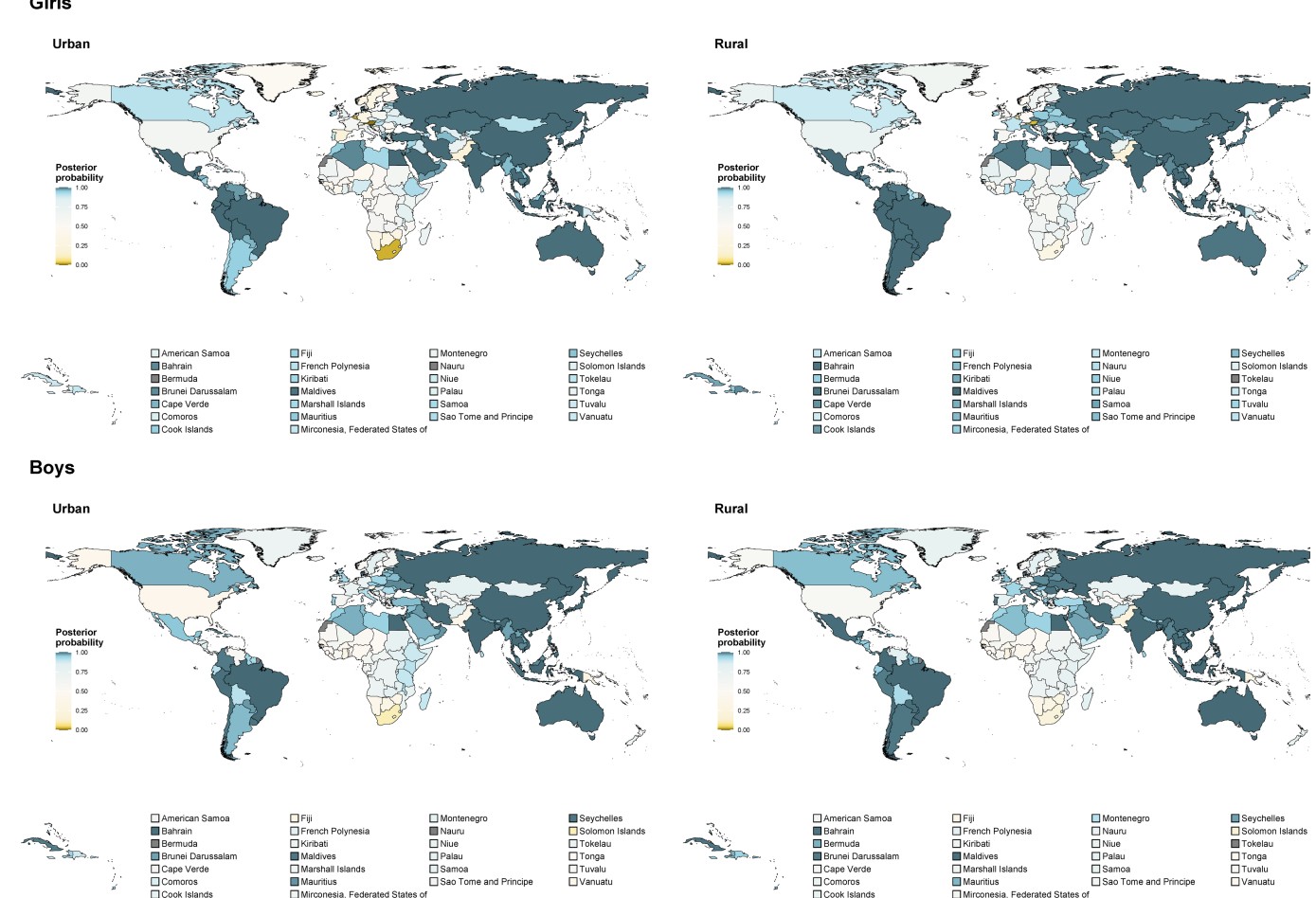

**Extended Data Fig. 4 | Posterior probability of increase in mean height in urban and rural areas from 1990 to 2020.** The maps show the PP that the age-standardised mean height increased from 1990 to 2020. The PP of a decrease is one minus that of an increase. If an increase in mean height is statistically indistinguishable from a decrease, the PP is 0.50. PPs closer to 0.50 indicate more uncertainty, those towards 1 indicate more certainty of an increase, and those towards 0 indicate more certainty of a decrease. We did not estimate PP for change in mean rural height for countries classified as entirely urban (Bermuda, Kuwait, Nauru and Singapore) or change in mean urban height for countries classified as entirely rural (Tokelau), as indicated in grey.

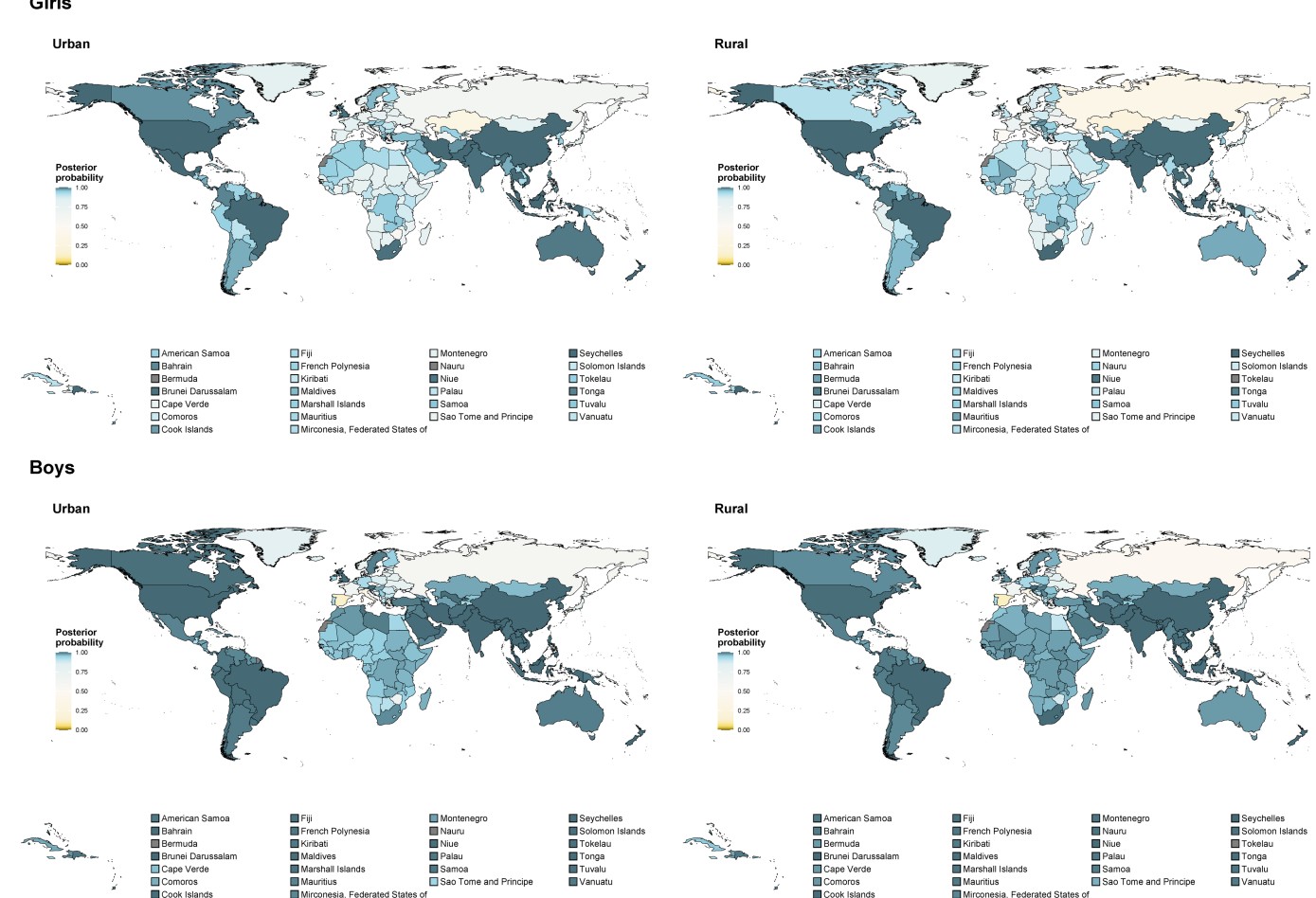

**Girls**

Urban

Rural

**Boys**

Urban

Rural

**Extended Data Fig. 5 | Posterior probability of increase in mean body-mass index (BMI) in urban and rural areas from 1990 to 2020.** The maps show the posterior probability (PP) that the age-standardised mean BMI increased from 1990 to 2020. The PP of a decrease is one minus that of an increase. We did not estimate PP for change in mean rural BMI in countries classified as entirely urban (Bermuda, Kuwait, Nauru and Singapore) or change in mean urban BMI in countries classified as entirely rural (Tokelau), as indicated in grey.

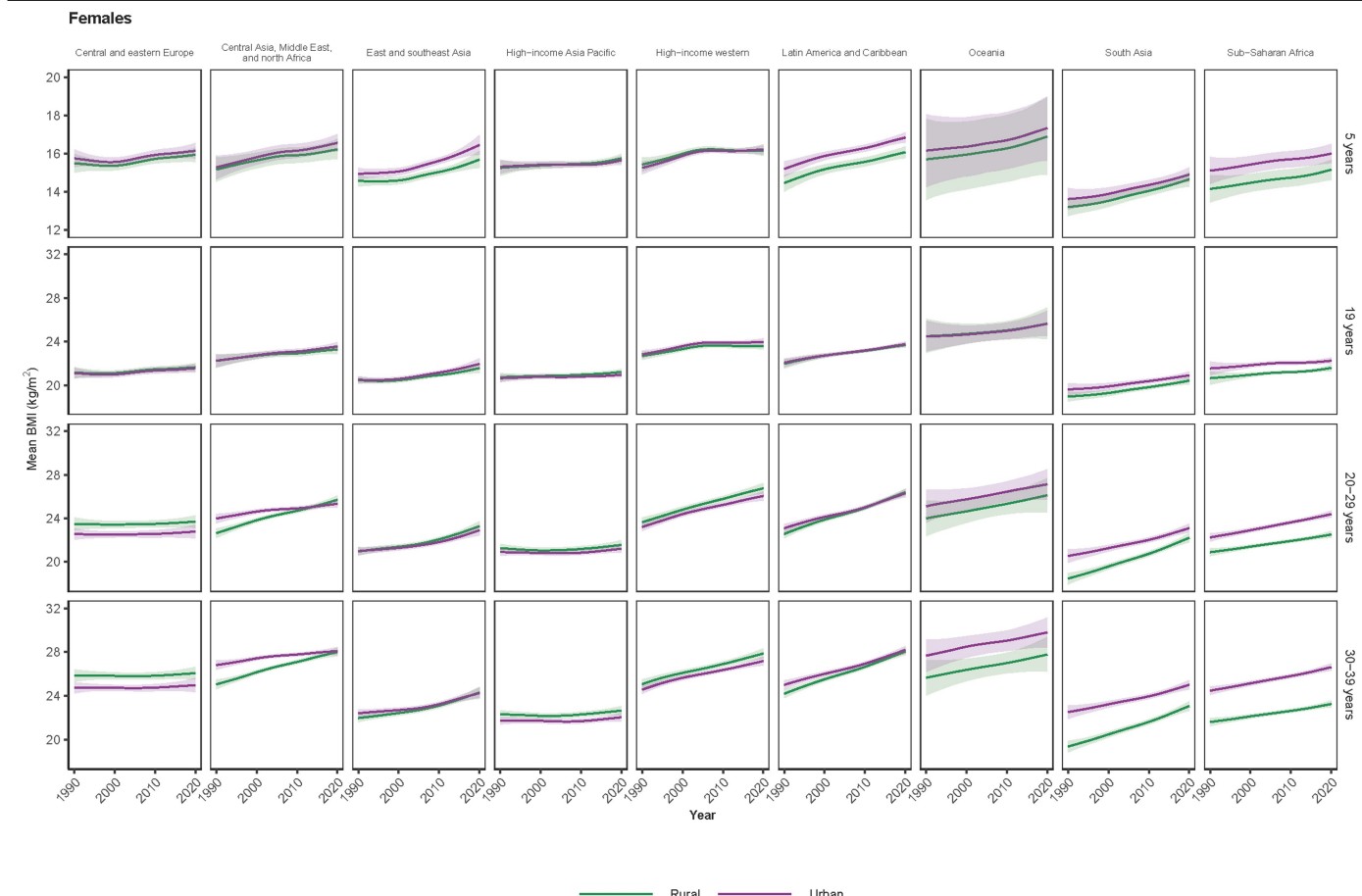

**Extended Data Fig. 6 | Trends in body-mass index (BMI) by place of residence for children, adolescents and young adults for females.** The figure shows trends in mean BMI at ages five and 19 years, and in age-standardised mean BMI for young adults (20–29 years and 30–39 years) for females. Shaded areas show the 95% CrIs. Trend for young adults were estimated using a model similar to the one described in Methods, where BMI-age patterns were allowed to vary flexibly via a cubic spline function without knots.

**Males**

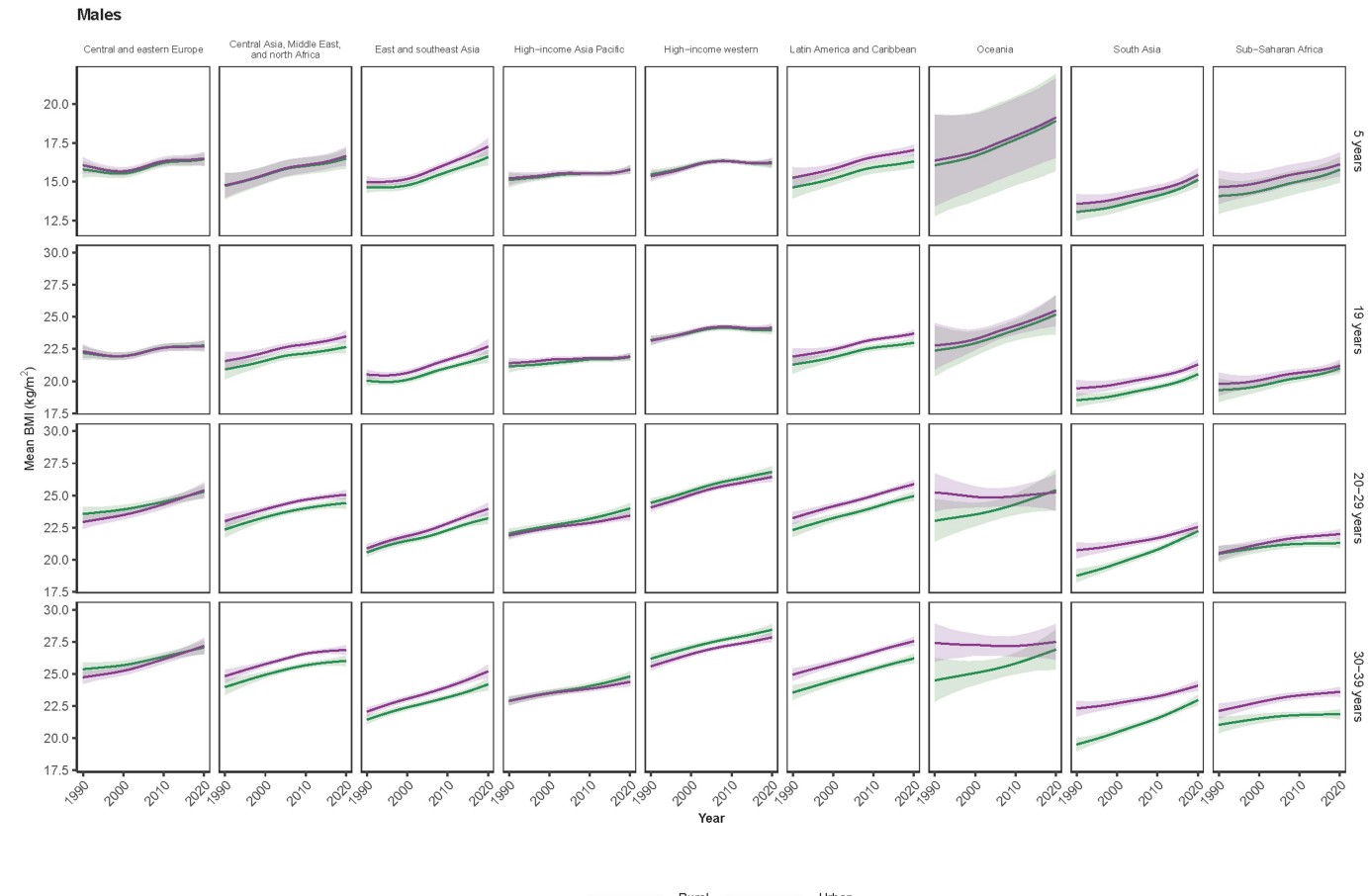

**Extended Data Fig. 7 | Trends in body-mass index (BMI) by place of residence for children, adolescents and young adults for males.** The figure shows trends in mean BMI at ages five and 19 years, and in age-standardised mean BMI for young adults (20–29 years and 30–39 years) for males. See Extended Data Fig. 6 caption for description of figure contents.

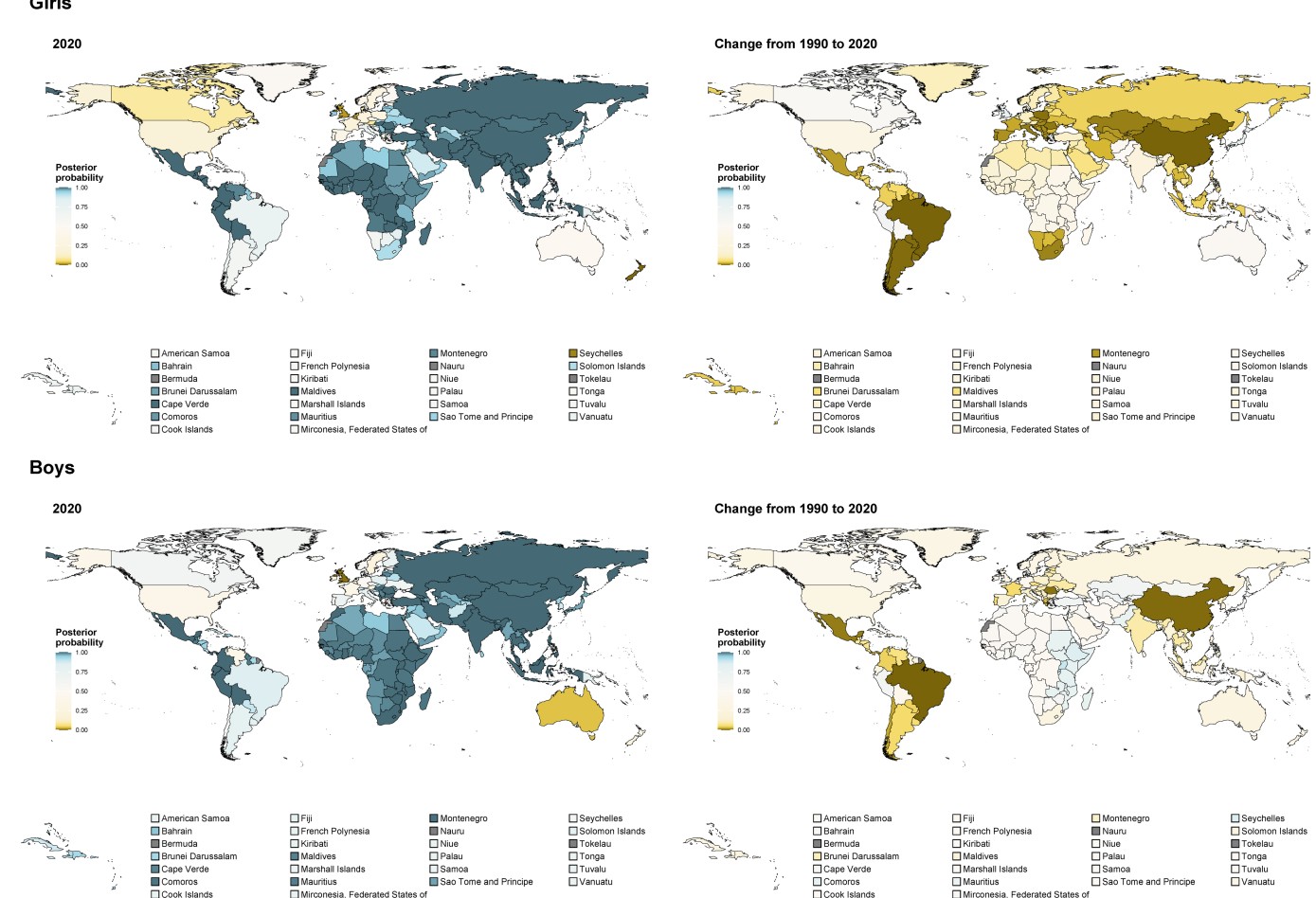

**Extended Data Fig. 8 | Posterior probability of urban-rural height difference in 2020 and its increase from 1990 to 2020.** The maps show the posterior probability (PP) that age-standardised mean height in 2020 in urban areas was higher than in rural areas (left-hand panels), and the PP that the urban-rural difference in age-standardised mean height increased from 1990 to 2020 (right-hand panels). For 2020, if estimated age-standardised mean urban height is statistically indistinguishable from rural height, the PP is 0.50. PPs closer to 0.50 indicate more uncertainty, those towards 1 indicate more certainty of urban children being taller, and those towards 0 indicate more certainty of rural being taller. For change, if an increase in urban-rural difference in mean height is statistically indistinguishable from a decrease, the PP is 0.50. PPs closer to 0.50 indicate more uncertainty, those towards 1 indicate more certainty of an increase in the urban-rural height difference, and those towards 0 indicate more certainty of a decrease. We did not estimate the PP for differences between rural and urban height for countries classified as entirely urban (Bermuda, Kuwait, Nauru and Singapore) or entirely rural (Tokelau), as indicated in grey.

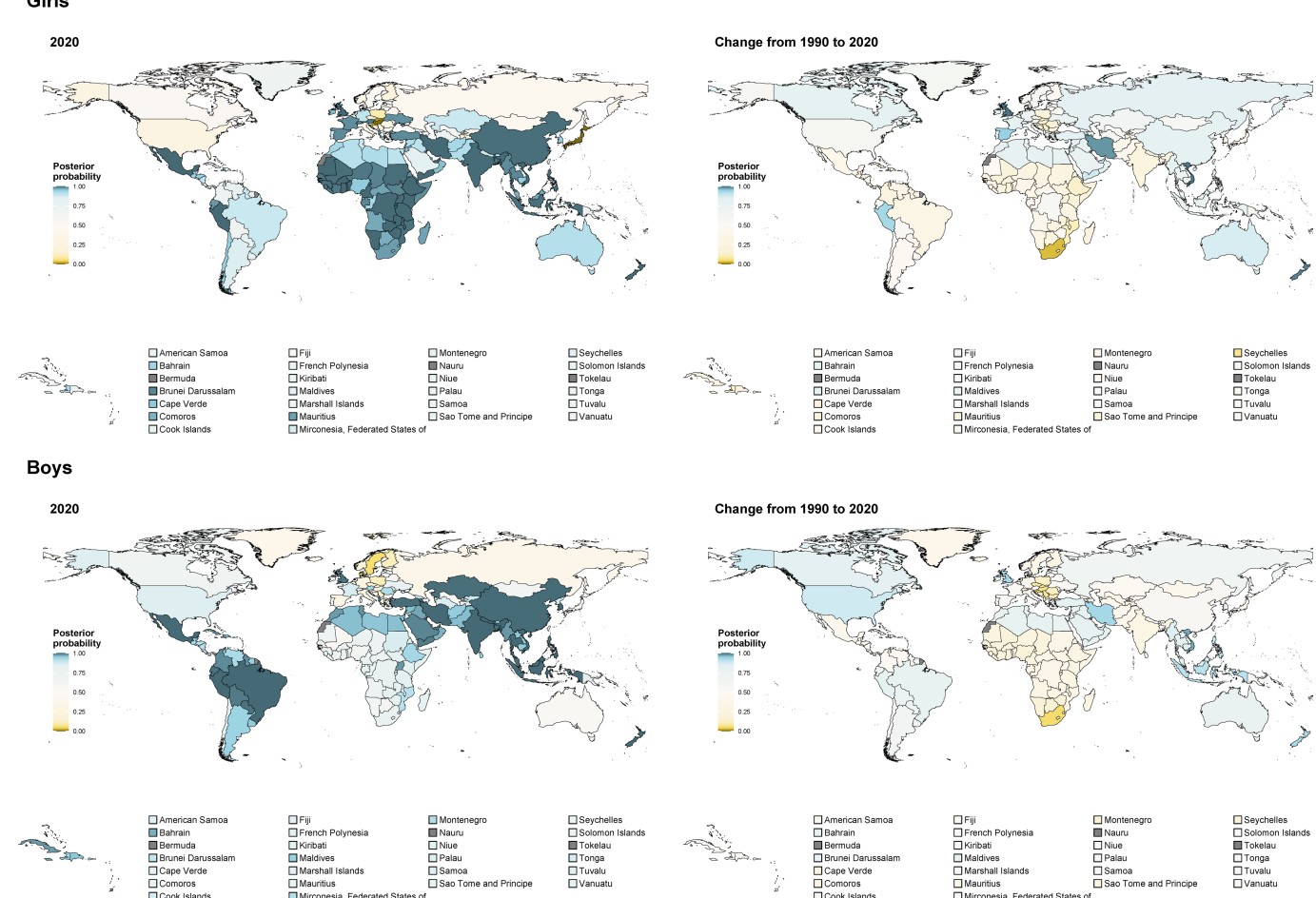

**Girls**

**2020**

Posterior probability
1.00
0.75
0.50
0.25
0.00

| | | | |
|---|---|---|---|
| American Samoa | Fiji | Montenegro | Seychelles |
| Bahrain | French Polynesia | Nauru | Solomon Islands |
| Bermuda | Kiribati | Niue | Tokelau |
| Brunei Darussalam | Maldives | Palau | Tonga |
| Cape Verde | Marshall Islands | Samoa | Tuvalu |
| Comoros | Mauritius | Sao Tome and Principe | Vanuatu |
| Cook Islands | Mirconesia, Federated States of | | |

**Change from 1990 to 2020**

Posterior probability
1.00
0.75
0.50
0.25
0.00

| | | | |
|---|---|---|---|
| American Samoa | Fiji | Montenegro | Seychelles |
| Bahrain | French Polynesia | Nauru | Solomon Islands |
| Bermuda | Kiribati | Niue | Tokelau |
| Brunei Darussalam | Maldives | Palau | Tonga |
| Cape Verde | Marshall Islands | Samoa | Tuvalu |
| Comoros | Mauritius | Sao Tome and Principe | Vanuatu |
| Cook Islands | Mirconesia, Federated States of | | |

**Boys**

**2020**

Posterior probability
1.00
0.75
0.50
0.25
0.00

| | | | |
|---|---|---|---|
| American Samoa | Fiji | Montenegro | Seychelles |
| Bahrain | French Polynesia | Nauru | Solomon Islands |
| Bermuda | Kiribati | Niue | Tokelau |
| Brunei Darussalam | Maldives | Palau | Tonga |
| Cape Verde | Marshall Islands | Samoa | Tuvalu |
| Comoros | Mauritius | Sao Tome and Principe | Vanuatu |
| Cook Islands | Mirconesia, Federated States of | | |

**Change from 1990 to 2020**

Posterior probability
1.00
0.75
0.50
0.25
0.00

| | | | |
|---|---|---|---|
| American Samoa | Fiji | Montenegro | Seychelles |
| Bahrain | French Polynesia | Nauru | Solomon Islands |
| Bermuda | Kiribati | Niue | Tokelau |
| Brunei Darussalam | Maldives | Palau | Tonga |
| Cape Verde | Marshall Islands | Samoa | Tuvalu |
| Comoros | Mauritius | Sao Tome and Principe | Vanuatu |
| Cook Islands | Mirconesia, Federated States of | | |

**Extended Data Fig. 9 | Posterior probability of urban-rural body-mass index (BMI) difference in 2020 and its increase from 1990 to 2020.** The maps show the posterior probability (PP) that age-standardised mean BMI in 2020 in urban areas was higher than in rural areas (left-hand panels), and the PP that the urban-rural difference in mean BMI increased from 1990 to 2020 (right-hand panels). We did not estimate the PP for differences between rural and urban BMI for countries classified as entirely urban (Bermuda, Kuwait, Nauru and Singapore) or entirely rural (Tokelau), as indicated in grey.

# Reporting Summary

## Statistics

For all statistical analyses, confirm that the following items are present in the figure legend, table legend, main text, or Methods section.

| n/a | Confirmed | |
|---|---|---|
| ☒ | ☐ | The exact sample size (*n*) for each experimental group/condition, given as a discrete number and unit of measurement |
| ☒ | ☐ | A statement on whether measurements were taken from distinct samples or whether the same sample was measured repeatedly |
| ☒ | ☐ | The statistical test(s) used AND whether they are one- or two-sided<br>*Only common tests should be described solely by name; describe more complex techniques in the Methods section.* |
| ☒ | ☐ | A description of all covariates tested |
| ☒ | ☐ | A description of any assumptions or corrections, such as tests of normality and adjustment for multiple comparisons |
| ☐ | ☒ | A full description of the statistical parameters including central tendency (e.g. means) or other basic estimates (e.g. regression coefficient) AND variation (e.g. standard deviation) or associated estimates of uncertainty (e.g. confidence intervals) |
| ☒ | ☐ | For null hypothesis testing, the test statistic (e.g. *F*, *t*, *r*) with confidence intervals, effect sizes, degrees of freedom and *P* value noted<br>*Give P values as exact values whenever suitable.* |
| ☐ | ☒ | For Bayesian analysis, information on the choice of priors and Markov chain Monte Carlo settings |
| ☒ | ☐ | For hierarchical and complex designs, identification of the appropriate level for tests and full reporting of outcomes |
| ☒ | ☐ | Estimates of effect sizes (e.g. Cohen's *d*, Pearson's *r*), indicating how they were calculated |

*Our web collection on statistics for biologists contains articles on many of the points above.*

## Software and code

Policy information about availability of computer code

| Data collection | Processing of secondary data was conducted using the statistical software R (version 4.1.2). We used R 'survey' package version 4.1-1. |
|---|---|
| Data analysis | All analyses were conducting using the statistical software R (version 4.1.2). The code for estimation of mean risk factor trends is available at www.ncdrisc.org. |

For manuscripts utilizing custom algorithms or software that are central to the research but not yet described in published literature, software must be made available to editors and reviewers. We strongly encourage code deposition in a community repository (e.g. GitHub). See the Nature Portfolio guidelines for submitting code & software for further information.

## Data

Policy information about availability of data

All manuscripts must include a data availability statement. This statement should provide the following information, where applicable:
- Accession codes, unique identifiers, or web links for publicly available datasets
- A description of any restrictions on data availability
- For clinical datasets or third party data, please ensure that the statement adheres to our policy

This is a data-pooling study that brings together 2,325 disparate data sources and uses a Bayesian hierarchical model to estimate population risk factor trends. Estimates of mean BMI and height by country, year, sex and place of residence (urban and rural) will be available from www.ncdrisc.org in machine-readable numerical format and as visualisations upon publication of the paper. Input data from publicly available sources can also be downloaded from www.ncdrisc.org and

# Human research participants

Policy information about studies involving human research participants and Sex and Gender in Research.

| | |
|---|---|
| Reporting on sex and gender | NA |
| Population characteristics | NA |
| Recruitment | NA |
| Ethics oversight | NA |

Note that full information on the approval of the study protocol must also be provided in the manuscript.

# Field-specific reporting

Please select the one below that is the best fit for your research. If you are not sure, read the appropriate sections before making your selection.

☐ Life sciences   ☒ Behavioural & social sciences   ☐ Ecological, evolutionary & environmental sciences

For a reference copy of the document with all sections, see nature.com/documents/nr-reporting-summary-flat.pdf

# Behavioural & social sciences study design

All studies must disclose on these points even when the disclosure is negative.

| | |
|---|---|
| Study description | We pooled and re-analysed population-based quantitative data that had measured height and weight for children and adolescents to estimate trends in mean BMI and height from 1990 to 2020 for 200 countries and territories, using a Bayesian hierarchical model. |
| Research sample | We used 2,325 population-based studies that had measured height and weight in 71 million participants in 194 countries. Studies were representative of a national, subnational or community population. We used all available and accessible data which met the criteria described below. |
| Sampling strategy | This is a data pooling study which used all available and accessible data. These are population-based studies, each with sample size set to detect measure of interest in that study. These were pooled in a meta regression which provides more confidence in results by borrowing strength across studies. We included data collected using a probabilistic sampling method with a defined sampling frame. We therefore included studies with simple random and complex survey designs but excluded convenience samples. |
| Data collection | We used 2,325 population-based studies that had measured height and weight in 71 million participants in 194 countries.We used data on measured height and weight to calculate mean BMI and height by sex and one-year age group. We excluded self-reported data. |
| Timing | For BMI, we pooled data collected from 1990 to 2020. For Height, we pooled data on those born from 1971 to 2015, after they had reached five years of age – i.e., data collected from 1976 to 2020.  For BMI, we included national studies for the 3 years prior to start year, assigning them to the start year, so that they can inform the estimates in countries with slightly earlier national data. We used all available data within these years which met the criteria described below. |
| Data exclusions | We excluded all data sources that were solely based on self-reported weight and height without a measurement component because these data are subject to biases that vary by geography, time, age, sex and socioeconomic characteristics. We also excluded data sources on population subgroups whose anthropometric status may differ systematically from the general population, including:<br>• studies that had included or excluded people based on their health status or cardiovascular risk;<br>• studies whose participants were only ethnic minorities;<br>• specific educational, occupational, or socioeconomic subgroups, with the exception noted below;<br>• those recruited through health facilities, with the exception noted below; and<br>• women aged 15-19 years in surveys which sampled only ever-married women or measured height and weight only among mothers.<br><br>We used school-based data in countries and age-sex groups with school enrolment of 70% or higher. We used data whose sampling frame was health insurance schemes in countries where at least 80% of the population were insured. Finally, we used data collected through general practice and primary care systems in high-income and central European countries with universal insurance, because contact with the primary care systems tends to be as good as or better than response rates for population-based surveys. |
| Non-participation | This was a secondary data analysis thus no participants were included in this study. |
| Randomization | Our study is an analysis of trends, and we did not carry out randomised experiments. |

# Reporting for specific materials, systems and methods

We require information from authors about some types of materials, experimental systems and methods used in many studies. Here, indicate whether each material, system or method listed is relevant to your study. If you are not sure if a list item applies to your research, read the appropriate section before selecting a response.

## Materials & experimental systems

| n/a | Involved in the study |
|---|---|
| ☒ | ☐ Antibodies |
| ☒ | ☐ Eukaryotic cell lines |
| ☒ | ☐ Palaeontology and archaeology |
| ☒ | ☐ Animals and other organisms |
| ☒ | ☐ Clinical data |
| ☒ | ☐ Dual use research of concern |

## Methods

| n/a | Involved in the study |
|---|---|
| ☒ | ☐ ChIP-seq |
| ☒ | ☐ Flow cytometry |
| ☒ | ☐ MRI-based neuroimaging |

