## [Peer Review File · Nature]

Manuscript Title: Diminishing benefits of urban living for children and adolescents' health

Reviewer Comments & Author Rebuttals

Reviewer Reports on the Initial Version:

Referee #1 (Remarks to the Author):

This collaboration investigating trends in childhood height and BMI, and differences between urban and rural areas around the world is excellent with the results indicating that in much of the world – except sub-Saharan Africa - the advantages of living in cities have diminished. What has been apparent in rich countries, such as the US, where health (e.g. obesity, hypertension) is worse in rural areas Rural Health Information Hub is similar to what was found in sub-Saharan Africa. Paradoxically, in the US, access to healthy foods in rural areas is limited. It would be interesting to add analyses looking at health care, diet or PA in urban and rural areas to understand what is driving the differences so that appropriate policies can be considered. It is difficult to make recommendations on policy without understanding exactly what is driving the differences.

Reviewing this manuscript would have been easier if a list of the data sources were included.

Some of the language in the paper related to growth trajectories suggests that longitudinal data would be the best source for the current analysis. Some of the data sources (e.g. electronic health records that appear to have been used can have multiple records on the same individuals). Were longitudinal data sources used? If not perhaps revise some of the language to not suggest a longitudinal analysis. If so, how was data cleaning handled related to change over time? EHR data is notoriously messy – just using the BIV cutoffs is not likely enough to clean the data. This reference might be useful:

Daymont C, Ross ME, Russell Localio A, Fiks AG, Wasserman RC, Grundmeier RW. Automated identification of implausible values in growth data from pediatric electronic health records. *J Am Med Inform Assoc.* 2017 Nov 1;24(6):1080-1087. doi: 10.1093/jamia/ocx037. Erratum in: *J Am Med Inform Assoc.* 2021 Dec 28;29(1):223. PMID: 28453637; PMCID: PMC7651915.

In high income countries rural areas are often less healthy than urban areas (in terms of diet, obesity, chronic disease). In the US much has been published on this. The intro of this paper doesn't distinguish between rich and poor countries and perhaps doesn't pick up the diversity of the differences between urban and rural areas around the world.

The conclusions about programs and policies go beyond the data that were analyzed since none of the reasons for differences were analyzed.

Were pregnant females excluded from BMI calculations?

Referee #2 (Remarks to the Author):

Thank you for the opportunity to review this paper. The authors bring together an unparalleled number of data sources of data on anthropometric measurements of children ages 5-19. They find that the urban height advantage has declined in most world regions and that this is due rural populations catching up to urban populations in mean age-standardized height. Additionally, the paper examines age-standardized mean BMI and finds generally small differences in BMI for urban versus rural children in this age range, with mean BMI generally increasing. The paper calls attention to regions in which height has stagnated in rural areas and yet urban-rural differences in BMI have narrowed. I believe the findings are novel. The methodology is transparent and rigorous. The findings are concisely and accurately presented, and the data visualizations provide a clear representation of the findings, although my preference would be to also present some numeric results in a set of supplemental tables, which, in some cases of comparison across metrics and regions, I would find easier to digest than the current figures. I think that presenting trends in this age group by rural-urban status for both height and BMI is of high importance and would be of wide interest to people in multiple disciplines, including my own. I include some specific questions about the conclusions below.

I have some minor specific questions and comments below:

Abstract

The abstract mentions as a key finding that in places where the urban height advantage increased, it was due to the fact heights stagnated or even decreased for boys in rural areas. While I find the figures useful, I think an additional supplemental table of number for each country and each region would be very useful. It would allow interested readers to see for themselves the height stagnation and baseline lower heights, more readily, in rural areas for these places. A large table that showed 1990 height, 2020 height and change for each region and each country within the region, by urban rural and by sex seems like it would be easier to digest/reference. Similarly for BMI.

Consider reframing the way the sentence in the Abstract, lines 24-26, so that it contains context for the root causes of this suboptimal growth. The current sentence could read as a bit of blaming the victim. The framing in lines 201-205 is helpful and perhaps a shorter version of these sentiments could be added to the abstract.

Line 62: Please confirm that the evidence about BMI leading to poor educational outcomes is likely not due to unobserved confounding factors, such as SES, i.e. please confirm the strength of the evidence base matches the causal connotation of this statement.

In the Abstract and the Discussion, rural Africa is singled out for having the most suboptimal growth trends, however, for both height and BMI, this region is not the only region or country with stagnating height or increasing BMI in rural areas. Is this the only region that experience this particular combination? Potentially the authors could say this more precisely in the Abstract so that it's clear why this one region is singled out.

Additionally, could the authors comment on how to interpret increasing BMI in the face of

stagnating height? Can it be determined that this is typically a suboptimal combination, even if the BMI level is not in the unhealthy range?

Line 126-128: How does the reader see this on the graph without knowing what level of age-standardized BMI would correspond with a BMI z-score in the unhealthy range. Is there a way to tell the reader what level of BMI is equal to underweight and overweight at different ages, or some other translation/interpretation that would help readers who are more familiar child BMI and viewing it as a z-score?

Line 135-136 and 157-158: Similar to comment above. If I look at BMI at age 19 in the Extended Data Figure 8, it appears to be well below 25, which is the adult threshold for unhealthy BMI.

Lines 169-170: In addition to programs that finance healthy development, potentially per capita income to adequate to prevent starvation combined with ready access to fortified foods and cheap, abundant calories could play a role?

Line 175-176: unclear of whether “emerging economies” is a synonym for low-income countries. If so, please check for consistency throughout.

Lines 176-178: This feels a little speculative and the references seem limited to a few countries. Can the authors clarify the degree to which this is speculation versus widely documented in the literature?

Line 181-182: Similarly, this statement seems a bit speculative. Have there been systematic evaluations to show that places that still have the urban height advantage tend to have more income inequality and gaps in the provision of programs between urban and rural populations.

Line 184: Seems like there should be more than one reference to support the assertion that it is widely assumed that cities would play a large role in obesity development in children.

Lines 202-210: In light of the information in this paragraph, the increase in BMI in sub-Saharan Africa seems unexpected. Can the authors comment more on the conditions that might be leading to increasing BMI without increasing height in this age group?

I find myself wondering about the role of survival from 0-5 and the role that differential changes in survival by urban status might play in these trends. In particular, I wonder if differential trends in survival into the 5th year of life might play a role in increasing BMI in rural areas?

Additionally, based on Extended Data Figure 1, it looks as though sub-Saharan Africa has the fewest surveys and yet some of the most notable trends in the results from this paper. Can the authors comment on the whether the conclusion from the sub-Saharan African countries should be accompanied by any caveats regarding the number of surveys or observations from this region?

Detailed methods

Line 565: The methods refer to receiving anonymized individual data from each of the studies. Does this mean that smaller studies just contributed fewer observations and therefore naturally 'count less' in the estimates (rather than some kind of weighting scheme)?

Line 575: what percentage of surveys were missing place of residence? seems important for this paper.

Supplemental Table 1. As a reader and reviewer, I would prefer to see this Table organized by region first and then alphabetically within region. This would help a reader see how much information is coming from each region.

Referee #3 (Remarks to the Author):

The authors estimated levels and trends in mean height and BMI of children aged 5 to 19 years old in rural and urban areas of 200 countries and territories from 1990 to 2020.

In the data-processing part, line 566-567, the authors mentioned that they "calculated mean height and mean BMI, and the associated standard errors". Can the authors add more information regarding how they computed these SEs? If the sampling designs from various data sources differ with each other, different approach should be used. E.g. the DHS employs multiple (usually two)-stage stratified sampling design. Hence, the usual approach to compute SE given such sampling design is the jackknife method. The authors could provide a table summarizing the data sources by sampling design and the approaches used to compute the SE.

About data availability, it would be nice to list how many data points are available for female and male over time in each country and by residence. Sometimes, the female and male questionnaires have different focuses and qualities. Sometimes the male questionnaire may not be available for a certain country-year whereas the female one is available.

About the model, the authors mentioned on line 674-675 that "all analyses were done separately by sex because age, geographical and temporal patterns of height or BMI differ between girls and boys". When using the term "separately", I assume it actually means "independently". I do agree that the levels of BMI between females and males are most likely different. However, this cannot be used as the reason to treat the female and male BMI independently. If the data availability are different between females and males, modeling the two sexes together can actually help to inform at least the pattern of the sex with lower data availability. But of course one needs to first assume the patterns (not levels) of height and BMI between females and males are similar over time.

About the MCMC computing, line 675-680: could the authors add the information on how many parallel chains they used for the MCMC algorithm? In addition, since the posterior samples are based on MCMC, the authors need to explain how did they check the model convergence. If the

results are based on non-converged parameters, the results are not reliable.

Referee #4 (Remarks to the Author):

Summary of the key results

- This paper brings together a huge number of population based surveys to estimate changes in height and BMI over the period from 1990 to 2020. Despite the impressive achievement of pulling thousands of surveys together, I'm struggling to find a way to adequately summarize the results, since there seems to be a large degree of heterogeneity in the findings. Aside from "the urban height advantage got smaller and urban-rural differences in BMI remained generally small", which is fine as it goes, but there are important caveats to the pattern that make it very hard to argue that the paper provides much basis for action, despite its value for benchmarking and assessing trends.

Originality and significance: if not novel, please include reference

- The data assembled are original in the sense that the NCD consortium has undertaken an enormous and labour-intensive effort to assemble a huge number of population-based surveys, but this was not done specifically for this paper. They have previously published a similar paper (Nature 2019; 569(7755): 260–264.) on secular trends in BMI by rural and urban status, so it is a little unclear to me how much is added by now doing a separate paper that also includes height. In some respects I think the paper would be easier to read and understand if it focused only on height, since a lot of the BMI results (and message) are similar to the prior paper.

- The significance is difficult to characterize. On the one hand, it is important to have as up to date and current descriptions of population trends in factors such as height and BMI cross-classified by gender. On the other hand, I don't find the evidence in the paper is sufficient to back up the significant policy action the authors suggest in much of the discussion.

Data & methodology: validity of approach, quality of data, quality of presentation

- In general, the methods are of high quality, and mostly transparently reported. I did not find the graphs or maps to be especially compelling (the panel figures are better), but then again it is quite hard to show changes over time in a spatially compelling way.

- p21. Happy to see the authors provide a link to the code for models for transforming the categorical BMI studies to a mean, but I do not see a more general link to where the entire pipeline for reproducing the tables and figures in the paper is available. I see a note that the details for the Bayesian model will be posted, but will this be enough for a third party to reproduce exactly (MCMC error aside) the estimates in the paper?

- p23. "data driven fixed effects" is confusing and readers will have a hard time knowing what "data driven" means in this context. Plus, it isn't clear whether "fixed-effect and random-effect terms for subnational and community data" means this was done with respect to whether the sample was subnational or community. What is the difference? Is it just saying that any subnational estimate was treated as a separate stratum with a fixed effect, i.e., sub-national = 1 for any subnational estimate and 0 otherwise? But what then were the random effects? I had trouble following this.

- I didn't see any discussion of the priors used for the Bayesian models. Were they flat?

Uninformative? Informative based on substantive knowledge? For this volume of data it's clear that

the data will likely overwhelm the prior, but as a matter of good practice it would seem important to at least make this clear to readers, especially those familiar with Bayesian inference. In fact, one of the Nature journals has recently published a review of good practices for reporting Bayesian analysis (Kruschke, Nat Hum Behav 2021) but not much of that guidance has been followed here. Some aspects might be hard for this paper, but at least reporting on priors and computational performance (convergence, chain mixing, etc.) should be straightforward.

- p24. Why are the authors age-standardizing these estimates? If part of what the authors want to do with the paper is to assess how the patterns of height and BMI have changed over time, why resort to a fictitious metric that generates counterfactual, rather than actual heights and BMIs over time? They may wish to argue for standardizing, but given that the discussion contains a number of suggestions for policy action, any policies are likely to have to consider the actual populations at risk rather than what the urban-rural height difference would be *if* rural and urban areas had age distributions identical to the WHO standard. Given the narrow age range it may not make any difference, but it would seem at least relevant to assess the impact of age-standardizing on the pattern and changes over time.

Appropriate use of statistics and treatment of uncertainties

- In general I found the main report did not emphasize reporting the magnitude and uncertainty of the main objectives (i.e., heights, BMIs, urban-rural differences and changes over time). Most of the results were framed as "the largest height differences were in countries X, Y, Z" but did not report anything regarding the magnitude and uncertainty of those estimates. Instead, the authors seem to focus a lot of attention on what one could argue is a more binary kind of outcome: e.g., the probability of whether the change in height between 1990 and 2020 was larger in rural vs. urban areas. It isn't clear to me how this helps readers get a sense of the public health importance of these differences. Don't we want to know what those differences are? I think providing estimates of uncertainty would be particularly helpful given the lack of data for sub-Saharan Africa and the role that this region plays in the results.

- More generally, I don't understand why the authors are reporting on the posterior probability that, for example, the urban-rural difference in height (or BMI) is positive, rather than giving readers an estimate and credible interval for the magnitude of the difference. I understand that using the PP allows them to generate maps that provide a lens on that question, but this seems to me more a limitation of relying on maps for evidence communication. It would be helpful instead to have a table or figure for each country that shows the magnitude and precision of the estimate, not just the PP of whether or not the differences is positive (or greater in one group than another). Given the already long appendix, I don't think it would be too harmful to extend it a little further.

Conclusions: robustness, validity, reliability

- In general I found that a number of the conclusions went beyond (sometimes considerably beyond) the data and evidence presented in the paper. For example, even in the summary the authors say that, based on their report, there is an "urgent need" for policies to reduce income/infrastructure inequalities across urban/rural regions, yet they have not provided any evidence in the paper that the changes they observe are due to differences in income/infrastructure. I think the same goes for nutrition/sport programs and health services. To be clear, I tend to agree with the authors that such policies are likely to be beneficial, but I think this empirical paper does not provide sufficient

evidence to back those claims. What to do about these changes in heights and BMIs feels like a different paper than this one describing trends.

- This concern is echoed as much of the introduction seems to rest on strong assumptions about the importance of relevant global patterns for local interventions. What evidence is there that "attention" to the heterogeneity of the urban-rural gradient across countries is useful or necessary for policy and programme development? Policies and programs for whom? Globally? At the country or municipality level? If there is compelling evidence that such large scale measurement and surveys are effective for designing policy at the country or subnational level, it would be good to see that argument made.

- I also think that, though it is mentioned in the limitations section, the authors should probably address the issue of migration with greater care. What is the likely role that rural-to-urban migration may play in these trends. If particular kinds of populations are migrating from rural-to-urban areas (e.g., smaller rural children accompanying parents migrating to urban areas), couldn't this meaningfully contribute to the change in the urban-rural difference? Given the pace of urbanization of the past few decades, this feels like it needs more attention in the paper as a plausible explanation for some of the results. Or, if there is evidence to refute migration as an explanation, it should be discussed.

Suggested improvements: experiments, data for possible revision

- Reporting on the magnitude and uncertainty of estimates rather than the PP of differences would improve the paper.

- Authors should drop the specific appeals for policy interventions without much better evidence as to the causes of the patterns evident in the paper.

- More discussion of migration or alternative explanations for the pattern observed.

References: appropriate credit to previous work?

- Yes

Clarity and context: lucidity of abstract/summary, appropriateness of abstract, introduction and conclusions

- Yes

Author Rebuttals to Initial Comments:

We thank the referees for their thoughtful comments and suggestions. We have used these to revise the paper, as detailed below. All page, line, figure, table and citation numbers refer to the tracked manuscript.

Referee #1

This collaboration investigating trends in childhood height and BMI, and differences between urban and rural areas around the world is excellent with the results indicating that in much of the world – except sub-Saharan Africa - the advantages of living in cities have diminished. What has been apparent in rich countries, such as the US, where health (e.g. obesity, hypertension) is worse in rural areas Rural Health Information Hub is similar to what was found in sub-Saharan Africa. Paradoxically, in the US, access to healthy foods in rural areas is limited. It would be interesting to add analyses looking at health care, diet or PA in urban and rural areas to understand what is driving the differences so that appropriate policies can be considered. It is difficult to make recommendations on policy without understanding exactly what is driving the differences.

We agree that understanding the contributions of food, energy expenditure, and healthcare to the differential trends in height and body-mass index (BMI) in rural and urban areas would provide important insights. The tasks of identifying suitable drivers/determinants, gathering data, and inferring their roles would form a distinct long-term research programme for at least two reasons: First, even without disaggregation to rural and urban place of residence, worldwide data on food, physical activity and healthcare are less reliable and more sparse than data on BMI, and largely absent for important aspects of diet such as processed carbohydrates. For example, our work on consumption of sugary-sweetened beverages and fruit juices at the national level¹ could identify only 193 data sources, i.e. less than one tenth of what we used here, and could only generate estimates for a single year; the number of data sources for animal-based foods² was less than a quarter of those used here, which did not allow modelling trends for many countries. Similarly, work by some of the authors of our paper on physical activity³ could only present estimates at two points in time due to limited data availability. None of these estimates were disaggregated by place of residence. Global data on quality of care are largely limited to antenatal care⁴, immunisation⁵ or school-based service provision⁶, with more complete measures available only for a few specific countries⁷. Second, even prospective cohort studies, with intensive measurement on a small group of individuals, have been unable to unambiguously and consistently decouple the role of food and physical activity/energy expenditure on BMI (and similarly for food and healthcare on height) both because difficulties in measuring diet, physical activity and quality of care⁸⁻¹², and because changes in physical activity influence food intake and vice versa¹³⁻²⁰.

Reviewing this manuscript would have been easier if a list of the data sources were included.

The complete list of data sources used in the analysis, and their characteristics, is provided in Supplementary Table 2.

Some of the language in the paper related to growth trajectories suggests that longitudinal data would be the best source for the current analysis. Some of the data sources (e.g. electronic health records that appear to have been used can have multiple records on the same individuals). Were longitudinal data sources used? If not perhaps revise some of the language to not suggest a longitudinal analysis.

Our data are longitudinal in the sense that they contain repeated observations over time for children of specific ages (e.g., 19-year-olds in different years), but we do not follow up individual participants because our aim was to estimate change over time in entire populations. Our results show changes over time in height and BMI, by age or age-standardised and by place of residence, for entire national populations. We have stated this design in the main paper (P. 5, Lines 88-93) and Methods (P. 21, Lines 656-659) and used language, e.g., “successive cohorts”, that indicates this population focus throughout the paper.

If so, how was data cleaning handled related to change over time? EHR data is notoriously messy – just using the BIV cutoffs is not likely enough to clean the data. This reference might be useful:

Daymont C, Ross ME, Russell Localio A, Fiks AG, Wasserman RC, Grundmeier RW. Automated identification of implausible values in growth data from pediatric electronic health records. J Am Med Inform Assoc. 2017 Nov 1;24(6):1080-1087. doi: 10.1093/jamia/ocx037. Erratum in: J Am Med Inform Assoc. 2021 Dec 28;29(1):223. PMID: 28453637; PMCID: PMC7651915.

Nearly all of our data are from population-representative health and nutrition surveys and epidemiological studies with well-established measurement protocols and procedures (see Supplementary Table 2) and not from electronic health records. The procedure described in the Daymont²¹ paper is designed to clean electronic health records containing repeated measurements in the same individual as they get older, by fitting a curve to their sequential measurements (converted to z-scores) and identifying records with unusually large deviation from a fitted moving average of individual measurements. Its aim of detecting periods of implausible growth velocity for an individual is different from the data used here, and from the underlying question of change in height/BMI across successive cohorts of children and adolescents in the same country.

In high income countries rural areas are often less healthy than urban areas (in terms of diet, obesity, chronic disease). In the US much has been published on this. The intro of this paper doesn't distinguish between rich and poor countries and perhaps doesn't pick up the diversity of the differences between urban and rural areas around the world.

As the Referee correctly points out, and as stated in discussion of our results (P. 11, Lines 257-260), some works, especially in the USA, have pointed out worse rural outcomes in BMI²² (but not the convergence of height) and some of its determinants. Despite these isolated studies, a substantial body of literature, both in global health and European public health, continues to present a negative perspective on cities²³⁻³¹. We have nonetheless modified the introductory part of the paper to acknowledge the important work on poor rural nutrition, especially in the USA (P. 3, Lines 48-49).

The conclusions about programs and policies go beyond the data that were analyzed since none of the reasons for differences were analyzed.

Our original policy recommendations had drawn on our results as well as on broader literature in countries that we found to have rural and urban convergence versus divergence. We have moderated the extent of recommendations, and have been explicit when we used broader knowledge on policy impacts (P.10-11, Lines 235-253; P. 13, Lines 301-314). We hope this is an appropriate balance between being within the scope of our work – which is benchmarking country outcomes – and having a paper that goes beyond presentation of quantitative results. We would of course be happy to further adjust the discussion based on recommendations by the Referee and Editors.

Were pregnant females excluded from BMI calculations?

Yes, pregnant females were excluded (P. 25, Lines 756-757), noting that being pregnant at the time of measurement is less common in the age ranges in our analysis than in data on adults.

Referee #2

Thank you for the opportunity to review this paper. The authors bring together an unparalleled number of data sources of data on anthropometric measurements of children ages 5-19. They find that the urban height advantage has declined in most world regions and that this is due rural populations catching up to urban populations in mean age-standardized height. Additionally, the paper examines age-standardized mean BMI and finds generally small differences in BMI for urban versus rural children in this age range, with mean BMI generally increasing. The paper calls attention to regions in which height has stagnated in rural areas and yet urban-rural differences in BMI have narrowed. I believe the findings are novel. The methodology is transparent and rigorous. The findings are concisely and accurately presented, and the data visualizations provide a clear representation of the findings, although my preference would be to also present some numeric results in a set of supplemental tables, which, in some cases of comparison across metrics and regions, I would find easier to digest than the current figures. I think that presenting trends in this age group by rural-urban status for both height and BMI is of high importance and would be of wide interest to people in multiple disciplines, including my own. I include some specific questions about the conclusions below.

As detailed below, we have implemented these good suggestions that would help make the results more accessible.

I have some minor specific questions and comments below:

Abstract

The abstract mentions as a key finding that in places where the urban height advantage increased, it was due to the fact heights stagnated or even decreased

for boys in rural areas. While I find the figures useful, I think an additional supplemental table of number for each country and each region would be very useful. It would allow interested readers to see for themselves the height stagnation and baseline lower heights, more readily, in rural areas for these places. A large table that showed 1990 height, 2020 height and change for each region and each country within the region, by urban rural and by sex seems like it would be easier to digest/reference. Similarly for BMI.

We have included such tables (Supplementary Tables 3 and 4) so that the numerical results are available with the paper itself. We have also stated that all country results can be visualised and downloaded in a machine-readable format from NCD-RisC website (P. 39, Lines 1096-1106).

Consider reframing the way the sentence in the Abstract, lines 24-26, so that it contains context for the root causes of this suboptimal growth. The current sentence could read as a bit of blaming the victim. The framing in lines 201-205 is helpful and perhaps a shorter version of these sentiments could be added to the abstract.

We have done as thoughtfully suggested (P. 2-3, Lines 28-30).

Line 62

Please confirm that the evidence about BMI leading to poor educational outcomes is likely not due to unobserved confounding factors, such as SES, i.e. please confirm the strength of the evidence base matches the causal connotation of this statement.

We have modified this section to state that these results are based on observational data, which in the strictest sense mean that the effect “might” be causal (P. 4, Lines 76-81).

In the Abstract and the Discussion, rural Africa is singled out for having the most suboptimal growth trends, however, for both height and BMI, this region is not the only region or country with stagnating height or increasing BMI in rural areas. Is this the only region that experience this particular combination? Potentially the authors could say this more precisely in the Abstract so that it's clear why this one region in singled out.

As appropriately noted by the Referee, we have revised the abstract to state this combination occurred in most countries in sub-Saharan Africa and in some countries in other specific regions (P. 2, Lines 15-17).

Additionally, could the authors comment on how to interpret increasing BMI in the face of stagnating height? Can it be determined that this is typically a suboptimal combination, even if the BMI level is not in the unhealthy range?

To our knowledge, except for the effects of stunting and wasting in under-five children³², epidemiological studies have largely analysed the effects of height and BMI on non-communicable diseases separately³³⁻³⁷ and have rarely considered whether the aetiological effects of one depends on the other. With this in mind, the harmful

effects of having excessively low BMI have been replicated in studies in different population, which have different average attained adult heights³⁵⁻³⁷. Therefore we would expect that gaining weight within underweight ranges is beneficial, even if height remains lower than optimal. In practice, however, in our results weight gain in most sub-Saharan countries in Africa went beyond remedying underweight and moved the population mean above the median of the WHO reference population, i.e., towards, and in some cases approaching, the unhealthy overweight range. We have nonetheless mentioned that for rural populations, rise in BMI mostly “shifted the mean BMI of rural boys and girls out of the range for being underweight” (P.8, Lines 174-177; P. 9, Lines 186-191).

Line 126-128: How does the reader see this on the graph without knowing what level of age-standardized BMI would correspond with a BMI z-score in the unhealthy range. Is there a way to tell the reader what level of BMI is equal to underweight and overweight at different ages, or some other translation/interpretation that would help readers who are more familiar child BMI and viewing it as a z-score?

This is a good point, related to the preceding comment. As the Referee would be aware, the median and underweight/overweight cut-offs are age-specific, whereas Figure 1 summarises 15 age-specific values (from 5 to 19 years of age) through age standardisation. We have shown the reference median and ± 1 SD thresholds in the new age-specific figures (Extended Data Figure 4).

Line 135-136 and 157-158: Similar to comment above. If I look at BMI at age 19 in the Extended Data Figure 8, it appears to be well below 25, which is the adult threshold for unhealthy BMI.

BMI of 25 kg/m² is the threshold for clinical overweight, but the harmful effects of BMI start from values as low as ~21 kg/m²^{35,36}. We have clarified this issue as stated in the two preceding two comments.

Lines 169-170: In addition to programs that finance healthy development, potentially per capita income to adequate to prevent starvation combined with ready access to fortified foods and cheap, abundant calories could play a role?

We have added these to the potential reasons for small urban-rural height differential in high-income countries (P. 10-11, Lines 235-242).

Line 175-176: unclear of whether “emerging economies” is a synonym for low-income countries. If so, please check for consistency throughout.

We used “emerging economies” for countries such as Argentina, Chile and Taiwan, which have transitioned from low- or middle-income towards upper-middle- or high-income industrialised status. We have stated this intended meaning (P. 7, Lines 151-152) and would be happy to use alternative vocabulary if needed.

Lines 176-178: This feels a little speculative and the references seem limited to a few countries. Can the authors clarify the degree to which this is speculation versus widely documented in the literature?

To our knowledge, the role of income redistribution and improved nutrition in rural/poor populations has largely been studied in case studies, of the sort we have cited. The paper by Smith et al³⁸, which we have cited, is the closest to a formal quantitative analysis and concludes that a series of socioeconomic factors are responsible for the urban-rural differential for under-five children. This would then imply that redistribution should reduce the differential, as also seen in evaluation of cash transfer studies³⁹, albeit without stratification on place of residence. We have toned down the sentence (P. 11, Lines 244-248) while also noting that the use of case studies is a common approach to illustrating policy impacts, e.g., for tobacco taxes and universal insurance.

Line 181-182: Similarly, this statement seems a bit speculative. Have there been systematic evaluations to show that places that still have the urban height advantage tend to have more income inequality and gaps in the provision of programs between urban and rural populations.

As above, to our knowledge this has only been looked at in the context of case studies, although the aforementioned paper by Smith et al.³⁸ goes some way towards a formal quantitative analysis. As above, we have toned down the sentence (P. 11, Lines 251-253).

Line 184: Seems like there should be more than one reference to support the assertion that it is widely assumed that cities would play a large role in obesity development in children.

We had based our statement on an article²⁴ that had reviewed and synthesised multiple other studies to support our assertion. We have now added other citations (P. 11, Line 257).

Lines 202-210: In light of the information in this paragraph, the increase in BMI in sub-Saharan Africa seems unexpected. Can the authors comment more on the conditions that might be leading to increasing BMI without increasing height in this age group?

We have specified that at least two factors may have contributed to this trend (P. 12, Lines 273-287). The more likely explanation is an increase in quantity of calories available without improvements in their quality that could result in more optimal height gain, e.g., animal-source foods⁴⁰⁻⁴⁵. The second explanation, although one with more uncertainty, would be that the age of puberty onset, which is influenced by weight gain during childhood, might affect height gain during the adolescent growth spurt and in late adolescence^{46,47}.

I find myself wondering about the role of survival from 0-5 and the role that differential changes in survival by urban status might play in these trends. In particular, I wonder if differential trends in survival into the 5th year of life might play a role in increasing BMI in rural areas?

We have done a review of studies in Africa that have stratified trends in survival by place of residence and found that some studies indicate faster improvement in rural areas while others have found the opposite^{48,49}. We have nonetheless stated briefly

that differential trends in survival beyond five years of age may affect urban-rural differences in height and BMI in school ages (P. 12, Lines 283-287).

Additionally, based on Extended Data Figure 1, it looks as though sub-Saharan Africa has the fewest surveys and yet some of the most notable trends in the results from this paper. Can the authors comment on the whether the conclusion from the sub-Saharan African countries should be accompanied by any caveats regarding the number of surveys or observations from this region?

We used a total of 276 data sources from sub-Saharan Africa, or 5.6 sources per country. This is more data than what analyses of other conditions have used for the entire world and, based on our work on a range of risk factors, should allow to have robust estimates. The stagnation and decline were also apparent, and perhaps particularly so, in countries with good data such as Ethiopia, Rwanda and Uganda. We have nonetheless stated the regional differential in data availability as a limitation of current data systems (P. 38, Lines 1071-1076).

Detailed methods

Line 565: The methods refer to receiving anonymized individual data from each of the studies. Does this mean that smaller studies just contributed fewer observations and therefore naturally 'count less' in the estimates (rather than some kind of weighting scheme)?

Smaller studies contribute less to estimates but not proportional to their sample size. As stated in Methods the SE of the mean depends on sample size, variability of the data and survey design (e.g., simple random sample versus clustered and stratified design) (P. 23, Lines 705-711). Further, two features of the Bayesian statistical model capture the fact that subnational and community studies are more variable than national ones, and facilitate larger influence from national studies, than from subnational and community studies, for any sample size. The first feature is the study-specific random-effect term (referred to as e_i in Methods [P.33, Lines 958-968]), which captures variation from the population average, for an entire study. These terms are estimated for national, sub-national and community level studies with the constraint that national studies have less variability than sub-national studies and community studies. The second feature is that the variance of the likelihood of the Bayesian model is made up of the sum of two terms: the first term is the (square of the) SE of the data point; the second term (referred to as τ^2 in Methods) captures additional variability among different age groups within each study (P.35, Lines 1007-1012). This term is estimated for national, sub-national and community level studies, again with the constraint that there is less residual variability in national studies than in sub-national and community-level studies. As a result of these two terms, with the same standard error, national studies have a larger influence on the results than non-national ones.

Line 575: what percentage of surveys were missing place of residence? seems important for this paper.

We have reported this information in the text (P. 28, Lines 836-837) and in the newly added Supplementary Figures 1 and 2.

Supplemental Table 1. As a reader and reviewer, I would prefer to see this Table organized by region first and then alphabetically within region. This would help a reader see how much information is coming from each region.

We have re-organised the table as suggested.

Referee #3

The authors estimated levels and trends in mean height and BMI of children aged 5 to 19 years old in rural and urban areas of 200 countries and territories from 1990 to 2020.

In the data-processing part, line 566-567, the authors mentioned that they "calculated mean height and mean BMI, and the associated standard errors". Can the authors add more information regarding how they computed these SEs? If the sampling designs from various data sources differ with each other, different approach should be used. E.g. the DHS employs multiple (usually two)-stage stratified sampling design. Hence, the usual approach to compute SE given such sampling design is the jackknife method. The authors could provide a table summarizing the data sources by sampling design and the approaches used to compute the SE.

As correctly raised, we our calculations of study-specific means and standard errors took into account (complex) sampling. We have expanded this statement to state how complex survey design was taken into account for calculation of mean and standard error (P. 22-23, Lines 704-711).

About data availability, it would be nice to list how many data points are available for female and male over time in each country and by residence. Sometimes, the female and male questionnaires have different focuses and qualities. Sometimes the male questionnaire may not be available for a certain country-year whereas the female one is available.

This information is available in Supplementary Table 2 at the country level. We have now added summaries in Supplementary Figures 1 and 2. The summaries are at the regional level, simply because the country equivalent of this figure would have 200 rows for each panel replicating the information in Supplementary Table 2. If of interest, we can easily generate the figure at the country level.

About the model, the authors mentioned on line 674-675 that "all analyses were done separately by sex because age, geographical and temporal patterns of height or BMI differ between girls and boys". When using the term "separately", I assume it actually means "independently". I do agree that the levels of BMI between females and males are most likely different. However, this cannot be used as the reason to treat the female and male BMI independently. If the data availability are different between females and males, modeling the two sexes together can actually help to inform at least the pattern of the sex with lower data availability. But of course one needs to first assume the patterns (not levels) of height and BMI between females and males are similar over time.

This is a good comment with which we have considered since we first designed our Bayesian model⁵⁰. As seen in Supplementary Figures 1 and 2, we have similar number of studies for boys and girls, and often from the same studies (Supplementary Table 2). Despite this balance, as the Referee points out, pooling female and male data could use the data more efficiently. At the same time, our model is relatively complex, with components for geography (hierarchy), time, age and place of residence, with time, age and place of residence having a hierarchical structure and, in some cases, interacting. Each of these components can vary in relation to sex, making a joint female and male model that allows all such interactions much more complex. We would be concerned about robustly fitting such a model if we have "too many" interactions, and misspecification if we have "too few".

About the MCMC computing, line 675-680: could the authors add the information on how many parallel chains they used for the MCMC algorithm? In addition, since the posterior samples are based on MCMC, the authors need to explain how did they check the model convergence. If the results are based on non-converged parameters, the results are not reliable.

We have provided this information including how we monitored model convergence (P. 35-36, Lines 1014-1042).

Referee #4

Summary of the key results

- This paper brings together a huge number of population based surveys to estimate changes in height and BMI over the period from 1990 to 2020. Despite the impressive achievement of pulling thousands of surveys together, I'm struggling to find a way to adequately summarize the results, since there seems to be a large degree of heterogeneity in the findings. Aside from "the urban height advantage got smaller and urban-rural differences in BMI remained generally small", which is fine as it goes, but there are important caveats to the pattern that make it very hard to argue that the paper provides much basis for action, despite its value for benchmarking and assessing trends.

Global analyses are influential in motivating and targeting policies and programmes that are needed to address various health outcomes and risk factors, both through identifying where action is needed, and distinguishing good versus poor performance, i.e., benchmarking.

Our paper addresses growth and development in ages that have traditionally received limited attention, consistently for all countries in the world so that comparisons can be made across them. It reports outcomes by rural and urban place of residence, which is a novel and important distinction, because many countries have distinct administrative and fiscal systems, and at times specific programmes, for cities and for rural areas.

Priority setting and benchmarking of course need to be accompanied with data on policy and intervention effectiveness, either in randomised studies or in programme/policy evaluations, now stated in the both the introduction (P. 3, Lines 51-55) and the concluding paragraph of our paper (P. 13, Lines 299-314).

Originality and significance: if not novel, please include reference

- The data assembled are original in the sense that the NCD consortium has undertaken an enormous and labour-intensive effort to assemble a huge number of population-based surveys, but this was not done specifically for this paper. They have previously published a similar paper (*Nature* 2019; 569(7755): 260–264.) on secular trends in BMI by rural and urban status, so it is a little unclear to me how much is added by now doing a separate paper that also includes height. In some respects I think the paper would be easier to read and understand if it focused only on height, since a lot of the BMI results (and message) are similar to the prior paper.

The 2019 paper in *Nature* focused on adults. The current paper focuses on school-aged children and adolescents, a group that has been mostly left out the global health policy, as recognised in recent years⁵¹⁻⁵³.

In the current paper, we included height and BMI because they are both important markers of growth and development, and health over the life course. When we started our analysis, we wondered how the BMI results for these understudied ages would compare those of adults. As seen in the paper, the results are distinct, and in some regions “opposite of the convergence in BMI of adults” (P. 11, Line 260-263) which led us to conduct the additional analyses (Extended Data Figure 8) on young adulthood as a transition period (PP. 11-13, Lines 263-269).

With two outcomes, we have tried to present the results so that important and novel patterns are highlighted, and of course welcome suggestions for alternative presentations.

- The significance is difficult to characterize. On the one hand, it is important to have as up to date and current descriptions of population trends in factors such as height and BMI cross-classified by gender. On the other hand, I don't find the evidence in the paper is sufficient to back up the significant policy action the authors suggest in much of the discussion.

As stated under related comment by Referee #1, we complemented insights from our own results on variations in rural and urban height and BMI with broader literature on policies and programmes that had reduced or amplified disparities to identify both where action is needed and what can be done. As also stated under related comment by Referee #1, we have moderated the recommendations, and have been explicit when we used broader knowledge on policy impacts.

Data & methodology: validity of approach, quality of data, quality of presentation

- In general, the methods are of high quality, and mostly transparently reported. I did not find the graphs or maps to be especially compelling (the panel figures are better), but then again it is quite hard to show changes over time in a spatially compelling way.

We tried to use a range of figures, in the main paper and Extended Data, that best show the results of interest to different readers. Maps allow users to visually compare results

within and across regions. For this reason, they tend to be of interest to readers across the globe who want to compare their country/region with others. But they do not numerically show specific results. We have added more numerical panels, which also include measures of uncertainty, along with maps in Figures 2 and 3, as well as the Supplementary Tables 3 and 4 for numerical reporting.

- p21. Happy to see the authors provide a link to the code for models for transforming the categorical BMI studies to a mean, but I do not see a more general link to where the entire pipeline for reproducing the tables and figures in the paper is available. I see a note that the details for the Bayesian model will be posted, but will this be enough for a third party to reproduce exactly (MCMC error aside) the estimates in the paper?

Upon publication of each paper, we provide a file online that either contains the input data to the Bayesian model or, for those data with specific governance requirements that prevent 3rd party data sharing, contacts for the data custodians (see examples at ncdrisc.org/downloads/bmi-urban-rural/NCD_RisC_Nature_2019_input_data.xlsx). These are provided in a structured and machine-readable format. To our knowledge, our provision of input data and access information goes beyond other global analyses. We of course would like to provide all input data but are prevented by governance arrangements (legal and/or data sharing contracts) of specific sources. We have also stated that we will additionally provide the pipeline for generation of figures and tables (P. 39, Lines 1104-1106), although these are not as such methodological.

- p23. "data driven fixed effects" is confusing and readers will have a hard time knowing what "data driven" means in this context. Plus, it isn't clear whether "fixed-effect and random-effect terms for subnational and community data" means this was done with respect to whether the sample was subnational or community. What is the difference? Is it just saying that any subnational estimate was treated as a separate stratum with a fixed effect, i.e., sub-national = 1 for any subnational estimate and 0 otherwise? But what then were the random effects? I had trouble following this.

We have clarified the distinction between subnational and community data (P. 22, Lines 686-690), and provided additional details on the fixed and random effects that characterise systematic differences as well as additional variability of these sources compared to national data (P. 33, Lines 958-968).

- I didn't see any discussion of the priors used for the Bayesian models. Were they flat? Uninformative? Informative based on substantive knowledge? For this volume of data it's clear that the data will likely overwhelm the prior, but as a matter of good practice it would seem important to at least make this clear to readers, especially those familiar with Bayesian inference. In fact, one of the Nature journals has recently published a review of good practices for reporting Bayesian analysis (Kruschke, Nat Hum Behav 2021) but not much of that guidance has been followed here. Some aspects might be hard for this paper, but at least reporting on priors and computational performance (convergence, chain mixing, etc.) should be straightforward.

We have added significantly more details on the statistical model, including on the priors and features of model implementation like convergence monitoring (P. 26-37, Lines 779-1054).

- p24. Why are the authors age-standardizing these estimates? If part of what the authors want to do with the paper is to assess how the patterns of height and BMI have changed over time, why resort to a fictitious metric that generates counterfactual, rather than actual heights and BMIs over time? They may wish to argue for standardizing, but given that the discussion contains a number of suggestions for policy action, any policies are likely to have to consider the actual populations at risk rather than what the urban-rural height difference would be *if* rural and urban areas had age distributions identical to the WHO standard. Given the narrow age range it may not make any difference, but it would seem at least relevant to assess the impact of age-standardizing on the pattern and changes over time.

As the Referee correctly points out, age-standardised results place all populations on a standard age distribution. The alternatives are to show crude all-age estimates or a series of age-specific estimates. The former is the “real” population but is not comparable over space and time when the outcome has an age association, as height and BMI do (i.e., children grow taller with age). For example, if mean age of children and adolescents in a country A is higher than in country B, crude mean height in country A could be larger (taller) than in country B, even if at every single age, children in country A are shorter than in B; rather the higher crude mean height would be only due age differentials between the two countries. Similarly, in the case of decreasing fertility, which leads to an older stock of children/adolescents over time, crude mean height can increase even if at every age height stays the same. In our experience, using crude estimates, which combine demographic and epidemiological variations, can be misleading and crude estimates are commonly misinterpreted as pure epidemiological change. Using age-specific estimates avoids this issue but would mean that a separate result is presented for each age, which would overwhelm the figures and text.

We have now complemented all age-standardised results with Extended Data Figures and Supplementary Figures that show the same result at key index ages of 5, 10, 15 and 19 years (Extended Data Figure 4 and Supplementary Figures 3-4) and highlighted any notable differences over age. We also show age-specific results in Supplementary Tables 3-4. We will also place all age-specific estimates (in 1-year age increments) online upon publication. We would welcome suggestions for alternative/additional presentations.

Appropriate use of statistics and treatment of uncertainties

- In general I found the main report did not emphasize reporting the magnitude and uncertainty of the main objectives (i.e., heights, BMIs, urban-rural differences and changes over time). Most of the results were framed as "the largest height differences were in countries X, Y, Z" but did not report anything regarding the magnitude and uncertainty of those estimates. Instead, the authors seem to focus a lot of attention on what one could argue is a more binary kind of outcome: e.g., the probability of whether the change in height between 1990 and 2020 was larger in rural vs. urban areas. It isn't clear to me

how this helps readers get a sense of the public health importance of these differences. Don't we want to know what those differences are? I think providing estimates of uncertainty would be particularly helpful given the lack of data for sub-Saharan Africa and the role that this region plays in the results.

As appropriately raised in this comment, we have added to Figures 2 and 3, and throughout results, a range of measures of uncertainty to the extent that the results remained readable. We would be happy to provide further numerical values if the Referee and Editors believe that doing so would not impede readability.

- More generally, I don't understand why the authors are reporting on the posterior probability that, for example, the urban-rural difference in height (or BMI) is positive, rather than giving readers an estimate and credible interval for the magnitude of the difference. I understand that using the PP allows them to generate maps that provide a lens on that question, but this seems to me more a limitation of relying on maps for evidence communication. It would be helpful instead to have a table or figure for each country that shows the magnitude and precision of the estimate, not just the PP of whether or not the differences is positive (or greater in one group than another). Given the already long appendix, I don't think it would be too harmful to extend it a little further.

We originally reported posterior probabilities because it falls on a continuous scale from 0 to 1, allowing readers' "normative" criteria to decide the uncertainty threshold at which change is relevant. We have now created the suggested Tables (Supplementary Tables 3-4), and added other measures of uncertainty, including 95% credible intervals and posterior standard deviations (equivalent to SE in a frequentist framework) to Figures 2 and 3 and to other Extended Data Figures and Supplementary Figures. We would be happy to provide additional information on uncertainty if they help readers to better grasp the extent of statistical confidence versus uncertainty in the results.

Conclusions: robustness, validity, reliability

- In general I found that a number of the conclusions went beyond (sometimes considerably beyond) the data and evidence presented in the paper. For example, even in the summary the authors say that, based on their report, there is an "urgent need" for policies to reduce income/infrastructure inequalities across urban/rural regions, yet they have not provided any evidence in the paper that the changes they observe are due to differences in income/infrastructure. I think the same goes for nutrition/sport programs and health services. To be clear, I tend to agree with the authors that such policies are likely to be beneficial, but I think this empirical paper does not provide sufficient evidence to back those claims. What to do about these changes in heights and BMIs feels like a different paper than this one describing trends.

Please see related comment under Referee #1 for how we have dealt with moderating our policy recommendations as raised by both Referees.

- This concern is echoed as much of the introduction seems to rest on strong assumptions about the importance of relevant global patterns for local

interventions. What evidence is there that "attention" to the heterogeneity of the urban-rural gradient across countries is useful or necessary for policy and programme development? Policies and programs for whom? Globally? At the country or municipality level? If there is compelling evidence that such large scale measurement and surveys are effective for designing policy at the country or subnational level, it would be good to see that argument made.

Comparative multi-country and global (macro-level) analyses of health outcomes should indeed go hand-in-hand with (micro-level) evaluation of specific interventions which tend to be done in specific populations and at specific points in time. Our discussion and recommendations attempted to bring the two together, by putting our novel results in the context of policies implemented in places with urban-rural convergence versus divergence. While we enormously value micro-level analysis, we also note that to focus only on policy evaluation without benchmarking of outcomes would risk missing wider, and at times larger, patterns or trends, which can only be seen in multi-country/global analyses. Areas of global health as varied as maternal and child mortality, tobacco and alcohol control, and cardiovascular disease prevention have shown that macro patterns and trends can reveal places with good practice, and generate opportunities and hypotheses for further investigation of effective actions.

- I also think that, though it is mentioned in the limitations section, the authors should probably address the issue of migration with greater care. What is the likely role that rural-to-urban migration may play in these trends. If particular kinds of populations are migrating from rural-to-urban areas (e.g., smaller rural children accompanying parents migrating to urban areas), couldn't this meaningfully contribute to the change in the urban-rural difference? Given the pace of urbanization of the past few decades, this feels like it needs more attention in the paper as a plausible explanation for some of the results. Or, if there is evidence to refute migration as an explanation, it should be discussed.

We have raised the possibility that, in some countries, migration may have played a role alongside changes in food, income, healthcare and the living environment in rural and/or urban areas (P. 12, Lines 283-287), while noting that even total change in the share of population who are urban or rural does not seem to be correlated with the change in the height or BMI gap (Figure below).

Figure: the relationship between change in the proportion of population who live in urban areas and change the urban-rural gap in height and BMI from 1990 to 2020.

Suggested improvements: experiments, data for possible revision

- Reporting on the magnitude and uncertainty of estimates rather than the PP of differences would improve the paper.

As above, we now report credible intervals and posterior standard deviations in addition to posterior probabilities so that readers are able to see a range of measures of uncertainty.

- Authors should drop the specific appeals for policy interventions without much better evidence as to the causes of the patterns evident in the paper.

As stated above, and under related comments from Referee #1, we have moderated our policy recommendations.

- More discussion of migration or alternative explanations for the pattern observed.

As detailed in response to an earlier comment, we have done so.

References: appropriate credit to previous work?

- Yes

Clarity and context: lucidity of abstract/summary, appropriateness of abstract, introduction and conclusions
- Yes

Other revisions

Since submission, we have downloaded or received from NCD-RisC collaborators additional data sources. We have included these data in the NCD-RisC database and re-run our statistical model. The additional data provide additional information for some countries but result in only small changes to the numerical results; the conclusions remain unchanged.

Editors' comments

We have addressed the issues raised by the Editors about the paper as stated in response to Referees' comments. We have also followed the instructions on statistics and reproducibility, extended data, supplementary data, and source data to the best of our understanding of what is required. We would be happy to further modify as needed to meet the Journal's requirements.

References

- 1 Singh, G. M. *et al.* Global, Regional, and National Consumption of Sugar-Sweetened Beverages, Fruit Juices, and Milk: A Systematic Assessment of Beverage Intake in 187 Countries. *PLoS One* **10**, e0124845, doi:10.1371/journal.pone.0124845 (2015).
- 2 Miller, V. *et al.* Global, regional, and national consumption of animal-source foods between 1990 and 2018: findings from the Global Dietary Database. *The Lancet Planetary Health* **6**, e243-e256, doi:10.1016/S2542-5196(21)00352-1 (2022).
- 3 Guthold, R., Stevens, G. A., Riley, L. M. & Bull, F. C. Global trends in insufficient physical activity among adolescents: a pooled analysis of 298 population-based surveys with 1.6 million participants. *The Lancet Child & Adolescent Health* **4**, 23-35, doi:10.1016/S2352-4642(19)30323-2 (2020).
- 4 Arsenault, C. *et al.* Equity in antenatal care quality: an analysis of 91 national household surveys. *The Lancet Global Health* **6**, e1186-e1195, doi:[https://doi.org/10.1016/S2214-109X\(18\)30389-9](https://doi.org/10.1016/S2214-109X(18)30389-9) (2018).
- 5 Galles, N. C. *et al.* Measuring routine childhood vaccination coverage in 204 countries and territories, 1980–2019: a systematic analysis for the Global Burden of Disease Study 2020, Release 1. *The Lancet* **398**, 503-521, doi:[https://doi.org/10.1016/S0140-6736\(21\)00984-3](https://doi.org/10.1016/S0140-6736(21)00984-3) (2021).
- 6 Baltag, V., Pachyna, A. & Hall, J. Global overview of school health services: data from 102 countries. *Health Behavior and Policy Review* **2**, 268-283 (2015).
- 7 Kruk, M. E. *et al.* Quality of basic maternal care functions in health facilities of five African countries: an analysis of national health system surveys. *The Lancet Global Health* **4**, e845-e855, doi:10.1016/S2214-109X(16)30180-2 (2016).
- 8 Freisling, H. *et al.* Dietary reporting errors on 24 h recalls and dietary questionnaires are associated with BMI across six European countries as evaluated with recovery biomarkers for protein and potassium intake. *British Journal of Nutrition* **107**, 910-920, doi:10.1017/S0007114511003564 (2012).
- 9 Lissner, L. *et al.* OPEN about obesity: recovery biomarkers, dietary reporting errors and BMI. *Int J Obes (Lond)* **31**, 956-961, doi:10.1038/sj.ijo.0803527 (2007).
- 10 Bauman, A. *et al.* Progress and pitfalls in the use of the International Physical Activity Questionnaire (IPAQ) for adult physical activity surveillance. *J Phys Act Health* **6 Suppl 1**, S5-8, doi:10.1123/jpah.6.s1.s5 (2009).
- 11 Gibson, R. S., Charrondiere, U. R. & Bell, W. Measurement Errors in Dietary Assessment Using Self-Reported 24-Hour Recalls in Low-Income Countries and Strategies for Their Prevention. *Adv Nutr* **8**, 980-991, doi:10.3945/an.117.016980 (2017).
- 12 Freedman, L. S. *et al.* Pooled results from 5 validation studies of dietary self-report instruments using recovery biomarkers for energy and protein intake. *Am J Epidemiol* **180**, 172-188, doi:10.1093/aje/kwu116 (2014).
- 13 Donnelly, J. E. *et al.* Effects of a 16-month randomized controlled exercise trial on body weight and composition in young, overweight men and women: the Midwest Exercise Trial. *Arch Intern Med* **163**, 1343-1350, doi:10.1001/archinte.163.11.1343 (2003).
- 14 Willett, W. in *Implications of Total Energy Intake for Epidemiologic Analyses* (Oxford University Press, 2012).
- 15 King, N. A., Hopkins, M., Caudwell, P., Stubbs, R. J. & Blundell, J. E. Individual variability following 12 weeks of supervised exercise: identification and characterization of compensation for exercise-induced weight loss. *Int J Obes (Lond)* **32**, 177-184, doi:10.1038/sj.ijo.0803712 (2008).

- 16 Rosenkilde, M. *et al.* Body fat loss and compensatory mechanisms in response to different doses of aerobic exercise--a randomized controlled trial in overweight sedentary males. *Am J Physiol Regul Integr Comp Physiol* **303**, R571-579, doi:10.1152/ajpregu.00141.2012 (2012).
- 17 Stensel, D. Exercise, appetite and appetite-regulating hormones: implications for food intake and weight control. *Ann Nutr Metab* **57 Suppl 2**, 36-42, doi:10.1159/000322702 (2010).
- 18 Whybrow, S. *et al.* The effect of an incremental increase in exercise on appetite, eating behaviour and energy balance in lean men and women feeding ad libitum. *Br J Nutr* **100**, 1109-1115, doi:10.1017/s0007114508968240 (2008).
- 19 Wing, R. R. Physical activity in the treatment of the adulthood overweight and obesity: current evidence and research issues. *Med Sci Sports Exerc* **31**, S547-552, doi:10.1097/00005768-199911001-00010 (1999).
- 20 Hall, K. D. *et al.* Quantification of the effect of energy imbalance on bodyweight. *Lancet* **378**, 826-837, doi:10.1016/s0140-6736(11)60812-x (2011).
- 21 Daymont, C. *et al.* Automated identification of implausible values in growth data from pediatric electronic health records. *J Am Med Inform Assoc* **24**, 1080-1087, doi:10.1093/jamia/ocx037 (2017).
- 22 Davis, A. M., Bennett, K. J., Befort, C. & Nollen, N. Obesity and Related Health Behaviors Among Urban and Rural Children in the United States: Data from the National Health and Nutrition Examination Survey 2003–2004 and 2005–2006. *Journal of Pediatric Psychology* **36**, 669-676, doi:10.1093/jpepsy/jsq117 (2011).
- 23 World Health Organization. Report of the commission on ending childhood obesity. (World Health Organization, Geneva, 2016).
- 24 Pirgon, Ö. & Aslan, N. The Role of Urbanization in Childhood Obesity. *Journal of clinical research in pediatric endocrinology* **7**, 163-167, doi:10.4274/jcrpe.1984 (2015).
- 25 World Health Organization & Habitat, U. N. Global report on urban health: equitable healthier cities for sustainable development. Report No. 9789241565271, (World Health Organization, Geneva, 2016).
- 26 Smith, D. M. & Cummins, S. Obese Cities: How Our Environment Shapes Overweight. *Geography Compass* **3**, 518-535, doi:10.1111/j.1749-8198.2008.00198.x (2009).
- 27 Fraser, B. Latin America's urbanisation is boosting obesity. *Lancet* **365**, 1995-1996, doi:10.1016/s0140-6736(05)66679-2 (2005).
- 28 Kirchengast, S. & Hagmann, D. "Obesity in the City" – urbanization, health risks and rising obesity rates from the viewpoint of human biology and public health. *Human Biology and Public Health* **2**, doi:10.52905/hbph.v2.11 (2021).
- 29 Congdon, P. Obesity and Urban Environments. *International Journal of Environmental Research and Public Health* **16**, doi:10.3390/ijerph16030464 (2019).
- 30 Gong, P. *et al.* Urbanisation and health in China. *Lancet* **379**, 843-852, doi:10.1016/S0140-6736(11)61878-3 (2012).
- 31 UNICEF. The state of the world's children 2012: children in an urban world. (New York, 2012).
- 32 McDonald, C. M. *et al.* The effect of multiple anthropometric deficits on child mortality: meta-analysis of individual data in 10 prospective studies from developing countries. *Am J Clin Nutr* **97**, 896-901, doi:10.3945/ajcn.112.047639 (2013).
- 33 Park, M. H., Falconer, C., Viner, R. M. & Kinra, S. The impact of childhood obesity on morbidity and mortality in adulthood: a systematic review. *Obes Rev* **13**, 985-1000, doi:10.1111/j.1467-789X.2012.01015.x (2012).

- 34 Nüesch, E. *et al.* Adult height, coronary heart disease and stroke: a multi-locus Mendelian randomization meta-analysis. *Int J Epidemiol* **45**, 1927-1937, doi:10.1093/ije/dyv074 (2016).
- 35 Global BMI Mortality Collaboration. Body-mass index and all-cause mortality: individual-participant-data meta-analysis of 239 prospective studies in four continents. *The Lancet* **388**, 776-786, doi:10.1016/S0140-6736(16)30175-1 (2016).
- 36 Singh, G. M. *et al.* The Age-Specific Quantitative Effects of Metabolic Risk Factors on Cardiovascular Diseases and Diabetes: A Pooled Analysis. *PLOS ONE* **8**, e65174, doi:10.1371/journal.pone.0065174 (2013).
- 37 Asia Pacific Cohort Studies Collaboration. Body mass index and cardiovascular disease in the Asia-Pacific Region: an overview of 33 cohorts involving 310 000 participants. *International Journal of Epidemiology* **33**, 751-758, doi:10.1093/ije/dyh163 (2004).
- 38 Smith, L. C., Ruel, M. T. & Ndiaye, A. Why Is Child Malnutrition Lower in Urban Than in Rural Areas? Evidence from 36 Developing Countries. *World Development* **33**, 1285-1305, doi:10.1016/j.worlddev.2005.03.002 (2005).
- 39 Manley, J., Alderman, H. & Gentilini, U. More evidence on cash transfers and child nutritional outcomes: a systematic review and meta-analysis. *BMJ Global Health* **7**, e008233, doi:10.1136/bmjgh-2021-008233 (2022).
- 40 Bentham, J. *et al.* Multidimensional characterization of global food supply from 1961 to 2013. *Nature Food* **1**, 70-75, doi:10.1038/s43016-019-0012-2 (2020).
- 41 Kray, H. A., Heumesser, C., Mikulcak, F., Giertz, Å. & Bucik, M. Productive Diversification in African Agriculture and its Effects on Resilience and Nutrition. (The World Bank, 2018).
- 42 Godha, D., Zafimanjaka, M., Bambara, E., Likhite, N. & Tharaney, M. Determinants of adolescent nutritional status and practices in Burkina Faso: A pooled secondary analysis. *Field Exchange* **66**, 68 (2021).
- 43 Weatherspoon, D. D., Miller, S., Ngabitsinze, J. C., Weatherspoon, L. J. & Oehmke, J. F. Stunting, food security, markets and food policy in Rwanda. *BMC Public Health* **19**, 882, doi:10.1186/s12889-019-7208-0 (2019).
- 44 Cockx, L., Colen, L., De Weerd, J., Gomez Y Paloma, S. *Urbanization as a driver of changing food demand in Africa: Evidence from rural-urban migration in Tanzania*. (Publications Office of the European Union, 2019).
- 45 Fraval, S. *et al.* Food Access Deficiencies in sub-Saharan Africa: Prevalence and Implications for Agricultural Interventions. *Frontiers in Sustainable Food Systems* **3**, doi:10.3389/fsufs.2019.00104 (2019).
- 46 Frisch, R. E. & Revelle, R. Height and Weight at Menarche and a Hypothesis of Critical Body Weights and Adolescent Events. *Science* **169**, 397-399, doi:10.1126/science.169.3943.397 (1970).
- 47 Holmgren, A. *et al.* Pubertal height gain is inversely related to peak BMI in childhood. *Pediatric Research* **81**, 448-454, doi:10.1038/pr.2016.253 (2017).
- 48 Alhassan, J. A. K., Adeyinka, D. A. & Olakunde, B. O. Equity dimensions of the decline in under-five mortality in Ghana: a joinpoint regression analysis. *Tropical Medicine & International Health* **25**, 732-739, doi:<https://doi.org/10.1111/tmi.13391> (2020).
- 49 Wolde, K. S. & Bacha, R. H. Trend and correlates of under-5 mortality in Ethiopia: A multilevel model comparison of 2000-2016 EDHS data. *SAGE Open Med* **10**, 20503121221100608, doi:10.1177/20503121221100608 (2022).

- 50 Finucane, M. M., Paciorek, C. J., Danaei, G. & Ezzati, M. Bayesian Estimation of Population-Level Trends in Measures of Health Status. *Statistical Science* **29**, 18-25 (2014).
- 51 Black, R. E. *et al.* Health and development from preconception to 20 years of age and human capital. *Lancet* **399**, 1730-1740, doi:[https://doi.org/10.1016/S0140-6736\(21\)02533-2](https://doi.org/10.1016/S0140-6736(21)02533-2) (2022).
- 52 Patton, G. C. *et al.* Nourishing our future: the Lancet Series on adolescent nutrition. *The Lancet* **399**, 123-125, doi:10.1016/S0140-6736(21)02140-1 (2022).
- 53 Norris, S. A. *et al.* Nutrition in adolescent growth and development. *The Lancet* **399**, 172-184, doi:10.1016/S0140-6736(21)01590-7 (2022).

Reviewer Reports on the First Revision:

Referee #1 (Remarks to the Author):

Thank you for the responding to comments and for revising the manuscript.

The conclusion could still be improved. The final sentence is a run on and does not make sense. What is "deteriorating global poverty and food crises"? What does "as macro policies are implemented....there is an urgent need..." Suggest using plain language to make whatever statement is intended, perhaps in 2 sentences. And, relate these final sentences directly to what was done in this analysis. I agree with reviewer 4 on the implications of the analyses and the lack of a basis for action.

Current sentence is below:

"Second, as these macro policies are implemented and with deteriorating global poverty and food crises, based on the evidence from specific countries on the benefits of targeted nutrition programmes and health and social services^{58,107-115}, there is an urgent need for programmes that equalise nutrition, and health and social services for children and adolescents at home, school and in the community."

Referee #2 (Remarks to the Author):

The revised version has addressed all of my previous concerns.

Referee #4 (Remarks to the Author):

This version is improved. I like the inclusion of some estimates of the magnitude of the urban-rural differences in the main text, as well as for some of the changes over time. I think this helps readers get a better sense of whether these changes are important.

The authors have also improved the reporting and transparency of the models and some of the modeling decisions that created the final results, as well as including more on potential explanations for the pattern of results (e.g., selective survival, selective migration).

Policy prescriptions. The authors have been somewhat responsive to multiple reviewer concerns about policy conclusions and prescriptions that were not justified by the evidence in the paper. In particular, I think taking out the "...urgent need..." paragraph of the Introduction is certainly a step in the right direction.

However, in much of the Discussion and concluding section, the authors say that they have "moderated" their language around policy, though I'm not sure that I agree, and I still think these statements are needlessly overreaching. In the original version they stated that there was "a need

for...policies that tackle poverty and address infrastructure inequalities", and now they have simply revised this to say that "the extant evidence..indicates that there is a need..." with exactly the same policy prescriptions. They cite Smith et al. to support this statement, but that paper not only says that associations between HAZ and socioeconomic factors are generally of a relatively low magnitude, it actually states [p.1301], "Thus, to a large extent, our overall hypothesis of the existence of fundamental urban–rural differences in the socioeconomic determinants of children’s nutritional status was not confirmed by our multicountry analysis." So how does this advance the authors case? I don't think it does, since the Smith et al. paper shows, at best, weak evidence on associations between individual-level socioeconomic factors and nutritional outcome (without a credible design to estimate their causal effects), and no evidence at all on poverty, poverty policies, or infrastructure. So I still don't think the authors statement about the need for these (vague, undefined) policies is justified and needlessly detracts from the main message of the paper.

This same pattern is evident for the other policies the authors also say are urgent to equalise nutrition and health / social services. If they really want to talk about this, then there has to be a discussion of specific policy environments and programs that could help to address the diverse and heterogenous findings they report across countries. I know I'm being critical here, but I just don't think that this improves the paper, precisely because there isn't room given all of the other findings to devote space to a serious discussion of policy. I think the paper would be much better served by a more in-depth consideration of what is driving these changes (i.e., what explains the patterns of descriptive findings). Rather than mention factors such as changes in puberty, differential survival, and migration in a single sentence regarding limitations, the paper would be stronger if they could describe precisely how changes in these factors may or may not have affected the pattern of results, at least qualitatively, if not quantitatively.

Author Rebuttals to First Revision:

We thank the Referees for reading the revised manuscript and for their additional comments and suggestions. We have used these to revise the paper, as detailed below. All page, line, figure and citation numbers refer to the tracked manuscript.

Referee #1

Thank you for the responding to comments and for revising the manuscript.

The conclusion could still be improved. The final sentence is a run on and does not make sense. What is "deteriorating global poverty and food crises"? What does "as macro policies are implemented....there is an urgent need..." Suggest using plain language to make whatever statement is intended, perhaps in 2 sentences. And, relate these final sentences directly to what was done in this analysis. I agree with reviewer 4 on the implications of the analyses and the lack of a basis for action.

Current sentence is below:

"Second, as these macro policies are implemented and with deteriorating global poverty and food crises, based on the evidence from specific countries on the benefits of targeted nutrition programmes and health and social services^{58,107-115}, there is an urgent need for programmes that equalise nutrition, and health and social services for children and adolescents at home, school and in the community."

We have restructured and revised the final paragraph of the paper to simply raise the need for action, which is related to what was done in this paper, rather than specifying the sorts of actions that can be taken, which had drawn on other works (P. 14, Lines 311-319).

Referee #2

The revised version has addressed all of my previous concerns.

We appreciate the confirmation that revisions were satisfactory.

Referee #4

This version is improved. I like the inclusion of some estimates of the magnitude of the urban-rural differences in the main text, as well as for some of the changes over time. I think this helps readers get a better sense of whether these changes are important.

The authors have also improved the reporting and transparency of the models and some of the modeling decisions that created the final results, as well as including more on potential explanations for the pattern of results (e.g., selective survival, selective migration).

Policy prescriptions. The authors have been somewhat responsive to multiple reviewer concerns about policy conclusions and prescriptions that were not justified by the evidence in the paper. In particular, I think taking out the "...urgent need..." paragraph of the Introduction is certainly a step in the right direction.

However, in much of the Discussion and concluding section, the authors say that they have "moderated" their language around policy, though I'm not sure that I agree, and I still think these statements are needlessly overreaching. In the original version they stated that there was "a need for...policies that tackle poverty and address infrastructure inequalities", and now they have simply revised this to say that "the extant evidence..indicates that there is a need..." with exactly the same policy prescriptions. They cite Smith et al. to support this statement, but that paper not only says that associations between HAZ and socioeconomic factors are generally of a relatively low magnitude, it actually states [p.1301], "Thus, to a large extent, our overall hypothesis of the existence of fundamental urban–rural differences in the socioeconomic determinants of children’s nutritional status was not confirmed by our multicountry analysis." So how does this advance the authors case? I don't think it does, since the Smith et al. paper shows, at best, weak evidence on associations between individual-level socioeconomic factors and nutritional outcome (without a credible design to estimate their causal effects), and no evidence at all on poverty, poverty policies, or infrastructure. So I still don't think the authors statement about the need for these (vague, undefined) policies is justified and needlessly detracts from the main message of the paper.

This same pattern is evident for the other policies the authors also say are urgent to equalise nutrition and health / social services. If they really want to talk about this, then there has to be a discussion of specific policy environments and programs that could help to address the diverse and heterogenous findings they report across countries. I know I'm being critical here, but I just don't think that this improves the paper, precisely because there isn't room given all of the other findings to devote space to a serious discussion of policy. I think the paper would be much better served by a more in-depth consideration of what is driving these changes (i.e., what explains the patterns of descriptive findings). Rather than mention factors such as changes in puberty, differential survival, and migration in a single sentence regarding limitations, the paper would be stronger if they could describe precisely how changes in these factors may or may not have affected the pattern of results, at least qualitatively, if not quantitatively.

We note that the conclusion of Smith paper¹ is not that these determinants don't drive urban-rural differences, but rather that their strengths of association (i.e., the magnitude of their effect) is the same in rural and urban areas. The full text of the conclusion from pages 1300-1301 reads as below (emphasis added):

“Our analysis of 36 DHS data sets from three regions of the developing world shows little evidence of differences in key socioeconomic determinants of child nutritional status—women’s education and status, access to safe water and sanitation, and household economic status— or in the strength of their

association between urban and rural areas. This is true across the three regions and for most of the determinants examined. *Where urban–rural differences in the strengths of associations are detected, they are usually of a small magnitude. ...*

Thus, to a large extent, our overall hypothesis of the existence of fundamental urban–rural differences in the socioeconomic determinants of children’s nutritional status was not confirmed by our multicountry analysis.

As expected, however, we found marked differences in the levels of the socioeconomic determinants themselves between urban and rural areas.”

That being said, the Smith paper is observational and the other evidence that we had used are country case studies. Therefore, we have restructured and revised the final paragraph of the paper to simply raise the need for action, which is related to what was done in this paper, rather than specifying the sorts of actions that can be taken, which had drawn on other works (P. 14, Lines 311-319). We have also expanded the text on explanations of the other factors that may have led to the results we see (P. 11-13, Lines 237-289), including economic determinants for results in high- and middle-income countries (P. 11, Lines 242-259), economic and nutritional factors specific to sub-Saharan Africa (P. 12, Lines 261-272) and demographic determinants (P. 12-13, Lines 278-289).

References

1 Smith, L. C., Ruel, M. T. & Ndiaye, A. Why Is Child Malnutrition Lower in Urban Than in Rural Areas? Evidence from 36 Developing Countries. *World Development* 33, 1285-1305, doi:10.1016/j.worlddev.2005.03.002 (2005).